# QuaDiM: A Conditional Diffusion Model For Quantum State Property Estimation

**Yehui Tang**[1]**, Mabiao Long**[1]**, Junchi Yan**[12*]

[1]Sch. of Computer Science & Sch. of Artificial Intelligence, Shanghai Jiao Tong University
[2]Shanghai Artificial Intelligence Laboratory
{yehuitang,albertlong007,yanjunchi}@sjtu.edu.cn

## Abstract

Quantum state property estimation (QPE) is a fundamental challenge in quantum many-body problems in physics and chemistry, involving the prediction of characteristics such as correlation and entanglement entropy through statistical analysis of quantum measurement data. Recent advances in deep learning have provided powerful solutions, predominantly using auto-regressive models. These models generally assume an intrinsic ordering among qubits, aiming to approximate the classical probability distribution through sequential training. However, unlike natural language, the entanglement structure of qubits lacks an inherent ordering, hurting the motivation of such models. In this paper, we introduce a novel, non-autoregressive generative model called **QuaDiM**, designed for **Qua**ntum state property estimation using **Di**ffusion **M**odels. QuaDiM progressively denoises Gaussian noise into the distribution corresponding to the quantum state, encouraging equal, unbiased treatment of all qubits. QuaDiM learns to map physical variables to properties of the ground state of the parameterized Hamiltonian during offline training. Afterwards one can sample from the learned distribution conditioned on previously unseen physical variables to collect measurement records and employ post-processing to predict properties of unknown quantum states. We evaluate QuaDiM on large-scale QPE tasks using classically simulated data on the 1D anti-ferromagnetic Heisenberg model with the system size up to 100 qubits. Numerical results demonstrate that QuaDiM outperforms baseline models, particularly auto-regressive approaches, under conditions of limited measurement data during training and reduced sample complexity during inference.

## 1 Introduction

Quantum state property estimation (QPE) is a pivotal and challenging problem in the field of quantum many-body physics (Carleo & Troyer, 2017; Torlai et al., 2018; Carrasquilla, 2020; Gebhart et al., 2023; Miles et al., 2023) and chemistry (Kandala et al., 2017; Schütt et al., 2019; Cao et al., 2019; Sajjan et al., 2022; Barrett et al., 2022). The primary goal is to accurately infer certain properties of the quantum state such as correlation and entanglement entropy using as few quantum measurements as possible. QPE has wide-ranging applications in quantum computing (Dunjko & Briegel, 2018; Torlai & Melko, 2020), quantum cryptography (Brakerski & Shmueli, 2019; Ananth et al., 2022), and quantum simulation (Jia et al., 2019; Schmitt & Heyl, 2020). As the number of qubits in a system grows, the dimensionality of the corresponding quantum state increases exponentially, making property estimation increasingly difficult. Traditional techniques, like quantum state tomography (D'Ariano et al., 2003; Jullien et al., 2014), become impractical for large systems due to their exponential computational overheads.

Recent advancements in machine learning have opened new avenues for QPE, leveraging the models to extract intricate features from quantum measurement records. An overview about the pipeline is provided in Fig. 1. A notable way relies on auto-regressive models, which assumes an inherent order among qubits (e.g. left-to-right for 1D chain, or ZigZag order for 2D lattice) and decompose

*Correspondence author. Work was partly supported by NSFC (62222607), Shanghai Municipal Science and Technology Major Project (2021SHZDZX0102).

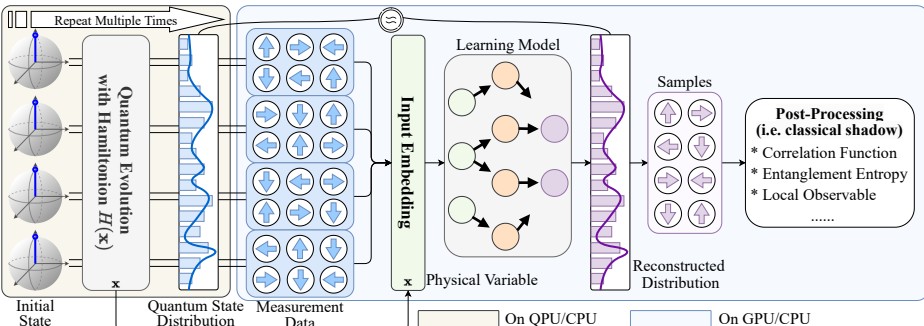

Figure 1: **An Overview of Quantum State Property Estimation.** Given the specified Hamiltonian $H(\mathbf{x})$ parameterized by $\mathbf{x}$, the quantum system initiates from state $|\psi_s\rangle$ and evolves controlled by $H(\mathbf{x})$ in either physical experiments on a quantum processing unit (QPU) or simulations on CPUs. Measurements are then performed on this final state, generating records stored in classical memory. The machine learning model is trained using measurement records conditioned on $\mathbf{x}$ and the model's output would reconstruct the distribution of $|\psi_s\rangle$. The properties of unseen states conditioned on new $\mathbf{x}$ such as fidelity can be obtained by post-processing the samples generated by the learned model.

their joint probability distribution into conditional probabilities. These approaches include RNN-based models (Carrasquilla et al., 2019; Hibat-Allah et al., 2020; Barrett et al., 2022), CNN-based models (Wu et al., 2019; Sharir et al., 2020) and transformer-based models (Cha et al., 2021; Wang et al., 2022; Du et al., 2023; Tang et al., 2024a;b).

In auto-regressive models, each qubit is conditioned on the states of previously modeled qubits, introducing an inherent bias due to the imposed order. While this method works well in domains where data naturally follows a sequential structure, such as language (Yang et al., 2019; Brown et al., 2020) or time series (Hewamalage et al., 2021), it comes with a significant limitation when applied to quantum systems, where qubit interactions are *non-sequential* and the complex entanglement does not adhere to a predefined order. Based on our experimental results, the reliance on sequential modeling could lead to oversimplified representations of the entanglement structure, limiting the model's ability to fully capture the intricate correlations between qubits. Consequently, modeling quantum systems with sequential dependencies introduces biases that may fail to capture the true complexity of the entanglement and correlations, leading to suboptimal estimations of quantum properties.

To address these challenges, we propose **QuaDiM** (**Qua**ntum state property estimation using **Di**ffusion **M**odels), a novel *non-autoregressive* generative model based on diffusion models (Ho et al., 2020; Dhariwal & Nichol, 2021; Song et al., 2021a;c). Unlike auto-regressive approaches, QuaDiM treats all qubits equally without imposing a predefined order, thereby avoiding potential biases. The model learns a mapping from physical variables, i.e. the parameters of a system's Hamiltonian, to the ground-state properties associated with those variables. Once trained, QuaDiM is capable of generalizing to previously unseen quantum systems, allowing it to predict quantum state properties for systems not encountered during training. This generalization capability is particularly valuable in practical scenarios where the quantum states of interest may cannot be prepared on modern noisy intermediate-scale quantum (NISQ) hardware, or quantum states difficult to simulated classically.

The underlying mechanism of QuaDiM is rooted in diffusion models, which iteratively denoise Gaussian noise to generate samples from the target distribution (Austin et al., 2021; Kong et al., 2021). In contrast with auto-regressive models, QuaDiM offers an advantage by treating all qubits equally throughout the denoising process, encouraging to remove the need for any predefined ordering of qubits. Mathematically, QuaDiM initiates from an random Gaussian distribution, which is progressively refined through a series of denoising steps to approximate the true distribution of the quantum state. This iterative process facilitates QuaDiM to capture correlations between qubits more effectively and without the biases introduced by a sequential models. By incrementally learning to reverse the noise process, the model naturally handles the complex interactions and entanglements inherent in many-body quantum systems. As a result, QuaDiM not only achieves higher accuracy in QPE but also demonstrates scalability when applied to large-scale systems up to 100 qubits. The model's iterative refinement mechanism encourages that all parts of the system are considered simultaneously, with the hope of being helpful for estimating properties like correlation and entanglement entropy, where the relationships between all qubits are equally important.

We validate the efficacy of QuaDiM through extensive numerical experiments on predicting the correlation and entanglement entropy of the 1D anti-ferromagnetic Heisenberg model with up to 100 qubits. The results show that QuaDiM outperforms baselines, especially auto-regressive approaches, in scenarios with limited training measurements data and reduced sample complexity. QuaDiM's ability to handle these challenging conditions highlights its potential for practical applications in quantum computing and related areas, where resource constraints and data scarcity are prevalent.

**Our contributions.** 1) We introduce QuaDiM, the first-ever (to the best of our knowledge) non-autoregressive conditional generative model for QPE. By utilizing diffusion models, QuaDiM iteratively denoises Gaussian noise to accurately approximate the distribution of unknown quantum states from limited measurements, encouraging equal and unbiased treatment of all qubits. 2) We classically simulate relatively large-scale quantum systems with up to 100 qubits to generate extensive training and test datasets for evaluation, showing QuaDiM's scalability and practical applicability. 3) We conduct experiments on QPE tasks involving the prediction of correlation and entanglement entropy. Our results show that QuaDiM achieves lower prediction errors compared to baselines, particularly outperforming auto-regressive models such as Carrasquilla et al. (2019) and Tang et al. (2024a).

## 2 RELATED WORK AND PRELIMINARIES

**Learning-based Quantum State Property Estimation.** In the domain of learning-based QPE, autoregressive models such as RNN-based (Carrasquilla et al., 2019; Hibat-Allah et al., 2020), CNN-based (Wu et al., 2019; Sharir et al., 2020), and transformer-based models (Cha et al., 2021; Wang et al., 2022; Du et al., 2023; Tang et al., 2024a) have been extensively explored. These models inherently assume a sequential ordering of qubits, introducing biases that could limit their ability to capture the complex entanglement and correlations in quantum systems where qubits interact non-sequentially. This sequential dependency may lead to oversimplified representations of the entanglement structure, hindering accurate property estimation.

Alternatively, generative models that learn variational wave functions—such as those based on deep Boltzmann machines (DBMs) (Gao & Duan, 2017; Nomura et al., 2017), variational autoencoders (VAEs) (Rocchetto et al., 2018), and generative adversarial networks (GANs) (Ahmed et al., 2021)—have been proposed to represent quantum states. While capable of capturing complex probability distributions, these models typically learn representations for a single specific quantum state and lack the ability to generalize to unseen states, restricting their applicability in predicting properties of new quantum systems. Although GANs (Ahmed et al., 2021) could, in principle, generalize to unknown quantum states, their learning objective involves reconstructing the density matrix. This significantly hinders scalability because the complexity of fully describing a density matrix increases exponentially w.r.t quantum system size, making such approaches impractical for large-scale systems.

**Quantum State and Quantum Measurement.** Quantum computing has the potential to foster advancements in a variety of fields, such as chemistry (Schütt et al., 2019; Cao et al., 2019; Lee et al., 2023), combinatorial optimization (Sanders et al., 2020; Ye et al., 2025), and machine learning (Biamonte et al., 2017; Sajjan et al., 2022; Tang & Yan, 2022; Xiong et al., 2024). A quantum bit, or *qubit*, serves as the fundamental unit in quantum computing. The collection of all qubits within a (sub)system constitutes the *quantum state*. Prior to measurement, a qubit exists in a superposition of states, but upon measurement, it collapses into a definite state. The mathematical representation of a quantum state depends on the choice of basis states. For instance, using the orthogonal *computational basis states* $|0\rangle = \left[\begin{smallmatrix} 1 \\ 0 \end{smallmatrix}\right]$ and $|1\rangle = \left[\begin{smallmatrix} 0 \\ 1 \end{smallmatrix}\right]$, a single qubit can be expressed as a linear combination $|\phi\rangle = \alpha|0\rangle + \beta|1\rangle = \left[\begin{smallmatrix} \alpha \\ \beta \end{smallmatrix}\right]$ in the complex vector space $\mathbb{C}^2$, where $\alpha, \beta \in \mathbb{C}$ are complex amplitudes satisfying the normalization condition $|\alpha|^2 + |\beta|^2 = 1$. An alternative representation of a quantum state employs the *density operator* or *density matrix*. For example, the density matrix corresponding to $|0\rangle$ is given by $\rho_0 = |0\rangle\langle 0| = \left(\begin{smallmatrix} 1 & 0 \\ 0 & 0 \end{smallmatrix}\right)$, where $\langle 0|$ denotes the conjugate transpose of $|0\rangle$.

A general quantum state comprising $L$ qubits can be represented by a *wave function*:

$$|\psi\rangle = \sum_{\sigma_1=1}^{K} \cdots \sum_{\sigma_L=1}^{K} \boldsymbol{\Psi}(\sigma_1, \ldots, \sigma_L)|\sigma_1, \ldots, \sigma_L\rangle, \tag{1}$$

where $\boldsymbol{\Psi} : \mathbb{Z}^L \to \mathbb{C}$ assigns a complex amplitude to each fixed configuration $\boldsymbol{\sigma} = (\sigma_1, \ldots, \sigma_L)$ of the $L$ qubits, satisfying the normalization condition $\sum_{\sigma_1=1}^{K} \cdots \sum_{\sigma_L=1}^{K} |\boldsymbol{\Psi}(\sigma_1, \ldots, \sigma_L)|^2 = 1$. Each

$\sigma_i \in \{1, \ldots, K\}$ represents one of the $K$ possible outcomes when a measurement is performed on the $i$-th qubit. The wave function resides in a complex Hilbert space, with the vector representation $|\psi\rangle \in \mathbb{C}^{K^L}$ and its density matrix $|\psi\rangle\langle\psi| \in \mathbb{C}^{K^L \times K^L}$, both of which grow exponentially large with increasing $L$. In this paper, we consider the Pauli-6 measurements such that $K = 6$.

Quantum measurements translate aspects of quantum information into classical data for subsequent processing, utilizing a set of *measurement operators* $\{\mathbf{O}_k\}_{k=1}^K$ that satisfy the completeness relation $\sum_k \mathbf{O}_k = \mathbf{I}$, where $K$ denotes the total number of possible outcomes. Measuring a qubit yields different results corresponding to the indices $k$ of the measurement operators. Specifically, for a quantum state $\rho$, the probability of obtaining outcome $k$ upon measurement is $p(k) = \mathrm{tr}(\rho \mathbf{O}_k)$. In systems with $L$ qubits, it is common practice to perform measurements on all qubits simultaneously or in *parallel* (Leibfried et al., 1996; Jullien et al., 2014). According to Born's rule (Born, 1926) in quantum mechanics, this measurement process produces a string of outcomes $\boldsymbol{\sigma} = (\sigma_1, \ldots, \sigma_L)$, where each $\sigma_i \in \{1, \ldots, K\}$, with probability $|\boldsymbol{\Psi}(\sigma_1, \ldots, \sigma_L)|^2$ as defined in Eq. 1.

**Diffusion Models**. Diffusion models are a class of generative models that have gained significant attention in recent years due to their ability to produce high-quality samples from complex distributions (Ho et al., 2020; Song et al., 2021a). These models typically consist of two main processes: a forward process that gradually adds noise to the data, and a reverse process that aims to reconstruct the original data from the noisy representation. In the forward process, a data point sampled from a real-world distribution, denoted as $\mathbf{z}_0 \sim p(\mathbf{z})$, is progressively corrupted by adding Gaussian noise. This process transforms $\mathbf{z}_0$ into a standard Gaussian noise vector $\mathbf{z}_T \sim \mathcal{N}(0, \mathbf{I})$ over a predefined number of steps $T$. For each step $t \in \{1, 2, \ldots, T\}$, the perturbation is governed by the conditional distribution $p(\mathbf{z}_t | \mathbf{z}_{t-1}) = \mathcal{N}(\mathbf{z}_t; \sqrt{1 - \beta_t} \mathbf{z}_{t-1}, \beta_t \mathbf{I})$, where $\beta_t \in (0, 1)$ represents varying variance scales at each time step. This formulation allows for a controlled degradation of the original data, enabling the model to learn to navigate through the noise space effectively. Once the forward process is complete, the reverse denoising process seeks to recover the original data $\mathbf{z}_0$ by sampling from $\mathbf{z}_T$. This is achieved through the training of a diffusion model $f_{\boldsymbol{\theta}}$, which learns the conditional distributions necessary to reverse the noise addition process. The learning objective typically involves minimizing the difference between the predicted and actual distributions, often through techniques such as score matching (Song et al., 2021b) or variational inference (Sohl-Dickstein et al., 2015). Diffusion models have demonstrated remarkable capabilities in generating high-fidelity samples across various domains, including image synthesis (Rombach et al., 2022), audio generation (Kong et al., 2021), and more recently, applications in solving combinatorial optimization problems (Li et al., 2023) and imitation learning (Pearce et al., 2023). Their inherent flexibility and robustness also make them an ideal candidate for capturing the complex distributions associated with quantum systems, thus paving the way for advancements in quantum machine learning applications.

**Classical Shadow for Post-Processing**. A standalone generative model and the samples drawn from its distribution cannot directly reconstruct a quantum state or extract certain features of it. Post-processing is necessary. In this paper, we consider using classical shadow (Huang et al., 2020) based on randomized single-qubit measurements to predict quantum state properties. This measurement procedure is highly efficient on both quantum experiments and classical simulations. Accordingly, we employ Pauli-6 positive operator-valued measure (POVM) to collect discrete measurement records, which are used to train the generative model. Concretely, the Pauli-6 POVM is given as $O_{\text{Pauli-6}} = \left\{ \frac{1}{3} \times |0\rangle\langle 0|, \frac{1}{3} \times |1\rangle\langle 1|, \frac{1}{3} \times |+\rangle\langle +|, \frac{1}{3} \times |-\rangle\langle -|, \frac{1}{3} \times |r\rangle\langle r|, \frac{1}{3} \times |l\rangle\langle l| \right\}$, where $\{|0\rangle, |1\rangle\}, \{|+\rangle, |-\rangle\}, \{|r\rangle, |l\rangle\}$ stand for the eigenbasis of the Pauli operators $Z, X$, and $Y$, respectively. Suppose that the quantum system has $L$ qubits, measuring a single qubit using $O_{\text{Pauli-6}}$ leads to a snapshot $\hat{\rho}_i^{(m)} = 3|s_i^{(m)}\rangle\langle s_i^{(m)}| - \mathbf{I}$ where $i \in \{1, \ldots, L\}$ denotes the $i$-th qubit and $m \in \{1, \ldots, M\}$ denotes the $m$-th measurement. Then the quantum state properties such as correlation and the entanglement entropy can be estimated using the $LM$ snapshots. The details can be found in Sec. 4. After the learning model have been trained, we sample from the model's distribution conditioned on the unseen parameters and collect the samples, i.e., the measurement data. Then classical shadow is used for post-processing to estimate the properties of unknown quantum states and the results are compared to the ground truth to evaluate the model's performance.

## 3 METHODOLOGY

We first define the task of QPE in Sec. 3.1. Key methods and insights underlying QuaDiM are then detailed in Sec. 3.2, where we discuss its mathematical foundations and core innovations.

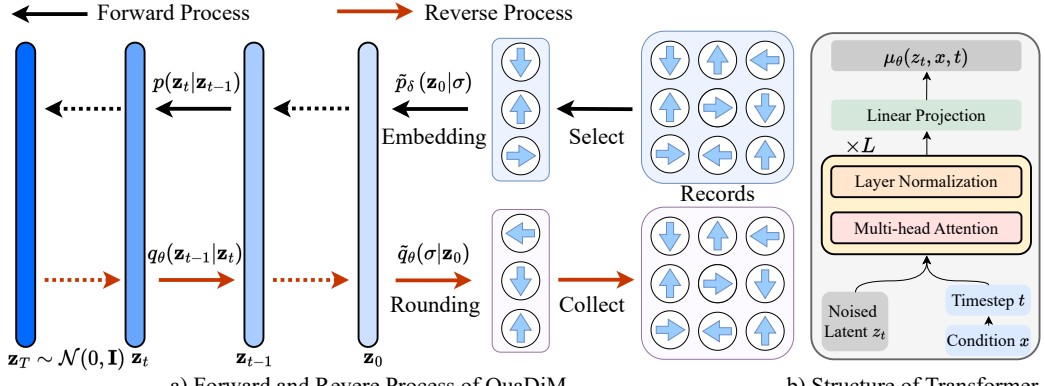

a) Forward and Revere Process of QuaDiM          b) Structure of Transformer

Figure 2: **a)** The diffusion forward process iteratively perturbs the input by adding Gaussian noise, while the reverse process incrementally removes the noise to recover the original distribution. To facilitate the transition between the latent variable $\mathbf{z}_0$ and the observed records $\boldsymbol{\sigma}$, an embedding function (Li et al., 2022) is employed, along with a rounding step (Gong et al., 2023). **b)** The main learnable part is a multi-layer transformer. It models the mean of the posterior distribution $q_{\boldsymbol{\theta}}$.

## 3.1 PROBLEM DEFINITION

The problem is to learn to predict the quantum state properties of ground states of parameterized Hamiltonians. We consider a family of Hamiltonians, where the Hamiltonian $H(\mathbf{x})$ is parameterized by a set of real parameters $\mathbf{x} \in \mathbb{R}^p$. These parameters are physical condition variables determining the evolution dynamics of the quantum system. Different physical conditions could lead to different ground states. The neural network is trained on training data consisting of sampled values of $\{\mathbf{x}_i\}_{i=1}^{N^{tr}}$, each accompanied by the corresponding measurement records $\mathbf{R}_i \in \mathbb{Z}^{M_{in} \times L}$ obtained from Pauli-6 measurements repeatedly operated $M_{in}$ times on the ground state $\rho(\mathbf{x}_i)$ of $H(\mathbf{x}_i)$. This training data could be obtained from either classical simulations or quantum experiments. During the inference phase, we sample $M_{out}$ times from the trained neural network conditioned on a new values of $\mathbf{x}^* \in \{\mathbf{x}_i\}_{i=1}^{N^{te}}$ in the test data and outputs a collection of measurement data $\mathbf{R}^* \in \mathbb{Z}^{M_{out} \times L}$ of the unknown quantum state $\rho(\mathbf{x}^*)$. Ground state properties can then be estimated using post-processing on the $\mathbf{R}^*$ such as the classical shadow.

## 3.2 QUADIM

We propose QuaDiM to extend vanilla diffusion models to learn conditional quantum state representations (as shown in Fig. 2), concerning the architecture and the training objective. For conciseness if not otherwise specified, we will omit the subscripts of the variables in the subsequent text.

### 3.2.1 FORWARD NOISING PROCESS WITH QUADIM

Although diffusion models have achieved success in image generation, applying continuous diffusion models to inherently discrete quantum measurement records remains non-trivial. We follow the Diffusion-LM (Li et al., 2022) in the text generation domain to design an embedding function that projects discrete measurement records into a high-order continuous feature space. Specifically, given a physical condition $\mathbf{x}$ and the corresponding measurement records $\mathbf{R}$, QuaDiM first randomly samples a measurement string $\boldsymbol{\sigma} = (\sigma_1, \ldots, \sigma_L)$ which is a single row of $\mathbf{R}$. To apply a continuous diffusion model to discrete measurements, a token embedding function $\text{EMB}(\sigma_l)$ is introduced to map the single-qubit measurement outcome $\sigma_l$ to a $d$-dimensional vector for each $l \in \{1, \ldots, L\}$. The token embeddings are jointly trained with diffusion model's parameters. We empirically find that these learnable embeddings can achieve better numerical results than the fixed embeddings. The token embedding is accompanied with trainable positional embedding (PE) with the same hidden dimension $d$. Thus the token embedding function is written as $\text{EMB}(\boldsymbol{\sigma}) = [\text{EMB}(\sigma_1) + \text{PE}(1), \ldots, \text{EMB}(\sigma_L) + \text{PE}(L)] \in \mathbb{R}^{L \times d}$. The initial input to the diffusion model in time step $t = 0$ is an additional Markov transition from $\text{EMB}(\boldsymbol{\sigma})$, given as

$$\tilde{p}_{\boldsymbol{\delta}}(\mathbf{z}_0 | \boldsymbol{\sigma}) = \mathcal{N}(\mathbf{z}_0; \text{EMB}(\boldsymbol{\sigma}), \beta_0 I), \tag{2}$$

where $\boldsymbol{\delta}$ is the set of parameters of learnable token embeddings and positional embeddings. The forward noising process gradually corrupts $\mathbf{z}_0$ into a standard Gaussian noise $\mathbf{z}_T \sim \mathcal{N}(0, \mathbf{I})$. For each time step $t \in \{1, \ldots, T\}$, the perturbation is defined as $p(\mathbf{z}_t | \mathbf{z}_{t-1}) = \mathcal{N}(\mathbf{z}_t; \sqrt{1 - \beta_t} \mathbf{z}_{t-1}, \beta_t \mathbf{I})$.

### 3.2.2 REVERSE CONDITIONAL DENOISING PROCESS WITH QUADIM

The proposed conditional denoising employs a classifier-free approach, i.e. we do not need to train an additional classifier to navigate the denoising process. Instead of performing conditional denoising directly on the discrete quantum measurement records, we perform it on the sequence of continuous latent variables $\mathbf{z}_{0:T}$ defined by the QuaDiM. It enables a simple gradient-based algorithm to perform complex, controllable quantum state generation based on diffusion models. Specifically, conditional denoising is equivalent to decoding from the posterior $q_{\boldsymbol{\theta}}(\mathbf{z}_{0:T}|\mathbf{x}) = q(\mathbf{z}_T)\prod_{t=1}^{T} q_{\boldsymbol{\theta}}(\mathbf{z}_{t-1}|\mathbf{z}_t, \mathbf{x})$. For each denoising step, we model it as a Gaussian $q_{\boldsymbol{\theta}}(\mathbf{z}_{t-1}|\mathbf{z}_t, \mathbf{x}) = \mathcal{N}(\mathbf{z}_{t-1}; \mu_{\boldsymbol{\theta}}(\mathbf{z}_t, \mathbf{x}, t), \gamma(t)\mathbf{I})$, where $\mu_{\boldsymbol{\theta}}$ and $\gamma(t)$ are the predicted mean and variance:

$$\mu_{\boldsymbol{\theta}}(\mathbf{z}_t, \mathbf{x}, t) = \frac{\overline{\beta}_{t-1}\sqrt{1-\beta_t}}{\overline{\beta}_t}\mathbf{z}_t + \frac{\beta_t\sqrt{1-\overline{\beta}_{t-1}}}{\overline{\beta}_t}f_{\boldsymbol{\theta}}(\mathbf{z}_t, \mathbf{x}, t), \quad \gamma(t) = \frac{\beta_t\overline{\beta}_{t-1}}{\overline{\beta}_t}, \tag{3}$$

where $\overline{\beta}_t = 1 - \prod_{i=1}^{t}(1-\beta_t)$. The network $\mu_{\boldsymbol{\theta}}(\mathbf{z}_t, \mathbf{x}, t)$ is akin to the multi-layer transformer decoder (Vaswani et al., 2017). The distinction is that an additional time step embedding is specified by the transformer sinusoidal position embedding (PE) (Ho et al., 2020). Besides, to achieve the controllable generation of quantum states conditioned on the physical variable $\mathbf{x}$, we use a feed-forward network (FFN) with one hidden layer to transform $\mathbf{x}$ into a hidden feature with the same dimension $d$. This feature is viewed as a global information added to the token embeddings of measurement records, such that the input to the transformer at time step $t$ is $\mathbf{H}^0 = \mathbf{z}_t + \mathrm{PE}(t) + \mathrm{FFN}(\mathbf{x})$. Suppose that the $(l-1)$-th layer's output $\mathbf{H}^{l-1} \in \mathbb{R}^{L \times d}$ where $d$ is the hidden dimension, we have:

$$\mathbf{H}^l = \mathrm{MULTIHEADATTENTION}(\mathbf{H}^{l-1}) = \llbracket \mathbf{O}_1^l, \ldots, \mathbf{O}_h^l \rrbracket \mathbf{W}_o^l,$$

$$\text{with} \quad \mathbf{O}_j^l = \frac{(\mathbf{H}^{l-1}\mathbf{W}_{q,j}^l)(\mathbf{H}^{l-1}\mathbf{W}_{k,j}^l)^\top}{\sqrt{d}}\mathbf{H}^{l-1}\mathbf{W}_{v,j}^l, \tag{4}$$

where $\llbracket \cdot \rrbracket$ denotes the concatenation operation, $\mathbf{W}_{q,j}^l, \mathbf{W}_{k,j}^l, \mathbf{W}_{v,j}^l \in \mathbb{R}^{d \times d/h}$ and $\mathbf{W}_o^l \in \mathbb{R}^{d \times d}$ are the parameters to be learned. In this paper, all the experimental results of QuaDiM are reported for a transformer configuration consisting of 4 heads, 4 layers, and 128 hidden dimensions. The maximum denoising time steps is set to $T = 2000$.

### 3.2.3 LEARNING OBJECTIVE

The canonical learning objective is to minimize the KL divergence between the joint probability distribution $p(\mathbf{z}_{0:T}|\boldsymbol{\sigma})$ and $q_{\boldsymbol{\theta}}(\mathbf{z}_{0:T}|\mathbf{x})$ corresponding to the forward and reverse processes defined by the diffusion model, respectively. The objective is given as

$$\mathcal{L}_{KL} = \mathbb{E}_{p(\mathbf{z}_{0:T}|\boldsymbol{\sigma})}\log \frac{p(\mathbf{z}_{0:T}|\boldsymbol{\sigma})}{q_{\boldsymbol{\theta}}(\mathbf{z}_{0:T}|\mathbf{x})}, \tag{5}$$

where $p(\mathbf{z}_{0:T}|\boldsymbol{\sigma}) = \tilde{p}_{\boldsymbol{\delta}}(\mathbf{z}_0|\boldsymbol{\sigma})\prod_{t=1}^{T} p(\mathbf{z}_t|\mathbf{z}_{t-1})$ denotes the distribution of the forward process, and $q_{\boldsymbol{\theta}}(\mathbf{z}_{0:T}|\mathbf{x}) = q(\mathbf{z}_T)\prod_{t=1}^{T} q_{\boldsymbol{\theta}}(\mathbf{z}_{t-1}|\mathbf{z}_t, \mathbf{x})$ represents the distribution of the reverse process. Note that in the forward process, $\tilde{p}_{\boldsymbol{\delta}}(\mathbf{z}_0|\boldsymbol{\sigma})$ is the Markov transition (as given in Eq. 2) from the discrete quantum measurements to their continuous embeddings, allowing the token embeddings and model parameters to jointly participate in gradient descent. The objective can be further simplified to

$$\mathcal{L}_{KL} = \mathbb{E}_{p(\mathbf{z}_{0:T}|\boldsymbol{\sigma})}\left[\log \frac{p(\mathbf{z}_T|\mathbf{z}_0)}{q(\mathbf{z}_T)} + \sum_{t=2}^{T}\log \frac{p(\mathbf{z}_{t-1}|\mathbf{z}_0, \mathbf{z}_t)}{q_{\boldsymbol{\theta}}(\mathbf{z}_{t-1}|\mathbf{z}_t)} + \log \frac{\tilde{p}_{\boldsymbol{\delta}}(\mathbf{z}_0|\boldsymbol{\sigma})}{q_{\boldsymbol{\theta}}(\mathbf{z}_0|\mathbf{z}_1)}\right]. \tag{6}$$

The detailed derivation and explanation of each term are given in Appendix A. In the reverse process, the diffusion model iteratively removes Gaussian noise $\mathbf{z}_T$ and outputs the embeddings $\mathbf{z}_0$. To estimate the reconstruction quality from the model's output distribution when training the model, a rounding step (Li et al., 2022; Gong et al., 2023) is added to map the continuous $\mathbf{z}_0$ obtained from the denoising process back to discrete measurement outcomes $\boldsymbol{\sigma}$ of length $L$. Specifically, the step is given as $\tilde{q}_{\boldsymbol{\theta}}(\boldsymbol{\sigma}|\mathbf{z}_0) = \prod_{l=1}^{L} \tilde{q}_{\boldsymbol{\theta}}(\sigma_l|\mathbf{z}_{0,l})$ and $\mathbf{z}_{0,l}$ is the $l$-th component of $\mathbf{z}_0 \in \mathbb{R}^{L \times d}$. Now we introduce the overall objective as the combination of two terms: maximizing the average negative log-likelihood $\mathbb{E}_{p(\mathbf{z}_0|\mathbf{x}, \mathbf{z}_{1:T})}\log \tilde{q}_{\boldsymbol{\theta}}(\boldsymbol{\sigma}|\mathbf{z}_0)$ and minimizing the KL divergence $\mathcal{L}_{KL}$. Thus the objective is as follows, where the detailed derivation is given in Appendix B.

$$\min_{\boldsymbol{\delta}, \boldsymbol{\theta}} \mathcal{L} = \min_{\boldsymbol{\delta}, \boldsymbol{\theta}}\left[\mathcal{L}_{KL} - \mathbb{E}_{p(\mathbf{z}_0|\mathbf{x}, \mathbf{z}_{1:T})}\log \tilde{q}_{\boldsymbol{\theta}}(\boldsymbol{\sigma}|\mathbf{z}_0)\right]$$

$$= \min_{\boldsymbol{\delta}, \boldsymbol{\theta}} \mathbb{E}_{\substack{\mathbf{x} \in \{\mathbf{x}_i\} \\ \boldsymbol{\sigma} \in \mathbf{R}}}\left[\|\mathrm{EMB}(\boldsymbol{\sigma}) - \mu_{\boldsymbol{\theta}}(\mathbf{z}_1, \mathbf{x}, 1)\|^2 + \sum_{t=2}^{T}\|\mathbf{z}_0 - \mu_{\boldsymbol{\theta}}(\mathbf{z}_t, \mathbf{x}, t)\|^2 - \log \tilde{q}_{\boldsymbol{\theta}}(\boldsymbol{\sigma}|\mathbf{z}_0)\right]. \tag{7}$$

Table 1: RMSE of predicting the correlations of all subsystems of size two on the test dataset. The result is averaged over Heisenberg model instances and each pair of adjacent qubits. For CS, $M$ denotes the number of input measurements. While for the neural network-based approaches $M$ denotes the number of sampled measurements $M_{out}$ from trained models. The best results are emphasized in red while the second-best results are distinguished in blue.

| Method | L = 10 | | | | L = 40 | | | | L = 70 | | | | L = 100 | | | |
|---|---|---|---|---|---|---|---|---|---|---|---|---|---|---|---|---|
| | M = 100 | 1000 | 10000 | 20000 | 100 | 1000 | 10000 | 20000 | 100 | 1000 | 10000 | 20000 | 100 | 1000 | 10000 | 20000 |
| CS | 0.1564 | 0.0509 | 0.0156 | 0.0107 | 0.1696 | 0.0538 | 0.0173 | 0.0121 | 0.1771 | 0.0545 | 0.0172 | 0.0121 | 0.1724 | 0.0547 | 0.0172 | 0.0122 |
| RBFK | 0.0796 | | | | 0.0639 | | | | 0.0578 | | | | 0.0493 | | | |
| NTK | 0.0775 | | | | 0.0622 | | | | 0.0565 | | | | 0.0470 | | | |
| RNN | 0.1328 | 0.0502 | 0.0145 | 0.0119 | 0.1795 | 0.0671 | 0.0164 | 0.0118 | 0.2137 | 0.0739 | 0.0240 | 0.0153 | 0.2325 | 0.0806 | 0.0251 | 0.0163 |
| LLM4QPE | 0.1316 | 0.0489 | 0.0136 | 0.0093 | 0.1624 | 0.0513 | 0.0142 | 0.0097 | 0.1814 | 0.0527 | 0.0155 | 0.0116 | 0.1759 | 0.0531 | 0.0152 | 0.0114 |
| Ours | 0.1269 | 0.0432 | 0.0097 | 0.0085 | 0.1582 | 0.0465 | 0.0113 | 0.0091 | 0.1679 | 0.0473 | 0.0117 | 0.0092 | 0.1686 | 0.0478 | 0.0125 | 0.0098 |

## 4 EXPERIMENTS

In this section, we first introduce dataset construction and baselines in Sec. 4.1 and Sec. 4.2. In Sec. 4.3 and Sec. 4.4, we evaluate QuaDiM's performance on QPE tasks compared to state-of-the-art baselines, focusing on predicting correlation and entanglement entropy in various system sizes.

### 4.1 DATASETS

We focus on investigating the ground state of the one-dimensional anti-ferromagnetic Heisenberg model, as it can be efficiently simulated classically at relatively large scale. Its Hamiltonian is parameterized by $\mathbf{x} \in \mathbb{R}^{L-1}$ and can be written as

$$H(\mathbf{x}) = \sum_i x_i \left( X_i X_{i+1} + Y_i Y_{i+1} + Z_i Z_{i+1} \right), \tag{8}$$

where $L$ is the number of qubits and $\mathbf{x}$ is a sequence of real numbers for the coupling strength among qubits. $X, Y, Z$ denotes the Pauli operator and $X_i$ indicates that Pauli $X$ is operated on the $i$-th qubit.

For the sake of simplicity, we first describe how a single sample is generated, with other samples in the dataset being produced in a similar manner. Given the length $L$ of the quantum system, we first randomly sample $L-1$ distinct values of $\mathbf{x}$ to construct the corresponding Hamiltonian $H(\mathbf{x})$. We then obtain the ground state of the $H(\mathbf{x})$ using classical simulation based on tensor networks (Fishman et al., 2022). Thanks to the efficiency of tensor networks, we are able to simulate quantum systems from 10 to 100 qubits on a CPU cluster. Subsequently, Pauli-6 positive operator-valued measure (POVM) is utilized to measure the ground state corresponding to $H(\mathbf{x})$ and the output is a string-like sequence $\boldsymbol{\sigma} = (\sigma_1, \ldots, \sigma_L)$ where $\sigma_l \in \{1, 2, \ldots, K\}$. The same measurement procedure is performed $M_{in}$ times independently on the $M_{in}$ copies of the ground state. As a result, the model input consists of $\mathbf{x}$ along with the corresponding measurement records $\mathbf{R} \in \mathbb{Z}^{M_{in} \times L}$. The above process is repeated $N^{tr}$ and $N^{te}$ times to obtain the training dataset $\{\mathbf{x}_i\}_{i=1}^{N^{tr}}$ with measurement records $\{\mathbf{R}_i\}_{i=1}^{N^{tr}}$ and test dataset $\{\mathbf{x}_j\}_{j=1}^{N^{te}}$ with $\{\mathbf{R}_j\}_{j=1}^{N^{te}}$.

As for the ground truth labels, which represent the ground state properties of the Hamiltonian $H(\mathbf{x})$ such as correlation and entanglement entropy, we utilize exact diagonalization (Weiße & Fehske, 2008) for quantum systems with $L \leq 10$ to obtain the true labels. For quantum systems with $L > 10$, we collect a substantial amount of measurement data by setting $M = 320,000$ and approximate the corresponding labels using classical shadow (Huang et al., 2020).

### 4.2 BASELINES

We consider the classical shadow (CS) (Huang et al., 2020) – a SOTA learning-free protocol for constructing the representation of an unknown quantum state. For learning-based baselines, we consider kernel methods including the Radial Basis Function Kernel (RBFK) and Neural Tangent Kernel (NTK) implemented in Huang et al. (2022). We also explore advanced deep learning approaches, including a Recurrent Neural Network (RNN)-based model (Carrasquilla et al., 2019), and a transformer-based SOTA model LLM4QPE (Tang et al., 2024a).

For all the methods, we set $N^{tr} = 100$ and $N^{te} = 20$, with the number of qubits in the quantum system $L \in \{10, 40, 70, 100\}$. To construct the training set, we perform repeated measurements of $M_{in} = 1000$ for each ground state. To evaluate the performance of the trained QuaDiM and auto-regressive baselines in predicting certain quantum state property w.r.t. different sampling counts, the model is sampled $M_{out}$ times, and the outputs are processed to predict the ground state property

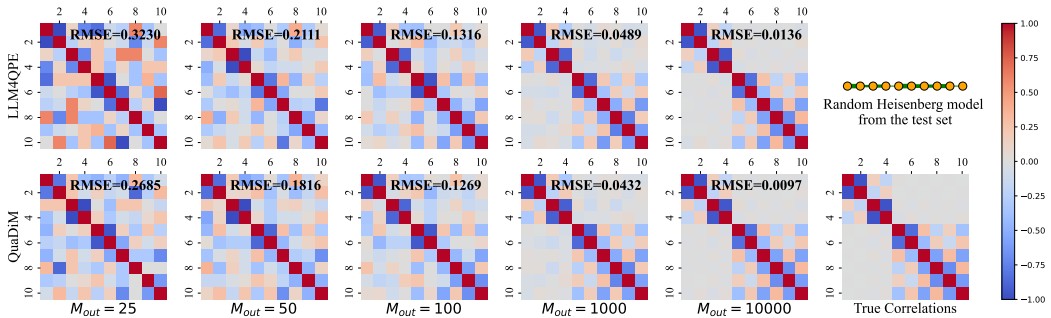

Figure 3: Visualization of predicted correlations $\widehat{C}_{ij}$ for the ground state of the 1D anti-ferromagnetic Heisenberg model of length $L = 10$ with different number of samples $M_{out}$ from the trained models. The chain in the upper right corner represents a Heisenberg model selected from the test set, where the width of the edges indicates the coupling strength $x_i$. The upper part shows the prediction results of LLM4QPE (Tang et al., 2024a), while the lower part presents those of proposed QuaDiM alongside the ground truth. For all the settings, we fix $M_{in} = 1000$ and use classical shadow for post-processing.

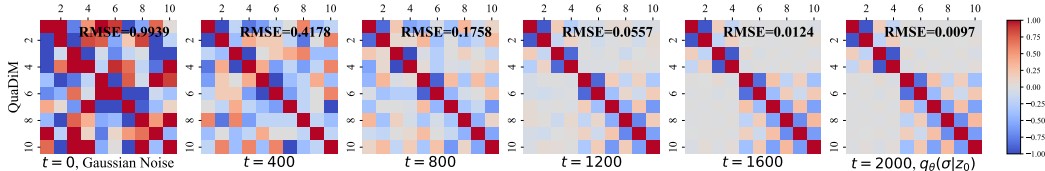

Figure 4: Visualization of predicted correlations of QuaDiM $\widehat{C}_{ij}$ for the ground state of the 1D anti-ferromagnetic Heisenberg model of length $L = 10$. The measurement samples are generated from the different denoising time step $t$. For all the settings, we fix $M_{in} = 1000$ and $M_{out} = 10000$.

of the Hamiltonian in the test dataset. We use the Root Mean Square Error (RMSE) to assess the difference between the predicted property and the true one, with smaller RMSE indicating that the model achieves better predictive accuracy with fewer measurement samples or sampling iterations. To adapt our proposed generative model, we made appropriate adjustments to all auto-regressive baselines including RNN and LLM4QPE, to facilitate generative training and sampling from the trained model, while employing the same classical shadow protocol for post-processing to ensure a fair comparison. For the kernel methods, since generative training is not possible, the model input retains only the physical condition **x**. For specific details about baselines, please refer to Appendix D.

### 4.3 PREDICT THE CORRELATION

The first learning task is to predict the correlations of all subsystems of size two. Denote the density matrix of the quantum state as $\rho$. For two qubits located at different positions $i$ and $j$, the correlation is described as $C_{ij} = \text{tr}(O_{ij}\rho)$ where $O_{ij} = \frac{1}{3}\left(X_iX_j + Y_iY_j + Z_iZ_j\right)$. Given a physical condition **x**, the model along with post-processing is learned to approximate the mapping $f : \mathbf{x} \to C_{ij}$.

The data sampled from the trained model will be fed into the classical shadow for post-processing. Specifically, assuming the number of samples is $M_{out}$, we obtain an output $R_{out} \in \mathbb{Z}^{M_{out} \times L}$, where each element can be converted into a snapshot $\hat{\rho}_i^{(m)} = 3|s_i^{(m)}\rangle\langle s_i^{(m)}| - \mathbf{I}$ where $i \in \{1, \ldots, L\}$ and $m \in \{1, \ldots, M_{out}\}$, as discussed in Sec. 2. Then the predicted correlation between the $i$-th qubit and the $j$-th qubit is calculated by

$$\widehat{C}_{ij} = \frac{1}{3M_{out}} \sum_{o\in\{x,y,z\}} \sum_{m}^{M_{out}} \text{tr}\left[\left(\hat{\rho}_i^{(m)} \otimes \hat{\rho}_j^{(m)}\right)\left(\boldsymbol{\sigma}_i^o \otimes \boldsymbol{\sigma}_j^o\right)\right]. \quad (9)$$

As shown in Tab. 1, QuaDiM achieves the lowest RMSE across all the configurations, showcasing its ability to deliver precise correlation predictions even with a moderate $M_{out}$. An important advantage of QuaDiM is its ability to maintain excellent performance even with fewer $M_{out}$. For instance, at $M_{out} = 100$, QuaDiM's RMSE of surpasses CS and remains competitive with the auto-regressive baselines. This trend persists across larger system sizes as well. QuaDiM demonstrates strong performance at low sample counts, indicating its efficiency in utilizing limited samples from the trained model for accurate quantum state property estimation.

As expected, the RMSE decreases with increasing $M_{out}$, reflecting that all models except for kernel methods improve in predictive accuracy as more measurement samples become available. However, the degree of improvement varies significantly between methods. The CS protocol performs reasonably well for small $M_{out}$ but shows diminishing returns at larger $M_{out}$, particularly for larger system sizes ($L = 40$ and beyond). In contrast, QuaDiM continues to benefit from increased $M_{out}$, which suggests that the model is better at learning and generalizing from larger quantum systems.

Kernel-based methods (RBFK and NTK) show competitive performance for medium system sizes ($L = 40$), but they struggle to match the performance of the generative models (QuaDiM, LLM4QPE and RNN) at higher sample sizes or larger system sizes. This could be due to the inherent limitations of kernel methods in capturing the complex structure of large quantum systems, which generative models like QuaDiM can better model through learned representations.



One of the most critical challenges in quantum state estimation is scalability. While some methods, such as the classical shadow approach, demonstrate robust performance for smaller system sizes ($L = 10$), their performance becomes inferior compared to the neural network-based methods as $L$ increases. This is consistent with the findings in Huang et al. (2022) that neural networks could more easily display non-local correlations, allowing in principle to capture quantum states with higher entanglement for relatively larger quantum systems. Notably, for $L = 100$, the RNN-based baseline yields significantly higher RMSE values compared to others, indicating that the auto-regressive training and generation mechanism makes oversimplified representations of the entangle-


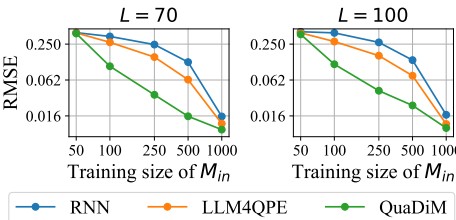

Figure 5: RMSE of predicting the correlations of all subsystems of size two on the test dataset of the auto-regressive baselines and the proposed QuaDiM for $L \in \{70, 100\}$. The models are trained using different of number of input measurements $M_{in}$.



ment structure, limiting the model's ability to fully capture the intricate correlations between qubits. Conversely, QuaDiM shows resilience to scaling, maintaining low RMSE values across all system sizes, with only a slight increase in RMSE as $L$ increases. This scalability suggests that QuaDiM is particularly well-suited for handling large quantum systems, making it a promising approach for real-world quantum computations where the number of qubits can be extensive (for example, IBM's latest commercial quantum computer has more than 1,000 qubits).

**More results about the performance w.r.t. the samples complexity of the trained models.** The experimental results visualized in Fig. 3 show the predicted correlations $\widehat{C}_{ij}$ for the ground state of the 1D anti-ferromagnetic Heisenberg model, with varying sample sizes $M_{out}$. QuaDiM outperforms the auto-regressive baseline LLM4QPE in achieving lower RMSE with fewer samples. As the sample size increases, QuaDiM continues to demonstrate superior performance, achieving an RMSE of 0.0097 at $M_{out} = 10000$, compared to LLM4QPE's 0.0136. We also illustrate QuaDiM's predicted correlations at various denoising time steps $t$ in Fig. 4. As the denoising process progresses, the RMSE steadily decreases. This demonstrates that QuaDiM effectively reconstructs the distribution of quantum states as the denoising process advances, with each time step reducing the noise and improving the fidelity of the sampled quantum states. A more fine-grained visualization of the prediction performance of QuaDiM is depicted in Fig. 6 in Appendix E.

**More results about the performance w.r.t. the number of measurements used for training.** The experimental results shown in Fig. 5 highlight the predictive error of auto-regressive baselines and our proposed QuaDiM when predicting correlations for subsystems of size two within quantum systems of $L \in \{70, 100\}$. The models are trained using different numbers of input measurements $M_{in}$ ranging from 50 to 1000. QuaDiM consistently outperforms the auto-regressive baselines across all training sizes. For both $L \in \{70, 100\}$, it achieves a lower RMSE with fewer input measurements compared to the other methods, showing a advantage in training efficiency. Notably, as $M_{in}$ increases, QuaDiM 's RMSE decreases more sharply than that of the baselines, indicating that it can better approximate the (classical) probability distribution of quantum states using fewer input measurements. In quantum computation, where measurement resources are often limited for both quantum experiments and classical simulation (Gebhart et al., 2023), QuaDiM's ability to perform well with fewer measurements offers a significant advantage over auto-regressive methods.

Table 2: RMSE of predicting the entanglement entropies of all subsystems of size two on the test dataset. The result is averaged over Heisenberg model instances and each pair of adjacent qubits. The best results are emphasized in red while the second-best results are distinguished in blue.

| Method | $L=10$ | | | | $L=40$ | | | | $L=70$ | | | | $L=100$ | | | |
|---|---|---|---|---|---|---|---|---|---|---|---|---|---|---|---|---|
| | $M=100$ | 1000 | 10000 | 20000 | 100 | 1000 | 10000 | 20000 | 100 | 1000 | 10000 | 20000 | 100 | 1000 | 10000 | 20000 |
| CS | 0.5966 | 0.0922 | 0.0204 | 0.0119 | 0.6487 | 0.0927 | 0.0294 | 0.0259 | 0.6421 | 0.0943 | 0.0312 | 0.0298 | 0.6518 | 0.0998 | 0.0357 | 0.0316 |
| RBFK | 0.1268 | | | | 0.1037 | | | | 0.0997 | | | | 0.0752 | | | |
| NTK | 0.1379 | | | | 0.1034 | | | | 0.0983 | | | | 0.0719 | | | |
| RNN | 0.5225 | 0.1164 | 0.0187 | 0.0115 | 0.6132 | 0.1054 | 0.0246 | 0.0212 | 0.7948 | 0.1305 | 0.0514 | 0.0385 | 0.8229 | 0.1476 | 0.0617 | 0.0439 |
| LLM4QPE | 0.4937 | 0.0948 | 0.0176 | 0.0102 | 0.5878 | 0.0896 | 0.0223 | 0.0207 | 0.6258 | 0.0912 | 0.0276 | 0.0251 | 0.6392 | 0.1055 | 0.0312 | 0.0286 |
| Ours | 0.5479 | 0.0867 | 0.0132 | 0.0089 | 0.5629 | 0.0861 | 0.0187 | 0.0145 | 0.5970 | 0.0879 | 0.0245 | 0.0218 | 0.6125 | 0.0928 | 0.0281 | 0.0243 |

## 4.4 PREDICT THE ENTANGLEMENT ENTROPY

Another learning task is to predict the second-order Rényi entanglement entropy of the subsystem $A$, which is formulated as $E_A = -\log(\mathrm{tr}(\rho_A^2))$, where $A$ is a subsystem of the $n$-qubit quantum system. The required number of measurements scales exponentially in the size of the subsystem $A$, but is independent of total system size (Huang et al., 2020). In this paper, we consider learning the entanglement entropy of all the subsystems of size two. This leads to a local quadratic feature for a unknown quantum state $\rho$, which can be also efficiently estimated by the classical shadow given by

$$E_{ij} = -\log(\mathrm{tr}(\rho_{A_{ij}}^2)) = -\log(\mathrm{tr}(\mathbf{S}_{A_{ij}}\rho \otimes \rho)) = -\log(\mathrm{tr}(\mathbf{S}_{A_{ij}}\mathbb{E}[\hat{\rho}_1] \otimes \mathbb{E}[\hat{\rho}_2]))$$
$$\approx \frac{1}{M_{out}(M_{out}-1)} \sum_{m \neq n} \mathrm{tr}(\mathbf{S}_{A_{ij}}\hat{\rho}_1^{(m)} \otimes \hat{\rho}_2^{(n)}) = \widehat{E}_{ij}, \tag{10}$$

where $\hat{\rho}_1$, $\hat{\rho}_2$ are two independent snapshots of the unknown state $\rho$ and $\mathbf{S}_{A_{ij}}$ denotes the local swap operator of two copies of the subsystem comprising the $i$-th and the $j$-th qubits. The $\mathbb{E}[\hat{\rho}]$ represents the averaged value among $M_{out}$ samples from the trained model. Given a specific physical condition $\mathbf{x}$, the model along with post-processing is learned to approximate the mapping $f : \mathbf{x} \to E_{ij}$.

Tab. 2 compares models' performance in predicting the entanglement entropy of subsystems of size two. The key metric is also the RMSE, with lower values indicating better predictive accuracy. The results are presented across different system size $L$ and sample sizes $M_{out}$ from the trained models. CS method performs reasonably well at lower system sizes and sample counts, but it struggles to maintain competitive RMSE values as both $M$ and $L$ increase. Similarly, the kernel-based methods demonstrate some degree of competitiveness but fall behind neural network-based models at larger system sizes and higher sample counts. It can be seen that QuaDiM achieves superior performance across most system sizes and sample counts, consistently achieving lower RMSE values compared to baseline models. This trend continues as the sample size increases, with QuaDiM maintaining its advantage over the other models. The RMSE reduction demonstrates that QuaDiM can accurately predict the entanglement entropy even with a relatively small number of samples.

**Performance across system sizes.** QuaDiM also scales effectively as the system size increases. This indicates that QuaDiM could be a practical choice for real-world quantum computations where the number of qubits can be large. The model's ability to maintain low RMSE values across different system sizes shows its scalability even for the relative large quantum system with 100 qubits.

**Sample efficiency from the trained models.** One of the most important advantages of QuaDiM is its sample efficiency. It consistently achieves lower RMSE values with fewer samples for post-processing. As the number of samples increases, the performance gap widens, with QuaDiM achieving an RMSE of 0.0145 with $M_{out} = 20,000$, while LLM4QPE and CS exhibit higher RMSE. This sample efficiency is critical in post-processing, as large samples leads to prohibitive post-processing costs.

## 5 CONCLUSION

In this paper, we introduced QuaDiM, a novel non-autoregressive model for QPE based on diffusion models. Our results demonstrate it achieves superior performance in predicting quantum properties such as correlation and entanglement entropy with limited measurement data and reduced samples.

**Limitations and Future Work.** While QuaDiM demonstrates promising improvements, there is still room for further investigation regarding its scalability to highly entangled quantum systems and its sampling efficiency. Future work will aim to extend the model to accommodate more complex quantum systems and incorporate advanced methods to enhance the speed of diffusion sampling.

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

## A  DETAILED DERIVATION OF EQ. 6

To derive the expanded form of the KL divergence in the learning objective, we start from the definition of the KL divergence, which is given as

$$\mathcal{L}_{\text{KL}} = \mathbb{E}_{p(\mathbf{z}_{0:T}|\boldsymbol{\sigma})} \left[ \log \frac{p(\mathbf{z}_T|\mathbf{z}_0)}{q(\mathbf{z}_T)} + \sum_{t=2}^{T} \log \frac{p(\mathbf{z}_{t-1}|\mathbf{z}_t, \mathbf{z}_0)}{q_{\boldsymbol{\theta}}(\mathbf{z}_{t-1}|\mathbf{z}_t)} + \log \frac{\tilde{p}_{\boldsymbol{\delta}}(\mathbf{z}_0|\boldsymbol{\sigma})}{q_{\boldsymbol{\theta}}(\mathbf{z}_0|\mathbf{z}_1)} \right]. \tag{11}$$

Note that the KL divergence between $p$ and $q$ is defined as

$$\mathcal{L}_{\text{KL}} = \mathbb{E}_p \left[ \log \frac{p}{q} \right]. \tag{12}$$

Given that $p$ is $p(\mathbf{z}_{0:T}|\boldsymbol{\sigma})$ and $q$ is $q_{\boldsymbol{\theta}}(\mathbf{z}_{0:T}|\mathbf{x})$, the KL divergence should be

$$\mathcal{L}_{\text{KL}} = \mathbb{E}_{p(\mathbf{z}_{0:T}|\boldsymbol{\sigma})} \left[ \log \frac{p(\mathbf{z}_{0:T}|\boldsymbol{\sigma})}{q_{\boldsymbol{\theta}}(\mathbf{z}_{0:T}|\mathbf{x})} \right]. \tag{13}$$

Both $p$ and $q$ are Markov chains and can be factorized. The forward process $p(\mathbf{z}_{0:T}|\boldsymbol{\sigma})$ can be factorized as

$$p(\mathbf{z}_{0:T}|\boldsymbol{\sigma}) = \tilde{p}_{\boldsymbol{\delta}}(\mathbf{z}_0|\boldsymbol{\sigma}) \prod_{t=1}^{T} p(\mathbf{z}_t|\mathbf{z}_{t-1}). \tag{14}$$

And the reverse process $q_{\boldsymbol{\theta}}(\mathbf{z}_{0:T}|\mathbf{x})$ can be also factorized as

$$q_{\boldsymbol{\theta}}(\mathbf{z}_{0:T}|\mathbf{x}) = q(\mathbf{z}_T) \prod_{t=1}^{T} q_{\boldsymbol{\theta}}(\mathbf{z}_{t-1}|\mathbf{z}_t). \tag{15}$$

Here, $\tilde{p}_{\boldsymbol{\delta}}(\mathbf{z}_0|\boldsymbol{\sigma})$ is the initial distribution, $p(\mathbf{z}_t|\mathbf{z}_{t-1})$ is the forward noising process, and $q_{\boldsymbol{\theta}}(\mathbf{z}_{t-1}|\mathbf{z}_t)$ is the reverse denoising process. Substitute the factorized forms into the KL divergence and we can get

$$\mathcal{L}_{\text{KL}} = \mathbb{E}_{p(\mathbf{z}_{0:T}|\boldsymbol{\sigma})} \left[ \log \tilde{p}_{\boldsymbol{\delta}}(\mathbf{z}_0|\boldsymbol{\sigma}) + \sum_{t=1}^{T} \log p(\mathbf{z}_t|\mathbf{z}_{t-1}) - \left( \log q(\mathbf{z}_T) + \sum_{t=1}^{T} \log q_{\boldsymbol{\theta}}(\mathbf{z}_{t-1}|\mathbf{z}_t) \right) \right]$$

$$= \mathbb{E}_{p(\mathbf{z}_{0:T}|\boldsymbol{\sigma})} \left[ (\log \tilde{p}_{\boldsymbol{\delta}}(\mathbf{z}_0|\boldsymbol{\sigma}) - \log q_{\boldsymbol{\theta}}(\mathbf{z}_0|\mathbf{z}_1)) + \sum_{t=1}^{T-1} (\log p(\mathbf{z}_t|\mathbf{z}_{t-1}) - \log q_{\boldsymbol{\theta}}(\mathbf{z}_t|\mathbf{z}_{t+1})) \right.$$

$$\left. + \log p(\mathbf{z}_T|\mathbf{z}_{T-1}) - \log q(\mathbf{z}_T) \right]. \tag{16}$$

We can express $p(\mathbf{z}_t|\mathbf{z}_{t-1})$ in terms of $p(\mathbf{z}_{t-1}|\mathbf{z}_t, \mathbf{z}_0)$ using the properties of the Gaussian distributions in the diffusion process. First, recall that in the forward process, the joint distribution $p(\mathbf{z}_{0:T}|\boldsymbol{\sigma})$ can be rewritten using the posterior distributions $p(\mathbf{z}_{t-1}|\mathbf{z}_t, \mathbf{z}_0)$:

$$p(\mathbf{z}_{0:T}|\boldsymbol{\sigma}) = \tilde{p}_{\boldsymbol{\delta}}(\mathbf{z}_0|\boldsymbol{\sigma}) p(\mathbf{z}_T|\mathbf{z}_0) \prod_{t=2}^{T} p(\mathbf{z}_{t-1}|\mathbf{z}_t, \mathbf{z}_0). \tag{17}$$

This is possible because the diffusion process allows us to compute $p(\mathbf{z}_{t-1}|\mathbf{z}_t, \mathbf{z}_0)$, which is Gaussian. Then we substitute the new expression into the KL divergence:

$$\mathcal{L}_{\text{KL}} = \mathbb{E}_{p(\mathbf{z}_{0:T}|\boldsymbol{\sigma})} \left[ \log \tilde{p}_{\boldsymbol{\delta}}(\mathbf{z}_0|\boldsymbol{\sigma}) - \log q_{\boldsymbol{\theta}}(\mathbf{z}_0|\mathbf{z}_1) + \log p(\mathbf{z}_T|\mathbf{z}_0) - \log q(\mathbf{z}_T) + \sum_{t=2}^{T} (\log p(\mathbf{z}_{t-1}|\mathbf{z}_t, \mathbf{z}_0) - \log q_{\boldsymbol{\theta}}(\mathbf{z}_{t-1}|\mathbf{z}_t)) \right]. \tag{18}$$

Now, the KL divergence becomes:

$$\mathcal{L}_{\text{KL}} = \mathbb{E}_{p(\mathbf{z}_{0:T}|\boldsymbol{\sigma})} \left[ \log \frac{\tilde{p}_{\boldsymbol{\delta}}(\mathbf{z}_0|\boldsymbol{\sigma})}{q_{\boldsymbol{\theta}}(\mathbf{z}_0|\mathbf{z}_1)} + \log \frac{p(\mathbf{z}_T|\mathbf{z}_0)}{q(\mathbf{z}_T)} + \sum_{t=2}^{T} \log \frac{p(\mathbf{z}_{t-1}|\mathbf{z}_t, \mathbf{z}_0)}{q_{\boldsymbol{\theta}}(\mathbf{z}_{t-1}|\mathbf{z}_t)} \right]. \tag{19}$$

The derivation is completed. We can further interpret each term in the expanded KL divergence as follows.

- Initial KL Term ($t = 0$).

$$\mathbb{E}_{p(\mathbf{z}_0|\boldsymbol{\sigma})} \left[ \log \frac{\tilde{p}_{\boldsymbol{\delta}}(\mathbf{z}_0|\boldsymbol{\sigma})}{q_{\boldsymbol{\theta}}(\mathbf{z}_0|\mathbf{z}_1)} \right]. \tag{20}$$

This term measures the discrepancy between the initial embedding distribution $\tilde{p}_{\boldsymbol{\delta}}(\mathbf{z}_0|\boldsymbol{\sigma})$ and the model's initial reconstruction $q_{\boldsymbol{\theta}}(\mathbf{z}_0|\mathbf{z}_1)$.

- Final KL Term ($t = T$).

$$\mathbb{E}_{p(\mathbf{z}_T|\mathbf{z}_0)} \left[ \log \frac{p(\mathbf{z}_T|\mathbf{z}_0)}{q(\mathbf{z}_T)} \right]. \tag{21}$$

This term compares the noise distribution at time $T$ conditioned on $\mathbf{z}_0$ with the standard Gaussian prior $q(\mathbf{z}_T)$.

- Intermediate KL Terms ($2 \leq t \leq T$).

$$\sum_{t=2}^{T} \mathbb{E}_{p(\mathbf{z}_{t-1}, \mathbf{z}_t|\mathbf{z}_0)} \left[ \log \frac{p(\mathbf{z}_{t-1}|\mathbf{z}_t, \mathbf{z}_0)}{q_{\boldsymbol{\theta}}(\mathbf{z}_{t-1}|\mathbf{z}_t)} \right]. \tag{22}$$

These terms measure the discrepancy between the true posterior $p(\mathbf{z}_{t-1}|\mathbf{z}_t, \mathbf{z}_0)$ and the model's predicted distribution $q_{\boldsymbol{\theta}}(\mathbf{z}_{t-1}|\mathbf{z}_t)$.

## B   DETAILED DERIVATION OF EQ. 7

To derive the learning objective of Eq. 7, we first review some notations and definitions of diffusion models and the KL divergence in the main body of the paper as follows.

*Measurement String*: $\boldsymbol{\sigma} = (\sigma_1, \ldots, \sigma_L)$, a sequence of discrete tokens. *Physical Condition*: $\mathbf{x}$, a sequence of physical variables conditioning the ground state of the Hamiltonian $H(\mathbf{x})$. *Token Embeddings*: $\text{EMB}(\boldsymbol{\sigma}) \in \mathbb{R}^{L \times d}$, continuous embeddings of the measurement string. *Initial Embedding Distribution*: $\tilde{p}_{\boldsymbol{\delta}}(\mathbf{z}_0|\boldsymbol{\sigma}) = \mathcal{N}(\mathbf{z}_0; \text{EMB}(\boldsymbol{\sigma}), \beta_0 I)$. *Forward Process*: Adds noise to $\mathbf{z}_0$ to obtain $\mathbf{z}_1, \ldots, \mathbf{z}_T$, with transitions $p(\mathbf{z}_t|\mathbf{z}_{t-1}) = \mathcal{N}(\mathbf{z}_t; \sqrt{1 - \beta_t}\mathbf{z}_{t-1}, \beta_t I)$. *Reverse Process*: Starts from $\mathbf{z}_T \sim \mathcal{N}(0, I)$ and predicts $\mathbf{z}_0$ using $q_{\boldsymbol{\theta}}(\mathbf{z}_{t-1}|\mathbf{z}_t, \mathbf{x}) = \mathcal{N}(\mathbf{z}_{t-1}; \mu_{\boldsymbol{\theta}}(\mathbf{z}_t, \mathbf{x}, t), \gamma(t)I)$.

Then we derive this equation. The KL divergence between the forward process $p(\mathbf{z}_{0:T}|\boldsymbol{\sigma})$ and the reverse process $q_{\boldsymbol{\theta}}(\mathbf{z}_{0:T}|\mathbf{x})$ is

$$\mathcal{L}_{\text{KL}} = \mathbb{E}_{p(\mathbf{z}_{0:T}|\boldsymbol{\sigma})} \left[ \log \frac{p(\mathbf{z}_{0:T}|\boldsymbol{\sigma})}{q_{\boldsymbol{\theta}}(\mathbf{z}_{0:T}|\mathbf{x})} \right]. \tag{23}$$

Expanding the joint distributions and we obtain

$$p(\mathbf{z}_{0:T}|\boldsymbol{\sigma}) = \tilde{p}_{\boldsymbol{\delta}}(\mathbf{z}_0|\boldsymbol{\sigma}) \prod_{t=1}^{T} p(\mathbf{z}_t|\mathbf{z}_{t-1}),$$

$$q_{\boldsymbol{\theta}}(\mathbf{z}_{0:T}|\mathbf{x}) = q(\mathbf{z}_T) \prod_{t=1}^{T} q_{\boldsymbol{\theta}}(\mathbf{z}_{t-1}|\mathbf{z}_t, \mathbf{x}). \tag{24}$$

Substituting back into the KL divergence and we obtain

$$\mathcal{L}_{\text{KL}} = \mathbb{E}_{p(\mathbf{z}_{0:T}|\boldsymbol{\sigma})} \left[ \log \tilde{p}_{\boldsymbol{\delta}}(\mathbf{z}_0|\boldsymbol{\sigma}) - \log q_{\boldsymbol{\theta}}(\mathbf{z}_0|\mathbf{z}_1, \mathbf{x}) + \sum_{t=1}^{T} \left( \log p(\mathbf{z}_t|\mathbf{z}_{t-1}) - \log q_{\boldsymbol{\theta}}(\mathbf{z}_{t-1}|\mathbf{z}_t, \mathbf{x}) \right) - \log q(\mathbf{z}_T) \right]. \tag{25}$$

Then the KL divergence is broken down and we can reorganize the KL divergence into terms corresponding to each timestep. The formulation is given as

$$\mathcal{L}_{\text{KL}} = \mathbb{E}_{p(\mathbf{z}_{0:T}|\boldsymbol{\sigma})} \left[ \text{KL}\left( p(\mathbf{z}_T|\mathbf{z}_0) \| q(\mathbf{z}_T) \right) + \sum_{t=2}^{T} \text{KL}\left( p(\mathbf{z}_{t-1}|\mathbf{z}_t, \mathbf{z}_0) \| q_{\boldsymbol{\theta}}(\mathbf{z}_{t-1}|\mathbf{z}_t, \mathbf{x}) \right) + \text{KL}\left( p(\mathbf{z}_0|\mathbf{z}_1, \boldsymbol{\sigma}) \| q_{\boldsymbol{\theta}}(\mathbf{z}_0|\mathbf{z}_1, \mathbf{x}) \right) \right]. \tag{26}$$

For Gaussian distributions, the KL divergence between $p = \mathcal{N}(\mu_p, \Sigma_p)$ and $q = \mathcal{N}(\mu_q, \Sigma_q)$ is

$$\text{KL}(p\|q) = \frac{1}{2} \left[ \text{tr}(\Sigma_q^{-1}\Sigma_p) + (\mu_q - \mu_p)^\top \Sigma_q^{-1}(\mu_q - \mu_p) - k + \ln\left( \frac{\det \Sigma_q}{\det \Sigma_p} \right) \right], \tag{27}$$

where $k$ is the dimensionality. Given that variances are fixed and known, and constants can be ignored during optimization, the KL divergence reduces to

$$\text{KL}\left(p\|q\right) \propto \|\mu_q - \mu_p\|^2. \tag{28}$$

Now we can specify the KL divergence at each time step. For $t = 1$, the KL divergence between $p(\mathbf{z}_0|\mathbf{z}_1, \boldsymbol{\sigma})$ and $q_{\boldsymbol{\theta}}(\mathbf{z}_0|\mathbf{z}_1, \mathbf{x})$ is:

$$\text{KL}\left(p(\mathbf{z}_0|\mathbf{z}_1, \boldsymbol{\sigma})\|q_{\boldsymbol{\theta}}(\mathbf{z}_0|\mathbf{z}_1, \mathbf{x})\right) \propto \|\mu_{\boldsymbol{\theta}}(\mathbf{z}_1, \mathbf{x}, 1) - \tilde{\mu}_1(\mathbf{z}_1, \text{EMB}(\boldsymbol{\sigma}))\|^2, \tag{29}$$

where $\tilde{\mu}_1(\mathbf{z}_1, \text{EMB}(\boldsymbol{\sigma}))$ is the true posterior mean from the forward process. When the time step $t \geq 2$, for each $t$ we have

$$\text{KL}\left(p(\mathbf{z}_{t-1}|\mathbf{z}_t, \mathbf{z}_0)\|q_{\boldsymbol{\theta}}(\mathbf{z}_{t-1}|\mathbf{z}_t, \mathbf{x})\right) \propto \|\mu_{\boldsymbol{\theta}}(\mathbf{z}_t, \mathbf{x}, t) - \tilde{\mu}_t(\mathbf{z}_t, \mathbf{z}_0)\|^2. \tag{30}$$

In the forward process, the posterior mean $\tilde{\mu}_t$ is a linear combination of $\mathbf{z}_t$ and $\mathbf{z}_0$ given by

$$\tilde{\mu}_t(\mathbf{z}_t, \mathbf{z}_0) = \frac{\sqrt{\overline{\alpha}_{t-1}}\beta_t}{1 - \overline{\alpha}_t}\mathbf{z}_0 + \frac{\sqrt{\alpha_t}(1 - \overline{\alpha}_{t-1})}{1 - \overline{\alpha}_t}\mathbf{z}_t. \tag{31}$$

When $\beta_t$ is small, we can approximate $\tilde{\mu}_t(\mathbf{z}_t, \mathbf{z}_0) \approx \mathbf{z}_0$. Substituting the approximations back into the KL divergence terms and we obtain the term for $t = 1$ is

$$\|\mu_{\boldsymbol{\theta}}(\mathbf{z}_1, \mathbf{x}, 1) - \text{EMB}(\boldsymbol{\sigma})\|^2. \tag{32}$$

The term for $t \geq 2$ is

$$\|\mu_{\boldsymbol{\theta}}(\mathbf{z}_t, \mathbf{x}, t) - \mathbf{z}_0\|^2. \tag{33}$$

The rounding step involves mapping $\mathbf{z}_0$ back to the discrete measurements $\boldsymbol{\sigma}$ using $\tilde{q}_{\boldsymbol{\theta}}(\boldsymbol{\sigma}|\mathbf{z}_0)$. The negative log-likelihood of this step is

$$-\mathbb{E}_{p(\mathbf{z}_0|\mathbf{x}, \mathbf{z}_{1:T})} \log \tilde{q}_{\boldsymbol{\theta}}(\boldsymbol{\sigma}|\mathbf{z}_0). \tag{34}$$

Then we can combine all terms, the learning objective becomes to minimize

$$\mathcal{L} = \mathbb{E}_{\mathbf{x}, \boldsymbol{\sigma}}\left[\|\text{EMB}(\boldsymbol{\sigma}) - \mu_{\boldsymbol{\theta}}(\mathbf{z}_1, \mathbf{x}, 1)\|^2 + \sum_{t=2}^{T} \|\mathbf{z}_0 - \mu_{\boldsymbol{\theta}}(\mathbf{z}_t, \mathbf{x}, t)\|^2 - \log \tilde{q}_{\boldsymbol{\theta}}(\boldsymbol{\sigma}|\mathbf{z}_0)\right]. \tag{35}$$

This objective encourages the model to minimize the difference between the predicted means $\mu_{\boldsymbol{\theta}}(\mathbf{z}_t, \mathbf{x}, t)$ and the true values ($\text{EMB}(\boldsymbol{\sigma})$ at $t = 1$ and $\mathbf{z}_0$ for $t \geq 2$), and encourages to maximize the likelihood of reconstructing the measurement string $\boldsymbol{\sigma}$ from $\mathbf{z}_0$.

## C  MORE BASICS OF QUANTUM STATES AND QUANTUM MEASUREMENTS

In this section, we provide more details about the quantum states and quantum measurements and their relationship with classical joint distribution. For a comprehensive discussion, we refer the readers who are interested in quantum computing and quantum information to the Section 2.1 of the book Nielsen & Chuang (2010).

A single qubit – the smallest unit of quantum computing – is mathematically represented as a vector $|\psi\rangle = \alpha|0\rangle + \beta|1\rangle$ parameterized by two complex numbers satisfying $|\alpha|^2 + |\beta|^2 = 1$. Operations on a qubit must preserve this norm, and thus are described by $2 \times 2$ unitary matrices. Of these, some of the most important are the Pauli operators; it is useful to list them here:

$$X \equiv \begin{bmatrix} 0 & 1 \\ 1 & 0 \end{bmatrix}, \quad Y \equiv \begin{bmatrix} 0 & -i \\ i & 0 \end{bmatrix}, \quad Z \equiv \begin{bmatrix} 1 & 0 \\ 0 & -1 \end{bmatrix}. \tag{36}$$

One could do some linear algebras and check that $|0\rangle = \begin{bmatrix} 1 \\ 0 \end{bmatrix}$ and $|1\rangle = \begin{bmatrix} 0 \\ 1 \end{bmatrix}$ are the eigenvectors of $Z$, $|+\rangle = \frac{1}{\sqrt{2}}\begin{bmatrix} 1 \\ 1 \end{bmatrix}$ and $|-\rangle = \frac{1}{\sqrt{2}}\begin{bmatrix} 1 \\ -1 \end{bmatrix}$ are the eigenvectors of $X$, $|\boldsymbol{i}_+\rangle = \frac{1}{\sqrt{2}}\begin{bmatrix} 1 \\ i \end{bmatrix}$ and $|\boldsymbol{i}_-\rangle = \frac{1}{\sqrt{2}}\begin{bmatrix} 1 \\ -i \end{bmatrix}$ are the eigenvectors of $Y$. The same qubit can be decomposed into different orthonormal basis. For example,

$$\begin{aligned}
|\psi\rangle &= \alpha|0\rangle + \beta|1\rangle \\
&= \frac{1}{\sqrt{2}}(\alpha + \beta)|+\rangle + \frac{1}{\sqrt{2}}(\alpha - \beta)|-\rangle \\
&= \frac{1}{\sqrt{2}}(\alpha - \beta\boldsymbol{i})|\boldsymbol{i}_+\rangle + \frac{1}{\sqrt{2}}(\alpha + \beta\boldsymbol{i})|\boldsymbol{i}_-\rangle.
\end{aligned} \tag{37}$$

Positive-operator valued measurement (POVM) is the testing or manipulation of a physical system to yield a numerical result. POVM is described by a set of measurement operators $\{\mathbf{\Pi}_k\}_{k=0}^{K-1}$ satisfying $\sum_k \mathbf{\Pi}_k = \mathbf{I}$ and each $\mathbf{\Pi}_k$ is positive semi-definite, where $K$ is the total number of measurement operators. In this paper, we consider the Pauli-6 POVM (also named as randomized single-qubit Pauli measurements in some literature) such that the measurement operators are $\{\frac{1}{3}|0\rangle\langle 0|, \frac{1}{3}|1\rangle\langle 1|, \frac{1}{3}|+\rangle\langle +|, \frac{1}{3}|-\rangle\langle -|, \frac{1}{3}|\boldsymbol{i}_+\rangle\langle \boldsymbol{i}_+|, \frac{1}{3}|\boldsymbol{i}_-\rangle\langle \boldsymbol{i}_-|\}$. It is easy to check that these operators satisfy the POVM definition and $K = 6$. The reason for choosing the Pauli-6 POVM is that this measurement protocol is easy to be implemented on current quantum devices (NISQ devices) and is informative-completed (IC).

Measuring a qubit leads to collapse of the qubit and produces an outcome $k$ with the probability $p(k)$ satisfying the Born rule, which states that $p(k) = \operatorname{tr}(\rho\mathbf{\Pi}_k)$, where $\rho = |\psi\rangle\langle\psi|$ and $\langle\psi|$ is the transpose conjugate of $|\psi\rangle$. We may consider a system of $L$ qubits. It can be described by the wave function:

$$|\mathbf{\Phi}\rangle = \sum_{\sigma_1=1}^{M} \cdots \sum_{\sigma_L=1}^{M} \mathbf{\Psi}(\sigma_1,\ldots,\sigma_L)|\sigma_1,\ldots,\sigma_L\rangle,$$

where $\mathbf{\Psi} : \mathbb{Z}^L \to \mathbb{C}$ maps a fixed configuration $\sigma = (\sigma_1,\ldots,\sigma_L)$ of $L$ qubits to a complex number which is the amplitude satisfying $\sum_{\sigma_1=1}^{K} \cdots \sum_{\sigma_L=1}^{K} |\mathbf{\Psi}(\sigma_1,\ldots,\sigma_L)|^2 = 1$, and $\sigma_i \in \{1,\ldots,K\}$ is one of the $K$ possible outcomes by performing quantum measurement on the $i$-th qubit. It is formulated in a complex Hilbert space where the vector representation of the quantum state $|\mathbf{\Phi}\rangle \in \mathbb{C}^{K^L}$ and its density matrix $|\mathbf{\Phi}\rangle\langle\mathbf{\Phi}| \in \mathbb{C}^{K^L \times K^L}$, which becomes astronomical for large $L$.

Performing quantum measurement independently on $L$ qubits is easy to be implemented. The most common strategy is to combine $L$ single-qubit measurement operators to $\mathbf{\Pi}_{k,1} \otimes \cdots \otimes \mathbf{\Pi}_{k,L}$ where $\otimes$ is the Kronecker product. Such measurement procedure outputs a measurement string $\sigma = (\sigma_1,\ldots,\sigma_L)$ where $\sigma_i \in \{1,\ldots,K\}$ with probability $|\mathbf{\Psi}(\sigma_1,\ldots,\sigma_L)|^2$. Define $p(\sigma_1,\ldots,\sigma_L) = |\mathbf{\Psi}(\sigma_1,\ldots,\sigma_L)|^2$. We can reformulate the wave function of quantum states to a classical joint distribution. It is a valid and legal joint distribution since $\sum_{\sigma_1} \cdots \sum_{\sigma_L} p(\sigma_1,\ldots,\sigma_L) = \sum_{\sigma_1} \cdots \sum_{\sigma_L} |\mathbf{\Psi}(\sigma_1,\ldots,\sigma_L)|^2 = 1$ and $p(\sigma_1,\ldots,\sigma_L) \geq 0$.

## D  DETAILS FOR BASELINE MODELS

We consider the classical shadow (CS) method (Huang et al., 2020), a state-of-the-art, learning-free approach for efficiently constructing representations of unknown quantum states. For learning-based comparisons, we include kernel methods including the Radial Basis Function Kernel (RBFK) and Neural Tangent Kernel (NTK) following the implementation in Huang et al. (2022). Additionally, we evaluate advanced neural network-based models, including a Recurrent Neural Network (RNN)-based architecture (Carrasquilla et al., 2019) and the transformer-based state-of-the-art model LLM4QPE (Tang et al., 2024a). The details about their configurations are as follows.

**Classical Shadow (Huang et al., 2020).** Classical Shadow (CS) is a learning-free protocol used to efficiently predict many properties of quantum states with a logarithmic number of measurements. By using randomized measurements and building a memory-efficient representation of the quantum state, it enables the estimation of various properties with high accuracy.

**Kernel Methods (Huang et al., 2022).** We utilize Radial Basis Function Kernel (RBFK) and Neural Tangent Kernel (NTK) to learn the feature map from physical conditions $\mathbf{x}$ to certain properties of the ground state of the Hamiltonian $H(\mathbf{x})$. A grid search is performed to identify the optimal regularization strength, with candidate values uniformly distributed on a logarithmic scale from 0.001 to 100. We employ a 5-fold cross-validation strategy on the training dataset and present the predictive performance of the model that achieves the highest accuracy on the test dataset.

**RNN (Carrasquilla et al., 2019).** Recurrent Neural Networks (RNNs) are employed to reconstruct quantum states from measurement data, leveraging their ability to capture temporal or sequential dependencies. In our implementation, the RNN takes a sequence of measurement outcomes as input and learns to predict the underlying quantum state properties by modeling correlations between successive measurements. The model architecture includes a hidden layer with 128 units and is trained using the Adam optimizer. To ensure fair comparison, we perform early stopping based on validation loss and use the test dataset for final evaluation of predictive performance.

Table 3: RMSE of predicting the correlations of all subsystems of size two on the test dataset. The result is averaged over Heisenberg model instances and each pair of adjacent qubits. For CS, $M$ denotes the number of input measurements. While for the neural network-based approaches $M$ denotes the number of sampled measurements $M_{out}$ from trained models. The standard deviations are distinguished in gray.

| Method | L = 10 | | | | L = 40 | | | | L = 70 | | | | L = 100 | | | |
|---|---|---|---|---|---|---|---|---|---|---|---|---|---|---|---|---|
| | M = 100 | 1000 | 10000 | 20000 | 100 | 1000 | 10000 | 20000 | 100 | 1000 | 10000 | 20000 | 100 | 1000 | 10000 | 20000 |
| CS | 0.1564 | 0.0509 | 0.0156 | 0.0107 | 0.1696 | 0.0538 | 0.0173 | 0.0121 | 0.1771 | 0.0545 | 0.0172 | 0.0121 | 0.1724 | 0.0547 | 0.0172 | 0.0122 |
| RBFK | | 0.0796 | | | | 0.0639 | | | | 0.0578 | | | | 0.0493 | | |
| NTK | | 0.0775 | | | | 0.0622 | | | | 0.0565 | | | | 0.0470 | | |
| RNN | 0.1328 | 0.0502 | 0.0145 | 0.0119 | 0.1795 | 0.0671 | 0.0164 | 0.0118 | 0.2137 | 0.0739 | 0.0240 | 0.0153 | 0.2325 | 0.0806 | 0.0251 | 0.0163 |
| | 0.0342 | 0.0098 | 0.0024 | 0.0014 | 0.0388 | 0.0132 | 0.0035 | 0.0016 | 0.0453 | 0.0149 | 0.0037 | 0.0025 | 0.0419 | 0.0094 | 0.0043 | 0.0039 |
| LLM4QPE | 0.1316 | 0.0489 | 0.0136 | 0.0093 | 0.1624 | 0.0513 | 0.0142 | 0.0097 | 0.1814 | 0.0527 | 0.0155 | 0.0116 | 0.1759 | 0.0531 | 0.0152 | 0.0114 |
| | 0.0379 | 0.0140 | 0.0032 | 0.0015 | 0.0392 | 0.0128 | 0.0032 | 0.0017 | 0.0462 | 0.0157 | 0.0038 | 0.0021 | 0.0493 | 0.0082 | 0.0056 | 0.0028 |
| Ours | 0.1269 | 0.0432 | 0.0097 | 0.0085 | 0.1582 | 0.0465 | 0.0113 | 0.0091 | 0.1679 | 0.0473 | 0.0117 | 0.0092 | 0.1686 | 0.0478 | 0.0125 | 0.0098 |
| | 0.0365 | 0.0097 | 0.0022 | 0.0008 | 0.0416 | 0.0115 | 0.0037 | 0.0004 | 0.0479 | 0.0138 | 0.0029 | 0.0011 | 0.0512 | 0.0083 | 0.0036 | 0.0009 |

Table 4: RMSE of predicting the entanglement entropies of all subsystems of size two on the test dataset. The result is averaged over Heisenberg model instances and each pair of adjacent qubits. For CS, $M$ denotes the number of input measurements. While for the neural network-based approaches $M$ denotes the number of sampled measurements $M_{out}$ from trained models. The standard deviations are distinguished in gray.

| Method | L = 10 | | | | L = 40 | | | | L = 70 | | | | L = 100 | | | |
|---|---|---|---|---|---|---|---|---|---|---|---|---|---|---|---|---|
| | M = 100 | 1000 | 10000 | 20000 | 100 | 1000 | 10000 | 20000 | 100 | 1000 | 10000 | 20000 | 100 | 1000 | 10000 | 20000 |
| CS | 0.5966 | 0.0922 | 0.0204 | 0.0119 | 0.6487 | 0.0927 | 0.0294 | 0.0259 | 0.6421 | 0.0943 | 0.0312 | 0.0298 | 0.6518 | 0.0998 | 0.0357 | 0.0316 |
| RBFK | | 0.1268 | | | | 0.1037 | | | | 0.0997 | | | | 0.0752 | | |
| NTK | | 0.1379 | | | | 0.1034 | | | | 0.0983 | | | | 0.0719 | | |
| RNN | 0.5225 | 0.1164 | 0.0187 | 0.0115 | 0.6132 | 0.1054 | 0.0246 | 0.0212 | 0.7948 | 0.1305 | 0.0514 | 0.0385 | 0.8229 | 0.1476 | 0.0617 | 0.0439 |
| | 0.0833 | 0.0263 | 0.0022 | 0.0014 | 0.0945 | 0.0314 | 0.0026 | 0.0015 | 0.0997 | 0.0302 | 0.0048 | 0.0026 | 0.0957 | 0.0269 | 0.0054 | 0.0032 |
| LLM4QPE | 0.4937 | 0.0948 | 0.0176 | 0.0102 | 0.5878 | 0.0896 | 0.0223 | 0.0207 | 0.6258 | 0.0912 | 0.0276 | 0.0251 | 0.6392 | 0.1055 | 0.0312 | 0.0286 |
| | 0.0932 | 0.0168 | 0.0023 | 0.0012 | 0.0845 | 0.0209 | 0.0016 | 0.0011 | 0.0940 | 0.0235 | 0.0016 | 0.0018 | 0.0979 | 0.0212 | 0.0020 | 0.0018 |
| Ours | 0.5479 | 0.0867 | 0.0132 | 0.0089 | 0.5629 | 0.0861 | 0.0187 | 0.0145 | 0.5970 | 0.0879 | 0.0245 | 0.0218 | 0.6125 | 0.0928 | 0.0281 | 0.0243 |
| | 0.0952 | 0.0079 | 0.0019 | 0.0011 | 0.0822 | 0.0196 | 0.0021 | 0.0009 | 0.0835 | 0.0194 | 0.0014 | 0.0012 | 0.0857 | 0.0163 | 0.0018 | 0.0014 |

**LLM4QPE (Tang et al., 2024a).** It presents a transformer-based approach for predicting various properties of quantum systems through a pre-training procedure that maximizes a likelihood function based on discrete measurement outcomes. This is also a generative process similar to the conditional quantum state modeling discussed in Wang et al. (2022). The trained model can generate measurement samples conditioned on the physical parameters which are unseen from the training dataset. Afterwards the quantum properties can be analyzed by using post-processing such as classical shadow. For a fair comparison, we set the model with 4 heads, 4 layers, and a hidden dimension of 128.

# E    VISUALIZED SUPPLEMENT OF THE COMPARISON BETWEEN QUADIM AND AUTOREGRESSIVE BASELINE

We include an additional plot in Fig. 6, where we present a more fine-grained visualization of the prediction performance of QuaDiM. In this figure, each point represents the absolute error between the predicted correlation and the ground truth for all pairs of qubits across different $M_{out}$.

# F    ADDITIONAL EXPERIMENT RESULTS

## F.1    IMPACT OF DIFFERENT PRE-DEFINED ORDER

To further compare the our model and the autoregressive baselines, we define an alternative ordering of qubits (from right to left or from the largest to the smallest index, as opposed to the left-to-right ordering used throughout the paper) and re-train the autoregressive baselines. The results of RMSE for predicting the correlations of all subsystems of size two on the test dataset, evaluated in the predefined order from right to left, are presented in Tab. 5. For comparison, we also include experimental results already reported in the main text of the paper.

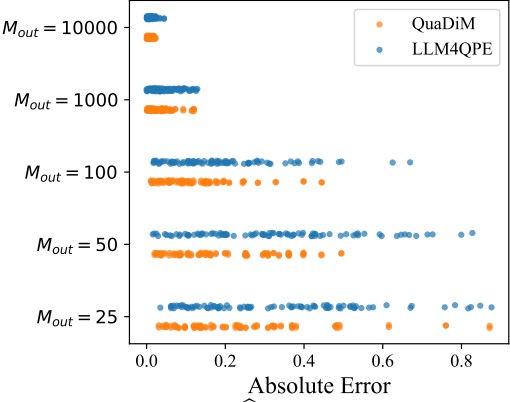

Figure 6: Visualization of predicted correlations $\widehat{C}_{ij}$ for the ground state of the 1D anti-ferromagnetic Heisenberg model of length $L = 10$ with different number of samples $M_{out}$ from the trained models. We fix $M_{in} = 1000$ and use classical shadow for post-processing. Each point represents the absolute error (lower is better) between the predicted correlation and the ground truth for all pairs of qubits across different $M_{out}$.

Table 5: RMSE of predicting the correlations of all subsystems of size two on the test dataset under different sequential order. $M$ denotes the number of sampled measurements $M_{out}$ from trained models. Notation $\leftarrow$ denotes from right to left and $\rightarrow$ denotes from left to right.

| Method | $L = 70$ | | | | $L = 100$ | | | |
|---|---|---|---|---|---|---|---|---|
| | $M = 100$ | $M = 1000$ | $M = 10000$ | $M = 20000$ | $M = 100$ | $M = 1000$ | $M = 10000$ | $M = 20000$ |
| RNN($\leftarrow$) | 0.2197 | 0.0721 | 0.0216 | 0.0165 | 0.2276 | 0.0763 | 0.0264 | 0.0159 |
| RNN($\rightarrow$) | 0.2137 | 0.0739 | 0.0240 | 0.0153 | 0.2325 | 0.0806 | 0.0251 | 0.0163 |
| LLM4QPE($\leftarrow$) | 0.1865 | 0.0538 | 0.0157 | 0.0108 | 0.1773 | 0.0542 | 0.0149 | 0.0122 |
| LLM4QPE($\rightarrow$) | 0.1814 | 0.0527 | 0.0155 | 0.0116 | 0.1759 | 0.0531 | 0.0152 | 0.0114 |
| QuaDiM | 0.1679 | 0.0473 | 0.0117 | 0.0092 | 0.1686 | 0.0478 | 0.0125 | 0.0098 |

### F.2 NUMERICAL RESULTS OF LEARNING LONG-RANGE XY MODEL

To further generalize the effectiveness of our proposed model, we supplement our experiments with results from a more physically general and classically challenging system: the long-range XY model in the presence of a transverse field, whose Hamiltonian is given by

$$H_{XY} = \sum_{i<j} J_{ij}(X_i X_j + Y_i Y_j) + \sum_j Z_j \tag{38}$$

where $J_{ij} = J_0/|i - j|^a$ with $a \in (1, 2)$. This quantum model inherits the long-range interactions between every two quantum sites, leading to a complex dynamics which is hard to be simulated by classical computers. We restrict the system size $L = 10$ due to memory limitations. The ground states of quantum systems with different physical conditions $J_{ij}$ are calculated by eigenvalue decomposition. Follow Xiao et al. (2022), we random sample a series of $J_{ij}$ and conduct classical simulations to collect the data. The experimental results for $L = 10$, with $M_{in} = 1000$ and $M_{out} \in \{1000, 10000\}$, are presented in Tab. 6. It can be observed that QuaDiM consistently outperforms baselines.

Table 6: RMSE of predicting the correlations of ground states of the long-range XY model.

| Method | $M = 1000$ | $M = 10000$ |
|---|---|---|
| CS | 0.2575 | 0.0517 |
| RBFK | 0.1158 | |
| NTK | 0.1039 | |
| RNN | 0.2234 | 0.0502 |
| LLM4QPE | 0.2139 | 0.0482 |
| QuaDiM | 0.1986 | 0.0367 |

### F.3 RESULTS OF VALIDATION ON OUT-OF-DISTRIBUTION DATASET

Here, we consider evaluating the QuaDiM with out-of-distribution (OOD) dataset, which means the dataset used for training and the dataset used for validation come from different distributions. We

divide the sampled physical parameters $\mathbf{x}$ into two segments: the training set is limited to [0, 1.5], while the test set exclusively spans [1.5, 2]. In alignment with setting of the paper, we set $N^{tr} = 100$ and $N^{te} = 20$. We report the RMSE of predicted correlations for both the SOTA baseline LLM4QPE, and our model QuaDiM under both OOD and non-OOD (both training and fine-tuning are sampled from the same distribution [0, 2]) conditions, with $M_{in} = 1000$ and $M_{out} = 10000$. The results are presented in the Tab. 7.

Table 7: RMSE of predicting the correlations of all subsystems of size two on the test dataset under the out-of-distribution (OOD) condition.

| Method | $L = 70$ | | $L = 100$ | |
|---|---|---|---|---|
| | no OOD | OOD | no OOD | OOD |
| LLM4QPE | 0.0155 | 0.0526 | 0.0152 | 0.0598 |
| QuaDiM | 0.0117 | 0.0417 | 0.0125 | 0.0465 |

### F.4 MODEL SENSITIVITY TO THE DENOISING STEPS AND HIDDEN DIMENSION

In this section, we study the relationship between the hyper-parameters including the number of denoising steps $T_f$ and the hidden dimension of QuaDiM and the prediction performance. We fix the number of diffusion steps during training for QuaDiM while shrinking the inference steps $T_f$ using the approach introduced in DDIM (Song et al., 2021a). We evaluate the model's performance under different inference step settings and compare it with both a learning-free baseline, i.e. CS, and a learning-based SOTA model LLM4QPE. We report results for the task of predicting correlations with $L = 100$, $M_{in} = 1000$, and $M_{out} = 1000$.

As shown in the Tab. 8, when reducing inference to $T_f = 500$ diffusion steps on a single GPU (2080Ti), QuaDiM achieves a lower RMSE score compared to the CS while demonstrating an inference speed comparable to LLM4QPE.

To further investigate, we evaluate the model's performance on the task of predicting correlations under a fixed dataset configuration $L = 10$, $M_{in} = 1000$, $M_{out} = 1000$ with different $d$ values from $\{64, 128, 256, 512\}$, and the resulting RMSE scores (lower is better) are: 0.0518, 0.0432, 0.0449, 0.0457, respectively. As the results show, setting $d = 128$ achieves the best performance.

Table 8: RMSE of predicting correlations and the sampling speed with $L = 100$, $M_{in} = 1000$, and $M_{out} = 1000$ under different denoising steps $T_f$.

| Method | RMSE | Generated samples per sec. |
|---|---|---|
| CS | 0.0547 | - |
| LLM4QPE | 0.0531 | 14.6 |
| QuaDiM ($T_f = 2000$) | 0.0478 | 5.7 |
| QuaDiM ($T_f = 1000$) | 0.0537 | 8.1 |
| QuaDiM ($T_f = 500$) | 0.0541 | 12.7 |
| QuaDiM ($T_f = 100$) | 0.0882 | 37.4 |

### F.5 MODEL SENSITIVITY TO THE POSITIONAL EMBEDDINGS

Positional embeddings in our context are employed to capture the structural information among qubits. Additionally, we further investigate the model's performance in predicting correlations when using relative positional encoding (following the implementation in Shaw et al. (2018)) and no positional encoding at all. We report results for $L = 100$ and $M_{in} = 1000$. The experimental results are provided in Tab. 9. The results show that absolute and relative positional embeddings yield comparable performance, but removing positional information altogether significantly degrades the model's ability to predict quantum correlations. This underscores the embeddings' role in preserving spatial relationships among qubits.

### F.6 EXPERIMENTS ON TETRAHEDRAL POVM

In the main text of the paper, Pauli-6 POVM is used for data collection because this measurement protocol is easy to implement on current quantum devices and is informationally complete (IC). This

Table 9: QuaDiM's RMSE of predicting correlations using different positional embedding (PE) techniques.

|  | Absolute PE | Relative PE | No PE |
|---|---|---|---|
| $M_{out} = 100$ | 0.1686 | 0.1681 | 0.4527 |
| $M_{out} = 1000$ | 0.0478 | 0.0482 | 0.3269 |
| $M_{out} = 10000$ | 0.0125 | 0.0139 | 0.2895 |
| $M_{out} = 20000$ | 0.0098 | 0.0110 | 0.2148 |

means that all the information of the quantum state can be recovered classically with a sufficiently large number of IC-POVM measurements. In other words, given the probability of each measurement outcome of IC-POVM, the quantum state can be uniquely determined.

To further validate our method, here we consider another type of IC-POVM: the tetrahedral POVM, to collect measurement data. The corresponding measurement operators are $\{\frac{1}{4}(\mathbf{I}+\mathbf{s}^{(a)}\cdot\mathbf{P})\}_{a\in\{0,1,2,3\}}$, where $\mathbf{I}$ is the identity matrix, $\mathbf{P}$ represents the ensemble of Pauli operators $(X, Y, Z)$ and $\mathbf{s}^{(0)} = (0,0,1), \mathbf{s}^{(1)} = (\frac{2\sqrt{2}}{3}, 0, -\frac{1}{3}), \mathbf{s}^{(2)} = (-\frac{\sqrt{2}}{3}, \sqrt{\frac{2}{3}}, -\frac{1}{3}), \mathbf{s}^{(3)} = (-\frac{\sqrt{2}}{3}, -\sqrt{\frac{2}{3}}, -\frac{1}{3})$. It is easy to check that $K = 4$ for the tetrahedral POVM. We fix $L = 10$ with $M_{in} = 1000$, re-run the simulations to collect data, and re-train our model and the baselines. The numerical results are reported in Tab. 10. As shown, QuaDiM still outperforms the baselines in this scenario.

Table 10: RMSE of predicting correlations using tetrahedral POVM.

| Method | M=1000 | M=10000 |
|---|---|---|
| CS | 0.0512 | 0.0164 |
| RBFK | 0.0735 | |
| NTK | 0.0747 | |
| RNN | 0.0514 | 0.0163 |
| LLM4QPE | 0.0503 | 0.0141 |
| QuaDiM | 0.0433 | 0.0107 |

