# OpenReview forum: "QuaDiM: A Conditional Diffusion Model For Quantum State Property Estimation"
_ICLR.cc/2025/Conference — ICLR 2025 Poster_

### Official Review · Reviewer_MPMV · 2024-11-02

**Soundness:** 3
**Presentation:** 3
**Contribution:** 3
**Rating:** 6
**Confidence:** 3

**Summary:**

The paper proposes Quantum State Property Estimation using Diffusion Models (QuaDiM), a conditional diffusion model designed to learn and generate distributions of quantum states and accordingly quantum properties, for a given Hamiltonian parameter, from measurement data. Traditionally, auto-regressive models have been used for this task, modeling the quantum state array as a sequential structure despite the lack of inherent sequential ordering in quantum states. Unlike these approaches, QuaDiM denoises and shapes all states in the array simultaneously, as is typical in diffusion models, allowing it to effectively capture non-sequential interactions between spatially separated states. The proposed structure essentially includes an embedding that allows discrete quantum states to be treated as continuous diffusive evolution within a (conditional) diffusion model. The authors validate that the proposed QuaDiM outperformed existing main models, including RNN-based models and LMM4QPE, for large-scale quantum problems (e.g., L = 100).

**Strengths:**

- The proposed QuaDiM outperforms the baselines based on sequential models, e.g., RNN-based and transformer-based ones. It is intriguing, especially to handle very large-scale quantum states (e.g., L = 100) as a benchmark.

- The paper is generally well-written.  I am not an expert in quantum computing, but I was able to follow this paper well and found it interesting to read.

**Weaknesses:**

- This paper is essentially an application paper that applies a diffusion model combined with a text embedding structure to the task of quantum state property estimation. It effectively addresses an interesting topic in the field of quantum computing through appropriate structural selection; however, it is uncertain whether this topic is broadly interesting to the ICLR audience.

- I am not fully convinced why a diffusion model is the most suitable choice for the (non-sequential) quantum property estimation task. To me, this task fundamentally seems to be manageable by other generative models, such as conditional VAEs, GANs, or energy models. The authors mention that these generative models can only handle a single specific quantum state and cannot generalize to unseen states, but I am curious why a diffusion model is capable of generalizing to unseen states. While it is true that diffusion models demonstrate superior performance in terms of both fidelity and diversity, achieving high performance for diffusion models generally requires a large amount of training data. Therefore, the authors would clarify why the proposed QuaDiM would be particularly beneficial for quantum state property estimation, compared to other generative models, in a practical setting (e.g., limited observation as the authors mentioned).

- The authors claim that the primary motivation behind the proposed QuaDiM model is to eliminate the sequential handling of quantum states present in existing models. However, QuaDiM employs an embedding function to represent discrete quantum states as continuous variables, following the approach in [1], which uses token and positional embeddings commonly used in text generation. It seems that this embedding, originally designed for text generation, inherently carries a sequential bias, which makes the motivation and effectiveness of the proposed QuaDiM less convincing to me.

- The authors should explicitly mention the number of parameters and training time of the models used to ensure a fair comparison. Although this is not a critical issue, the network architecture and number of parameters of the main competitor, LLM4QPE, and the proposed QuaDiM appear to be nearly identical.

- Some minor issues: In Table 2, for L = 70, M = 100 and QuaDiM (ours), is the result 0.0597 correct? Line 048, Fig. 2? Line 072, the the?, ...

***
[1] Li, X., Thickstun, J., Gulrajani, I., Liang, P. S., & Hashimoto, T. B. (2022). Diffusion-LM improves controllable text generation. Advances in Neural Information Processing Systems, 35, 4328-4343.

**Questions:**

- Could the authors clarify why diffusion models have a suitable structure for learning non-sequential quantum state estimation compared to other generative models?

- Could the authors explain why the embedding method used in QuaDiM has a structure suitable for non-sequential quantum state estimation?

---

> ### Author Response · Authors · 2024-11-22
> **Response to Reviewer MPMV (Part 1)**
>
> We are truly grateful for your time, detailed feedback, and thoughtful suggestions. We would do our best to provide clarifications step by step. We have addressed the typos in the revised version of the PDF and added necessary clarifications along with additional numerical experiments in the Appendix. The changes made in the PDF are highlighted in blue for easy reference.
>
> **Q1: This paper is essentially an application paper that applies a diffusion model combined with a text embedding structure to the task of quantum state property estimation. It effectively addresses an interesting topic in the field of quantum computing through appropriate structural selection; however, it is uncertain whether this topic is broadly interesting to the ICLR audience.**
>
>
> **A1:** Thank you for the feedbacks. We will try our best to address the reviewer's concerns.
>
> Firstly, we would like to highlight that leaning quantum states hold critical importance for both the quantum computing and deep learning communities.
>
> **Significance of Quantum Applications for Machine Learning**
>    - Applying machine learning methods to quantum many-body problems is a rapidly growing area of interest [1,2]. **Quantum state property estimation (QPE)** is a challenging problem in the field of quantum physics and computing, with implications for **quantum simulation, cryptography, and hardware validation**. By addressing the critical challenge of efficiently predicting quantum state properties in large-scale systems, QuaDiM highlights the potential feasibility of diffusion-based models in quantum sciences.
>    - Examining the application of deep learning approaches in the context of QPE - one of the challenging problems in the field of quantum many-body problem - from both experimental [6,7,8] and theoretical [9,10,11] perspectives holds indispensable value. Our validation of the proposed method’s feasibility is based on empirical and experimental evidence.
>    - Note that many advanced works have been published in learning quantum states in AI conferences, focusing on both the adaptation and application of deep learning methods such as autoregressive models [3], pretraining strategies [4], and deep equilibrium models [5].
>
> Next, we would like to reiterate our contributions and innovations:
>
>    - **Non-Autoregressive Diffusion-Based Approach:** QuaDiM is the **first-ever (to the best of our knowledge) non-autoregressive conditional generative model** specifically designed for QPE. Unlike standard approaches that rely on sequential dependencies (e.g., auto-regressive models), we propose QuaDiM, a diffusion-based model to iteratively denoise Gaussian noise into the target quantum state distribution, with the hope of encouraging **equal treatment of all qubits** and removing bias introduced by sequential modeling. This framework is novel in the QPE context.
>    - **Scalability to Large Systems:** Our empirical evaluation extends to quantum systems up to **100 qubits** (up to $2^{100}$ dimension). This involved a large-scale data collection process from classical simulation. Notably, the simulation and data collection process utilizing the Matrix Product State (MPS) algorithm required nearly two weeks of computation on a cluster equipped with 4 Intel Xeon Gold 6248 CPUs (total cores: 80, total threads: 160). Notably, with reduced sample complexity, our model outperforms state-of-the-art models.
>    - **Generalizability Across Quantum States and Properties**: A significant strength of QuaDiM is its generalizability. Once trained, QuaDiM can predict a variety of quantum state properties—including correlation and entanglement entropy—without the need for retraining. Moreover, it is not limited to a single quantum state but can generalize to previously unseen quantum systems. This includes states that are difficult to prepare on current noisy intermediate-scale quantum (NISQ) devices or impossible to simulate classically. Such flexibility highlights QuaDiM’s potential for addressing broader quantum tasks efficiently.
>
> In conclusion, QuaDiM introduces a novel approach with its non-autoregressive architecture in solving quantum-many body problems, with its scalability and generalizability. We believe its interdisciplinary perspective and empirical results provide a meaningful step forward, and we hope it will be considered as a valuable contribution for the ICLR audience.
>
> [1] Gebhart V, Santagati R, Gentile A A, et al. Learning quantum systems[J]. Nature Reviews Physics, 2023, 5(3): 141-156.
>
> [2] Carrasquilla J. Machine learning for quantum matter[J]. Advances in Physics: X, 2020, 5(1).
>
> [3] Chen Z, Newhouse L, Chen E, et al. Antn: Bridging autoregressive neural networks and tensor networks for quantum many-body simulation. NeurIPS, 2023.
>
> [4] Tang Y, Xiong H, Yang N, et al. Towards LLM4QPE: Unsupervised Pretraining of Quantum Property Estimation and A Benchmark. ICLR, 2024.

---

> ### Author Response · Authors · 2024-11-22
> **Response to Reviewer MPMV (Part 2)**
>
> [5] Wang Z, Liu C, Zou N, et al. Infusing Self-Consistency into Density Functional Theory Hamiltonian Prediction via Deep Equilibrium Models[J]. NeurIPS, 2024.
>
> [6] Carrasquilla J, Torlai G, Melko R G, et al. Reconstructing quantum states with generative models[J]. Nature Machine Intelligence, 2019.
>
> [7] Xiao T, Huang J, Li H, et al. Intelligent certification for quantum simulators via machine learning[J]. npj Quantum Information, 2022.
>
> [8] García-Pérez G, Rossi M A C, Sokolov B, et al. Learning to measure: Adaptive informationally complete generalized measurements for quantum algorithms[J]. PRX quantum, 2021.
>
> [9] Huang H Y, Kueng R, Torlai G, et al. Provably efficient machine learning for quantum many-body problems[J]. Science, 2022.
>
> [10] Lewis L, Huang H Y, Tran V T, et al. Improved machine learning algorithm for predicting ground state properties[J]. Nature Communications, 2024.
>
> [11] Huang H Y, Broughton M, Cotler J, et al. Quantum advantage in learning from experiments[J]. Science, 2022.
>
> **Q2: I am not fully convinced why a diffusion model is the most suitable choice for the (non-sequential) quantum property estimation task. To me, this task fundamentally seems to be manageable by other generative models, such as conditional VAEs, GANs, or energy models. The authors mention that these generative models can only handle a single specific quantum state and cannot generalize to unseen states, but I am curious why a diffusion model is capable of generalizing to unseen states. While it is true that diffusion models demonstrate superior performance in terms of both fidelity and diversity, achieving high performance for diffusion models generally requires a large amount of training data. Therefore, the authors would clarify why the proposed QuaDiM would be particularly beneficial for quantum state property estimation, compared to other generative models, in a practical setting (e.g., limited observation as the authors mentioned).**
>
> **A2:** We sincerely thank the reviewer for their thoughtful and constructive feedback.
>
> It is worth noting that predicting properties of quantum states using deep learning techniques is indeed a cutting-edge research area [1]. We acknowledge that generative models like VAEs, GANs, and energy-based methods remain valuable tools in quantum machine learning. Our work with QuaDiM demonstrates the potential of non-autoregressive approaches and aims to inspire further exploration of these methods for QPE.
>
> In the following response, we would try our best to clarify the motivation behind our choice of diffusion models, highlight their potential advantages over alternative generative methods, and provide further insights into their suitability for quantum state property estimation tasks. We are happy to maintain an open stance for further discussion with the reviewer.

---

> ### Author Response · Authors · 2024-11-22
> **Response to Reviewer MPMV (Part 3)**
>
> ### **1. Clarification of Motivations and Contributions**
>
> To address your concerns more effectively, we would like to begin by briefly restating the motivations behind our work.
> 1. **Encourage to Eliminate Autoregressive Bias**: QuaDiM leverages diffusion models to provide a novel non-autoregressive method for quantum state property estimation (QPE), a challenging problem of quantum many-body problems [1,2]. Recent studies [3,4] have found that the inherent constraints of autoregressive methods—specifically, the pre-defined factorization order (e.g., left-to-right or right-to-left order in 1D chains, or zig-zag orders in 2D systems)—often **introduce biases into the model**. This bias **influences** the ability of variational models including deep learning models to accurately describe correlations among qubits and **constrains** the expressivity of these models in learning quantum systems. **Unlike autoregressive methods, QuaDiM encourages to eliminate the need of predefined qubit ordering.**
> 2. **Generalization to unseen states**: Diffusion models learn to progressively refine Gaussian noise into the target distribution conditioned on the hidden representations of physical variables, with the hope of capturing the entire distribution of quantum states rather than a specific set of states. This mechanism naturally enables generalization to unseen quantum states since we can sample new measurement outcomes from the trained models conditioned on the unseen physical variables (even hard to simulate in a real quantum computer or hard to simulate classically) out of the training set.
> 3. **Large-scale experiments and Practical Feasibility**: QuaDiM demonstrates superior scalability and generalization capabilities compared to both learning-free baseline classical shadow (CS) and learning-based SOTA LLM4QPE (autoregressive model). It accurately predicts quantum properties like correlation and entanglement entropy for systems up to 100 qubits (up to $2^{100}$ dimension). Moreover, our results reveal that QuaDiM achieves lower prediction errors with reduced training data and fewer samples during inference, making it more practical for real-world scenarios where quantum measurements are expensive and time-consuming both in quantum experiments (**1.6\$ per second for IBM quantum computer**) and classical simulations.
>
> ### **2. Potential Advantages Over VAEs, GANs, and Energy-Based Models**
>
> Our choice of diffusion models was primarily motivated by their intuitive alignment with the requirements (three points above) of quantum property estimation; the following represents our preliminary analysis, and we remain open to further discussions and deeper explorations with the reviewer on this topic.
> 1. **Limitations of VAEs, GANs and Energy-Based Models**:
>    - VAEs are effective for learning latent representations but often struggle with accurately modeling sharp or highly complex distributions [5].
>    - While GANs can model complex distributions, their training is notoriously unstable due to issues like mode collapse [6]. This instability is particularly problematic for quantum systems, where capturing the full range of quantum state correlations is crucial.
>    - Energy-based models (such as RBMs [7,8] for modeling quantum states) are powerful but computationally expensive, especially when scaling to quantum systems with a large number of qubits. Diffusion models provide a more computationally efficient alternative while retaining the ability to model complex distributions.
>
> 2. **Practical Efficiency of Diffusion Models**:
>    - As shown in our experiments, QuaDiM achieves state-of-the-art performance even with a limited number of training samples (Table 1, 2 and Figure 5). The iterative denoising process encourages to high fidelity and diversity in generated quantum states, outperforming baselines in predictive accuracy.
>
> In conclusion, we deeply appreciate the reviewer’s insightful feedback, which has allowed us to reflect on and further clarify the motivations and design choices behind QuaDiM.
>
> While our work demonstrates the potential of diffusion models for quantum state property estimation and highlights their advantages in terms of scalability, generalization, and efficiency, we acknowledge that this is an emerging and dynamic research area. We hope that our contributions can serve as a foundation to inspire further exploration and innovation, including the application of alternative generative models.
>
> We remain open to constructive discussions and suggestions from the reviewer to strengthen this line of research further. Thank you again for your valuable input.
>
> [1] Gebhart V, Santagati R, Gentile A A, et al. Learning quantum systems[J]. Nature Reviews Physics, 2023, 5(3): 141-156.
>
> [2] Carrasquilla J. Machine learning for quantum matter[J]. Advances in Physics: X, 2020, 5(1).

---

> ### Author Response · Authors · 2024-11-22
> **Response to Reviewer MPMV (Part 4)**
>
> [3] Bortone M, Rath Y, Booth G H. Impact of conditional modelling for a universal autoregressive quantum state[J]. Quantum, 2024, 8: 1245.
>
> [4] Ibarra-García-Padilla E, Lange H, Melko R G, et al. Autoregressive neural quantum states of Fermi Hubbard models[J]. arXiv preprint arXiv:2411.07144, 2024.
>
> [5] Vahdat A, Kautz J. NVAE: A deep hierarchical variational autoencoder[J]. Advances in neural information processing systems, 2020.
>
> [6] Saxena D, Cao J. Generative adversarial networks (GANs) challenges, solutions, and future directions[J]. ACM Computing Surveys (CSUR), 2021, 54(3): 1-42.
>
> [7] Carleo G, Troyer M. Solving the quantum many-body problem with artificial neural networks[J]. Science, 2017, 355(6325): 602-606.
>
> [8] Deng D L, Li X, Das Sarma S. Quantum entanglement in neural network states[J]. Physical Review X, 2017, 7(2): 021021.
>
>
>
> **Q3: The authors claim that the primary motivation behind the proposed QuaDiM model is to eliminate the sequential handling of quantum states present in existing models. However, QuaDiM employs an embedding function to represent discrete quantum states as continuous variables, following the approach in [1], which uses token and positional embeddings commonly used in text generation. It seems that this embedding, originally designed for text generation, inherently carries a sequential bias, which makes the motivation and effectiveness of the proposed QuaDiM less convincing to me.**
>
> **A3:** We appreciate the reviewer's insightful feedback on the use of embeddings inspired by language models for quantum measurement data. We would like to clarify our reasoning and demonstrate the appropriateness of this approach.
>
> 1. **Relation Between Quantum States and Classical Joint Distributions**
>    Consider the quantum state represented as a wave function
>    $$
>    |\psi\rangle = \sum_{\sigma_1=1}^{K}\cdots\sum_{\sigma_L=1}^{K} \Phi(\sigma_1,\ldots,\sigma_L)|\sigma_1,\ldots,\sigma_L\rangle,
>    $$ where $\Phi(\sigma_1,\ldots,\sigma_L)$ is the amplitude of the basis $(\sigma_1,\ldots,\sigma_L)$ satisfying the normalization condition (according to Born rule in quantum mechanics) $\sum_{\sigma_1=1}^{K}\cdots\sum_{\sigma_L=1}^{K} |\Phi(\sigma_1,\ldots,\sigma_L)|^2 = 1$.
>    The quantum state inherently exhibits a probabilistic nature governed by the Born rule, where measurement outcomes follow a joint probability distribution:
> $$
>    P(\sigma_1, \sigma_2, \dots, \sigma_L) = |\Phi(\sigma_1, \sigma_2, \dots, \sigma_L)|^2,
> $$ The joint distribution $p$ is a valid probability distribution since $p(\sigma_1,\ldots,\sigma_L)\geq 0$ and $\sum_{\sigma_1}\cdots\sum_{\sigma_L}p(\sigma_1,\ldots,\sigma_L)=1$.
> This formulation mirrors classical joint distributions frequently encountered in NLP tasks, where dependencies among tokens (qubits) are encoded probabilistically. Representing quantum measurement data as continuous embeddings aligns with this probabilistic perspective and facilitates modeling the complex dependencies among qubits.
>
> 2. **Analogies Between Quantum Measurement Data and Text Corpora**
>    - A **quantum measurement string** (e.g., $\sigma = (\sigma_1, \dots, \sigma_L)$) is analogous to a **sentence** in a corpus, encoding interdependencies among qubits.
>    - The possible measurement outcomes for each qubit correspond to **tokens**, akin to words in a vocabulary $\mathcal{V}$ (with size $|\mathcal{V}|=K$ and $K$ is the total number of possible measurement outcomes, which is 6 in our experiments), while the quantum system's physical condition can be likened to **contextual information** that influences the sentence structure.
>    - Given these parallels, applying language model-inspired embeddings to quantum measurement data is a natural extension, facilitating scalable and generalizable representation learning.

---

> ### Author Response · Authors · 2024-11-22
> **Response to Reviewer MPMV (Part 5)**
>
> 3. **Reasoning Behind Positional Embeddings**
>    Traditional autoregressive models used for modeling quantum states introduce sequential bias through predefined sampling orders (e.g., left-to-right), as evidenced by [1,2]. In contrast, our diffusion-based framework generates all measurement outcomes simultaneously, avoiding sequential dependencies during sampling.
>    - **Positional embeddings** in our context are employed to capture the structural information among qubits. Additionally, we further investigate the model’s RMSE in predicting correlations when using **relative positional embedding** (follow the implementation in [3]) and **no positional embedding** at all. Due to time constraints, we only report results for $L=100, M_{in}=1000$. The experimental results are provided below (lower is better) and are involved in the PDF's Appendix.
>    Experimental results show that absolute and relative positional embeddings yield comparable performance, but removing positional information altogether significantly degrades the model's ability to predict quantum correlations. This underscores the embeddings' role in preserving spatial relationships among qubits.
>    - We acknowledge that the current positional encoding method may not perfectly reflect quantum system-specific structures. As a step forward, we plan to explore customized positional encodings tailored to quantum systems in future work.
>
> |                 | Absolute Position Embedding | Relative Position Embedding | No Position Embedding |
> | --------------- | --------------------------- | --------------------------- | --------------------- |
> | $M_{out}=100$   | 0.1686                      | 0.1681                      | 0.4527                |
> | $M_{out}=1000$  | 0.0478                      | 0.0482                      | 0.3269                |
> | $M_{out}=10000$ | 0.0125                      | 0.0139                      | 0.2895                |
> | $M_{out}=20000$ | 0.0098                      | 0.0110                      | 0.2148                |
>
>
> In conclusion, we appreciate the reviewer’s thoughtful concerns regarding the use of embeddings inspired by language models for quantum measurement data. By highlighting the relationship between quantum systems and classical joint distributions, as well as drawing analogies to text corpora, we hope our statement clarifies the rationale behind this design choice. Our empirical results further demonstrate the effectiveness of positional embeddings in capturing structural information among qubits, while recognizing that there is room to refine these embeddings to better align with quantum-specific characteristics.
>
> We sincerely thank the reviewer for their valuable feedback, which has provided us with meaningful directions for future research to enhance the adaptability and rigor of our approach.
>
> [1] Bortone M, Rath Y, Booth G H. Impact of conditional modelling for a universal autoregressive quantum state[J]. Quantum, 2024, 8: 1245.
>
> [2] Ibarra-García-Padilla E, Lange H, Melko R G, et al. Autoregressive neural quantum states of Fermi Hubbard models[J]. arXiv preprint arXiv:2411.07144, 2024.
>
> [3] Shaw P, Uszkoreit J, Vaswani A. Self-attention with relative position representations. arXiv preprint arXiv:1803.02155, 2018.
>
> **Q4: The authors should explicitly mention the number of parameters and training time of the models used to ensure a fair comparison. Although this is not a critical issue, the network architecture and number of parameters of the main competitor, LLM4QPE, and the proposed QuaDiM appear to be nearly identical.**
>
> **A4:** The main part of our proposed QuaDiM is the transformer, which shares a similar structure of the sota baseline LLM4QPE, a transformer-based autoregressive model. For a fair comparison, the transformer of both is fixed at 4 heads, 4 layers, and 128 hidden dimension, along with a conditional embedding layer FFN to encoding the physical parameters. The total parameters $\approx$ 800,000. On a single 2080Ti GPU and for $L=100$ with batch size is 64 and 100,000 iterations, the training time of QuaDiM and LLM4QPE is nearly 9 hours.
>
> **Q5: Some minor issues: In Table 2, for L = 70, M = 100 and QuaDiM (ours), is the result 0.0597 correct? Line 048, Fig. 2? Line 072, the the?**
>
> **A5:** Thank you for the feedback. We have corrected these errors in the updated version of the paper. That is, 0.0597 -> 0.5970 and delete double the.
>
> **Q6: Could the authors clarify why diffusion models have a suitable structure for learning non-sequential quantum state estimation compared to other generative models?**
>
> **A6:** We sincerely thank the reviewer for the question. Kindly refer to our response in A2 for details.
>
>
> **Q7: Could the authors explain why the embedding method used in QuaDiM has a structure suitable for non-sequential quantum state estimation?**
>
> **A7:** Thank you for your thoughtful question. Please see our detailed response provided in A3 for further clarification.

---

> ### Author Response · Authors · 2024-11-25
> **Sincerely Awaiting Your Feedback**
>
> Dear Reviewer MPMV,
>
> I hope this message finds you well. As the rebuttal period is quickly approaching its deadline, we kindly request your valuable feedback to ensure a comprehensive revision. Your insights are crucial to us, and we look forward to your response.
>
> Best regards

---

> ### Author Response · Authors · 2024-11-26
> **Anticipating Your Feedback with Gratitude**
>
> Dear Reviewer MPMV,
>
> As the discussion window is approaching its closure, we are eager to hear back from you regarding our submission. We wish to ensure that our responses have adequately addressed your concerns. Should there be any remaining issues or new concerns that have arisen, please do not hesitate to inform us. We are more than willing to provide further clarifications and engage in discussions to enhance the quality and relevance of our work.
>
> Thank you very much for your review and support.
>
> Best regards

---

> > ### Comment · Reviewer_MPMV · 2024-11-27
> >
> > I appreciate the authors' detailed responses, which have clarified my questions regarding the motivation behind the use of the diffusion model and positional encoding. I now recognize this paper as a solid contribution to the application of quantum science, and accordingly have increased the review score. That said, while the proposed algorithm is well-defined and thoughtfully constructed, the absence of substantial advancements in ML theory or notable algorithmic innovations makes it challenging to provide a higher evaluation.

---

> > > ### Author Response · Authors · 2024-11-27
> > >
> > > Dear Reviewer MPMV,
> > >
> > > We sincerely thank you for your insightful feedback and for considering an increase in your score. Your detailed suggestions have greatly contributed to improving the clarity and depth of our manuscript. We appreciate the time and effort you have invested in reviewing our work and providing constructive recommendations.
> > >
> > > Best regards

---

### Official Review · Reviewer_kHhj · 2024-11-02

**Soundness:** 3
**Presentation:** 3
**Contribution:** 1
**Rating:** 6
**Confidence:** 3

**Summary:**

This paper addresses the challenge of Quantum state property estimation (QPE) in quantum many-body physics, focusing on predicting characteristics like correlation and entanglement entropy from measurement data. The authors introduce QuaDiM, a non-autoregressive generative model using diffusion models, which avoids the need for an intrinsic qubit ordering and treats all qubits equally and unbiasedly. QuaDiM learns to map physical variables to ground state properties during offline training and can sample from learned distributions for unseen variables to predict unknown quantum states. Empirical results on the 1D anti-ferromagnetic Heisenberg model with up to 100 qubits show that QuaDiM outperforms baseline models, particularly auto-regressive approaches, under limited training data and reduced inference sample complexity.

**Strengths:**

I am not an expert in quantum physics, so I can only provide my insights from the perspective of general generative models.

First of all, the paper provides empirical evidence that models like diffusion models, which can capture the correlations between qubits, can outperform sequential-based models such as transformers and RNNs. This observation is an important finding in itself and can potentially steer research focus in the relevant AI4Science area.

**Weaknesses:**

As mentioned earlier, I would like to raise some concerns from a generative model perspective:

Figure 3 is not informative to me; I can barely discern any difference between QuaDiM and LLM4QPE visually.

Based on Figure 2, the methodological novelty appears limited.

Could you please provide details about the computational resources used for the experiments? For example, the number of GPUs used, the training time, and the model size.

It would be beneficial if the authors could provide the standard deviation over seeds in Tables 1 and 2.

Minor typo:

Line 72: there is a double 'the'.

**Questions:**

Please correct me if I was wrong in terms of my previous comments :)

---

> ### Author Response · Authors · 2024-11-22
> **Response to Reviewer kHhj (Part 1)**
>
> We sincerely appreciate the time and effort you have taken to provide thoughtful feedback and helpful suggestions. We are glad to address your questions in detail. We have addressed the typos in the revised version of the PDF and added necessary clarifications along with additional numerical experiments in the Appendix. The changes made in the PDF are highlighted in blue for easy reference.
>
> **Q1: Figure 3 is not informative to me; I can barely discern any difference between QuaDiM and LLM4QPE visually.**
>
> **A1:** Thank you for the valuable feedback. In Figure 3 of our main paper, our intention is to present a comparative visualization of the performance of our proposed model, QuaDiM, and the baseline LLM4QPE under varying numbers of samples $M_{out}$. Each subplot corresponds to a specific $M_{out}$, and the root mean squared error (RMSE) values, which indicate the predictive accuracy (lower is better), are highlighted in the upper-right corner of each subplot for reference.
>
> We recognize that the color contrast in Figure 3 may not be sufficiently distinct. To address this, **we include an additional plot Figure 6 in the Appendix**, where we present a more fine-grained visualization of the prediction performance of QuaDiM. In this figure, each point represents the absolute error (lower is better) between the predicted correlation and the ground truth for all pairs of qubits across different $M_{out}$ values.
>
> From Figure 6, it can be seen that **QuaDiM consistently outperforms LLM4QPE** and QuaDiM achieves **lower variance** compared with LLM4QPE. Specifically, the absolute errors for QuaDiM is more narrowly concentrated around lower values, highlighting its consistent predictive performance. In contrast, LLM4QPE exhibits wider variability.
>
> These results highlight the practical advantages of QuaDiM, especially in scenarios where sampling a large number of measurements is computationally expensive. We hope this additional clarification enhances your understanding of our model's strengths.
>
>
> **Q2: Based on Figure 2, the methodological novelty appears limited.**
>
> **A2:** We sincerely thank the reviewer for the feedback. We would like to highlight that Quantum state property estimation (QPE) is a challenging problem in the field of quantum physics and computing [1,2], with implications for quantum simulation, cryptography, and hardware validation. By investigating the critical challenge of efficiently predicting quantum state properties in large-scale systems (up to $2^{100}$ dimensions studied in our work), QuaDiM highlights the practical feasibility of deep learning in quantum sciences.
>
> Firstly, We emphasize leveraging deep learning frameworks to QPE, **a challenging problem in quantum many-body problems that has garnered increasing attention in recent years.** Many advanced works have been published in AI conferences, focusing on both the adaptation and application of deep learning methods such as autoregressive models [3], pretraining strategies [4], and deep equilibrium models [5]. Beyond these, a series of contributions have emerged, arising from both experimental [6,7,8] and theoretical [9,10,11] perspective. We aspire for this paper to serve as a source of inspiration for researchers across both the AI and quantum computing domains.

---

> ### Author Response · Authors · 2024-11-22
> **Response to Reviewer kHhj (Part 2)**
>
> Note that while diffusion models and transformers are established in the ML community, their adaptation to the quantum many-body domain is **non-trivial**. Unlike typical applications like language modeling, quantum measurement data involve **non-sequential, exponentially large dimensions** with complex entanglement [9]. QuaDiM encourages to alleviate these issues by introducing a **non-autoregressive generative framework** for QPE. Here, we would like to briefly reiterate our contributions and innovations:
> - **Non-Autoregressive Diffusion-Based Approach:** QuaDiM is the **first-ever (to the best of our knowledge) non-autoregressive conditional generative model** specifically designed for QPE. Unlike standard approaches that rely on sequential dependencies (e.g., auto-regressive models), QuaDiM employs **diffusion models** to iteratively denoise Gaussian noise into the target quantum state distribution, with the hope of encouraging **equal treatment of all qubits** and removing bias introduced by sequential modeling. This framework is novel in the QPE context, addressing the limitations of existing machine learning models that fail to capture non-sequential, high-dimensional entanglement structures.
> - **Scalability to Large Systems:** Our empirical evaluation extends to quantum systems up to **100 qubits** (up to $2^{100}$ dimension). This involved a large-scale data collection process from classical simulation. Notably, the simulation and data collection process utilizing the Matrix Product State (MPS) algorithm required nearly two weeks of computation on a cluster equipped with 4 Intel Xeon Gold 6248 CPUs (total cores: 80, total threads: 160). Notably, with reduced sample complexity, our model outperforms state-of-the-art baselines.
>
> We again sincerely appreciate the reviewers’ thoughtful feedback. We hope that this clarification will aid in further understanding and help to underscore QuaDiM's potential as a valuable contribution to the field.
>
> [1] Gebhart V, Santagati R, Gentile A A, et al. Learning quantum systems[J]. Nature Reviews Physics, 2023, 5(3): 141-156.
>
> [2] Carrasquilla J. Machine learning for quantum matter[J]. Advances in Physics: X, 2020, 5(1).
>
> [3] Chen Z, Newhouse L, Chen E, et al. Antn: Bridging autoregressive neural networks and tensor networks for quantum many-body simulation. NeurIPS, 2023.
>
> [4] Tang Y, Xiong H, Yang N, et al. Towards LLM4QPE: Unsupervised Pretraining of Quantum Property Estimation and A Benchmark. ICLR, 2024.
>
> [5] Wang Z, Liu C, Zou N, et al. Infusing Self-Consistency into Density Functional Theory Hamiltonian Prediction via Deep Equilibrium Models. NeurIPS, 2024.
>
> [6] Carrasquilla J, Torlai G, Melko R G, et al. Reconstructing quantum states with generative models[J]. Nature Machine Intelligence, 2019.
>
> [7] Xiao T, Huang J, Li H, et al. Intelligent certification for quantum simulators via machine learning[J]. npj Quantum Information, 2022.
>
> [8] García-Pérez G, Rossi M A C, Sokolov B, et al. Learning to measure: Adaptive informationally complete generalized measurements for quantum algorithms[J]. PRX quantum, 2021.
>
> [9] Huang H Y, Kueng R, Torlai G, et al. Provably efficient machine learning for quantum many-body problems[J]. Science, 2022.
>
> [10] Lewis L, Huang H Y, Tran V T, et al. Improved machine learning algorithm for predicting ground state properties[J]. Nature Communications, 2024.
>
> [11] Huang H Y, Broughton M, Cotler J, et al. Quantum advantage in learning from experiments[J]. Science, 2022.
>
> **Q3: Could you please provide details about the computational resources used for the experiments? For example, the number of GPUs used, the training time, and the model size.**
>
> **A3:** Thanks for your questions. Utilizing classical machine learning methods to address quantum-related problems primarily faces challenges due to the significant complexity involved in simulating quantum systems. Our simulations reached up to 100 qubits (dimensions of $2^{100}$).
>
> The computational bottleneck is not training and inference with deep learning models, but generating synthetic data, i.e. classically generating target states and simulating quantum measurements. (Needless to say, this classical bottleneck does not occur in actual experiments.) Even using Matrix Product State (MPS) algorithm to accelerate simulation, it spends nearly two weeks collecting the data on a cluster equipped with 4 Intel Xeon Gold 6248 CPUs (total cores: 80, total threads: 160).

---

> ### Author Response · Authors · 2024-11-22
> **Response to Reviewer kHhj (Part 3)**
>
> For training and inference in our experiments, a single 2080Ti GPU is used. Training our QuaDiM (the main part is the transformer with 4 heads, 4 layers, and 128 hidden dimension, and a conditional embedding layer FFN to encoding the physical parameters, totally $\approx$ 800,000 parameters) requires nearly 9 hours. We report the inference speed under the task of predicting correlations with $L=100, M_{in}=1000$ and $M_{out}=1000$. The results are as follows.
>
> | Method              | RMSE   | Generated samples per sec. |
> | ------------------- | ------ | -------------------------- |
> | CS                  | 0.0547 | -                          |
> | LLM4QPE             | 0.0531 | 14.6                       |
> | QuaDiM ($T_f=2000$) | 0.0478 | 5.7                        |
> | QuaDiM ($T_f=1000$) | 0.0537 | 8.1                        |
> | QuaDiM ($T_f=500$)  | 0.0541 | 12.7                       |
> | QuaDiM ($T_f=100$)  | 0.0882 | 37.4                       |
>
> As shown in the table above, when reducing inference to $T_f=500$ diffusion steps on a single GPU (2080Ti), QuaDiM achieves a lower RMSE score compared to the classical shadow baseline while demonstrating an inference speed comparable to LLM4QPE.
>
> We acknowledge that the sampling time of diffusion-based models poses a significant challenge when applied to quantum many-body problems (as we already discussed in the limitation and conlusion section of our paper). This limitation is inherent to the nature of diffusion-based approaches. In our future work, we seek to explore the incorporation of techniques such as consistency training, potentially enabling one-step sampling for diffusion-based applications in the domain of quantum many-body problems. We remain optimistic about these possibilities and appreciate the reviewers' understanding of this ongoing effort.
>
>
>
> **Q4: It would be beneficial if the authors could provide the standard deviation over seeds in Tables 1 and 2.**
>
> **A4:** We sincerely thank the reviewers for the suggestion. In the Appendix, we have provided results including standard deviations, which are over four different random seed initializations. To comply with the page limitation by the ICLR committee, the new tables with standard deviations have been temporarily placed in the appendix. Once the paper is officially published, the tables in the main text will be updated accordingly.
>
>
> **Q5: Minor typo: Line 72: there is a double 'the'.**
>
> **A5:** Thank you for your feedback. We have correct this in the updated version of the paper.

---

> ### Author Response · Authors · 2024-11-25
> **Sincerely Awaiting Your Feedback**
>
> Dear Reviewer kHhj,
>
> I hope this message finds you well. As the rebuttal phase deadline is approaching, we would like to kindly request your further feedback to ensure a comprehensive revision and to ensure our clarifications have addressed your concerns effectively. Your feedback is invaluable, and we greatly appreciate your time and insights.
>
> Best regards

---

> ### Comment · Reviewer_kHhj · 2024-11-25
> **thanks for response**
>
> Thanks for the response.
>
> I have last question, can you report the plot of the number of trained samples vs evaluation results (hopefully it won't be too complicated, and I am sorry for the last minute request)? I am curious about this. I slightly doubt that the training does not need such large scale of the data (2^100) since the qubit has structure.

---

> ### Author Response · Authors · 2024-11-25
>
> We sincerely thank the reviewer for the additional feedback. Your suggestions are invaluable in helping us refine our manuscript, and we will do our best to address your concerns thoroughly.
>
> It is worth noting that predicting properties of quantum states using deep learning techniques is indeed a cutting-edge research area, and the efficiency of these techniques in this domain remains an open question. We would like to briefly share some of our observations here and maintain an open stance for further discussion with the reviewer.
>
> From a theoretical perspective [1,2], for learning and predicting correlations (equivalent to the expectation of local observables) and two-body entropy as investigated in our paper, the required $M_{in}$ **does not** scale exponentially and **is independent of the number of qubits $L$**. However, in general cases, reconstruction of the full representations of the quantum states (known as quantum tomography, a orthogonal research direction to our work) typically requires exponentially large number of training measurements.
>
> From a numerical perspective, recent work [3] based on autoregressive models has explored new deep learning methodologies to reduce $M_{in}$, achieving SOTA results. In our paper, we approached this challenge from the perspective of the **non-sequential nature of the distribution corresponding to quantum states**. Our experiments demonstrated that the non-autoregressive diffusion model achieves a reduction in $M_{in}$ compared to the SOTA autoregressive model when predicting correlations and entropy of the 1-D Heisenberg model (as shown in Figure 5).
>
> Additionally, we provide the RMSE (lower is better) of predicting correlations for both our model and the baselines under varying $M_{in}$ values. Due to time constraints, we report results only for $L=100, M_{out}=1000$ in the following table.
>
> |         | $M_{in}=200$ | $M_{in}=400$ | $M_{in}=600$ | $M_{in}=800$ | $M_{in}=1000$ |
> |---------|--------------|--------------|--------------|--------------|---------------|
> | CS      | 0.1296       | 0.0814       | 0.0682       | 0.0598       | 0.0547        |
> | RNN     | 0.1679       | 0.1132       | 0.0976       | 0.0853       | 0.0806        |
> | LLM4QPE | 0.1358       | 0.0781       | 0.0633       | 0.0565       | 0.0531        |
> | QuaDiM  | **0.1224**   | **0.0735**   | **0.0597**   | **0.0526**   | **0.0478**    |
>
>
> These discussions and results will be involved in the revised version of our paper. Once again, we deeply appreciate the reviewer’s feedback. In the limited time remaining for the rebuttal process, we are eager to further discuss any remaining concerns or suggestions you might have. Thank you for your constructive engagement and valuable insights!
>
> [1] Huang H Y, Kueng R, Torlai G, et al. Provably efficient machine learning for quantum many-body problems[J]. Science, 2022.
>
> [2] Lewis L, Huang H Y, Tran V T, et al. Improved machine learning algorithm for predicting ground state properties[J]. Nature Communications, 2024.
>
> [3] Tang Y, Xiong H, Yang N, et al. Towards LLM4QPE: Unsupervised Pretraining of Quantum Property Estimation and A Benchmark. ICLR, 2024.

---

> ### Author Response · Authors · 2024-11-27
> **Looking Forward to Your Reply**
>
> Dear Reviewer kHhj,
>
> I hope this message finds you well. I would like to **kindly ask whether our latest responses adequately addressed your concerns and clarified the points raised in your previous question.** Your detailed and constructive feedback has been invaluable in shaping our work, and we deeply value your insights.
>
> Our primary goal is to ensure that the paper meets the rigorous standards of ICLR while also contributing meaningfully to the intersection of deep learning and quantum physics. If there are any remaining concerns or areas where further clarification might help, we would be more than happy to address them in the spirit of collaboration and continuous improvement.
>
> Thank you once again for your time and effort in reviewing our responses for your last question. We greatly appreciate your guidance in helping us refine our work.
>
> Warm regards

---

> > ### Comment · Reviewer_kHhj · 2024-11-28
> > **Thanks for the reply**
> >
> > Thank you for the clarification. After revisiting [1], which shares the same technical backbone (diffusion model), and reviewing the response regarding training scalability, I realized that the task difficulty of [1] is significantly higher than what the author proposed. Therefore, the criticism directed at [1] seems unfair, although I agree that [1] cannot yet scale to hundreds of qubits. I suggest the author clarify this point in the paper. It took me some time to understand the fundamental differences, and I hope future readers will not have to go through the same process.
> >
> > I will increase the score once the manuscript is updated.
> >
> >
> > [1] Yuchen Zhu, Tianrong Chen, Evangelos A Theodorou, Xie Chen, and Molei Tao. Quantum state
> > generation with structure-preserving diffusion model. arXiv preprint arXiv:2404.06336, 2024

---

> ### Author Response · Authors · 2024-11-28
> **Further Response to Reviewer kHhj (Part 1)**
>
> We sincerely thank the reviewers for their thoughtful feedback and the time and effort they have devoted to reviewing our work.
>
> Before addressing the specific concerns, we would like to clarify that we were aware of the method proposed in [1] prior to writing this paper, and we have discussed it in our paper (Line 140).
>
> However, after thorough analysis, we contend that their method **falls short** of providing a viable approach for applying machine learning (diffusion-based models in that paper) to quantum physics. Their technique, which involves using density matrix (a $2^{2L}$-size  complex-valued matrix) of quantum states directly as inputs, is largely **impractical** due to the foundational requirement in learning of quantum states: **the necesity to first gather quantum measurement data via quantum measurements on the states themselves.**
>
> Given this practical limitation, the numerical experiments in [1] are restricted to **toy demos** of systems with only $L=4$ qubits (in contrast to our experiments, which scale up to $L=100$ qubits). To further enhance the reviewers’ understanding, we provide a detailed discussion below on the impracticality of [1] and the key distinctions between their approach and ours. We are deeply grateful for the reviewers’ patience in reading our response.
>
> 1. **The impracticality of using full density matrix as input in [1]:**
>    One of the principal features distinguishing classical systems from quantum many-body systems is that quantum systems require exponentially many parameters in the system size to fully specify the state [2].
>
>    To obtain information from a real quantum system, measurement is required (Such as the pauli measurement used in our paper). Measurement results in discrete outcomes due to the collapse of quantum states. Obtaining the full density matrix to construct the training set in [1] typically demands an exponentially large number of quantum measurements on an actual quantum computer, followed by extensive post-processing of these measurement outcomes, which generally entails exponential overhead [2]. As a result, in [1] each sample in the training set (i.e., a full density matrix representing a specific quantum state) would require **exponential storage space** and **computational resources**, making this approach **impractical** for real-world applications.
>
> 2. **The orthogonality of our task (quantum state property estimation, QPE) to quantum state tomography (QST):**
>    The task in [1] is more aligned with QST (however, it is not a typical QST approach, as [1] explicitly requires the density matrix to be used as training input, whereas QST typically does not impose such a constraint), which seeks to reconstruct the full density matrix. Neural network-based QST [3] generally approximates the probability distribution over the **outcomes of an informationally complete measurement** using a variational manifold represented by a neural network. **However**, [1] bypasses measurement data entirely and directly uses the density matrix as input. To the best of our knowledge, this approach is **impractical**, as it **contradicts** the fundamental constraints of data acquisition, i.e., quantum measurement in quantum systems.
>
> 3. **Our focus on quantum state property estimation (QPE):**
>    Unlike QST, QPE specifically targets the prediction of specific properties of quantum states **without** reconstructing the full density matrix. This task has recently garnered significant **attention** in the quantum physics and machine learning communities, with several cutting-edge theoretical [4,5] and empirical [6,7] studies already published. In terms of model design, unlike [1]’s approach of directly modeling the density matrix using generative models, our work delves into bridging the gap between quantum states and the **classical joint distributions** modeled by generative models. Additionally, we explore how the continuous latent variables of diffusion models can be **decoded** into discrete quantum measurement data while **preserving** physical validity. We have elaborated on these in response A3 to Reviewer 9ozo, and response A7 and A8 to Reviewer oe1d. These discussion has also been incorporated into Appendix C of the revised version of our paper. We appreciate the reviewers’ patience in reviewing this material.
>
> 4. **The Potential of Adapting Our Model to QST:**
>    We also note that a similar concern was raised by another reviewer regarding whether our model could be used to obtain the log probabilities of each measurement outcome. For a detailed discussion from our preliminary and immature view, please refer to our response to Reviewer 9ozo in A7.

---

> ### Author Response · Authors · 2024-11-28
> **Further Response to Reviewer kHhj (Part 2)**
>
> In summary, we argue that the approach in [1] is fundamentally limited in practical applications due to the **infeasibility** of using full density matrices as input. Our work, in contrast, is well-aligned with recent advancements in QPE and provides **practical** solutions to bridge quantum states and classical generative models. To provide further clarity, we will include additional discussion about [1] in the revised version of our paper. Due to the page limitations set by the ICLR committee, we have temporarily placed the discussion in the appendix G, but they will be updated in the main text once the paper is officially published.
>
> Once again, we sincerely thank the reviewers for their valuable comments, which have greatly contributed to improving the quality and clarity of our paper.
>
> [1] Yuchen Zhu, Tianrong Chen, Evangelos A Theodorou, Xie Chen, and Molei Tao. Quantum state generation with structure-preserving diffusion model. arXiv preprint arXiv:2404.06336, 2024
>
> [2] Cramer M, Plenio M B, Flammia S T, et al. Efficient quantum state tomography[J]. Nature communications, 2010, 1(1): 149.
>
> [3] Torlai G, Mazzola G, Carrasquilla J, et al. Neural-network quantum state tomography[J]. Nature physics, 2018, 14(5): 447-450.
>
> [4] Huang H Y, Kueng R, Torlai G, et al. Provably efficient machine learning for quantum many-body problems[J]. Science, 2022.
>
> [5] Lewis L, Huang H Y, Tran V T, et al. Improved machine learning algorithm for predicting ground state properties[J]. Nature Communications, 2024.
>
> [6] Tang Y, Xiong H, Yang N, et al. Towards LLM4QPE: Unsupervised Pretraining of Quantum Property Estimation and A Benchmark. ICLR, 2024.
>
> [7] Wu Y D, Zhu Y, Wang Y, et al. Learning quantum properties from short-range correlations using multi-task networks[J]. Nature Communications, 2024, 15(1): 8796.

---

> > ### Comment · Reviewer_kHhj · 2024-11-28
> > **thanks for the reply**
> >
> > Thanks for the careful clarification. I have increased the score.

---

### Official Review · Reviewer_9ozo · 2024-11-03

**Soundness:** 3
**Presentation:** 3
**Contribution:** 3
**Rating:** 8
**Confidence:** 3

**Summary:**

This paper introduces QuaDiM, a non-autoregressive diffusion model for quantum state property estimation (QPE). Unlike traditional autoregressive approaches that impose a (potentially biased) sequential ordering on qubits, QuaDiM treats all qubits equally through an iterative denoising process. The model is evaluated on the 1D anti-ferromagnetic Heisenberg model for systems up to 100 qubits, demonstrating superior performance in predicting correlation and entanglement entropy compared to baseline methods, especially with limited measurement data.

**Strengths:**

1. The proposed method of using diffusion model for quantum state tomography is novel. The paper effectively adapts diffusion models to quantum systems by introducing a token embedding function that maps discrete measurement outcomes to continuous features, allowing seamless integration with the diffusion process.
2. The method seems to scale well to large systems. The model demonstrates strong performance on systems up to 100 qubits, which is significant for quantum computing applications.

**Weaknesses:**

1. While the paper mentions T=2000 denoising steps, there's no ablation study on how the number of steps affects performance versus computational cost.
2. The paper fixes the embedding dimension at d=128 without analyzing how different dimensions affect the model's performance and computational requirements.
3. The paper only considers Pauli-6 POVM, while evaluations on other POVM measurements may be beneficial.
4. Although the authors made reasonable efforts, it is unclear to non-physicists how wave functions become probability distributions. Adding a section in the appendix discussing POVM measurements is preferable.

**Questions:**

1. How does the choice of diffusion schedule (β_t values) affect the model's performance? Would adaptive scheduling improve results?
2. The paper uses Pauli-6 POVM measurements. How would the model perform with other measurement bases, and could this be made basis-independent?
3. Could the authors comment on if the model can be used to obtain the log probabilities of each measurement outcome?
4. POVM probabilities can be non-physical. Does the model always learn physical solutions?
5. Could the model be modified to predict other quantum properties beyond correlation and entanglement entropy? What architectural changes would be needed?

**Details Of Ethics Concerns:**

No concerns.

---

> ### Author Response · Authors · 2024-11-22
> **Response to Reviewer 9ozo  (Part 1)**
>
> Thank you for taking the time to provide detailed comments and valuable suggestions. We are happy to address your questions step by step. We have addressed the typos in the revised version of the PDF and added necessary clarifications along with additional numerical experiments in the Appendix. The changes made in the PDF are highlighted in blue for easy reference.
>
>
> **Q1: While the paper mentions T=2000 denoising steps, there's no ablation study on how the number of steps affects performance versus computational cost.**
>
> **A1:** We sincerely thank the reviewers for their valuable suggestions. The slow sampling speed is indeed a significant concern when it comes to diffusion models (which we have already mentioned in the conclusion and limitation section of the paper).
>
> To evaluate this, we fix the number of diffusion steps during training for QuaDiM while shrinking the inference steps $T_f$ using the approach introduced in DDIM (Song et al., 2020). We evaluate the model’s performance under different inference step settings and compare it with both a learning-free baseline (classical shadow, CS) and a learning-based SOTA model LLM4QPE. Due to time constraints, we only report results for the task of predicting correlations with $L = 100$, $M_{in} = 1000$, and $M_{out} = 1000$.
>
> | Method              | RMSE   | Generated samples per sec. |
> | ------------------- | ------ | -------------------------- |
> | CS                  | 0.0547 | -                          |
> | LLM4QPE             | 0.0531 | 14.6                       |
> | QuaDiM ($T_f=2000$) | 0.0478 | 5.7                        |
> | QuaDiM ($T_f=1000$) | 0.0537 | 8.1                        |
> | QuaDiM ($T_f=500$)  | 0.0541 | 12.7                       |
> | QuaDiM ($T_f=100$)  | 0.0882 | 37.4                       |
>
> As shown in the table above, when reducing inference to $T_f=500$ diffusion steps on a single GPU (2080Ti), QuaDiM achieves a lower RMSE score compared to the classical shadow baseline while demonstrating an inference speed comparable to LLM4QPE.
>
> We acknowledge that the sampling time of diffusion-based models poses a significant challenge when applied to quantum many-body problems (as we already discussed in the limitation and conlusion section of our paper). This limitation is inherent to the nature of diffusion-based approaches. In our future work, we seek to explore the incorporation of techniques such as consistency training, potentially enabling one-step sampling for diffusion-based applications in the domain of quantum many-body problems. We remain optimistic about these possibilities and appreciate the reviewers’ understanding of this ongoing effort.
>
> **Q2: The paper fixes the embedding dimension at d=128 without analyzing how different dimensions affect the model's performance and computational requirements.**
>
> **A2:** We sincerely thank the reviewer for raising this question. In all our experiments, we set the embedding hidden dimension to $d = 128$ (as well as other  Transformer parameters such as the number of heads to 4 and the number of layers to 4) primarily for two reasons: first, it follows the default settings used in prior work LLM4QPE proposed by Tang, et al, and second, based on empirical observations, $d = 128$ often strikes a balance between good generalization performance and avoiding excessive dimensionality.
>
> To further investigate, we evaluate the model's performance on the task of predicting correlations under a fixed dataset configuration $L = 10, M_{in} = 1000,M_{out} = 1000$ with different $d$ values from $\{64, 128, 256, 512\}$, and the resulting RMSE scores (lower is better) are: 0.0518, 0.0432, 0.0449, 0.0457, respectively. As the results show, setting $d = 128$ achieves the best performance, confirming its suitability in this context.
>
> We hope that these additional results provide more insight into the rationale behind our parameter choices. Should the reviewers have further concerns or suggestions, we would be delighted to address them in detail.

---

> ### Author Response · Authors · 2024-11-22
> **Response to Reviewer 9ozo (Part 2)**
>
> **Q3: Although the authors made reasonable efforts, it is unclear to non-physicists how wave functions become probability distributions. Adding a section in the appendix discussing POVM measurements is preferable.**
>
>
> **A3:** Thank you for your valuable comments. We will add the necessary definitions and notations related to quantum wave functions and POVM measurements to facilitate a clearer understanding for the reviewers. The additional content has already been incorporated into the appendix of the updated version of the paper.
>
> We aim to explain the fundamental concepts of quantum computing as clearly and step-by-step as possible using mathematical notions. We sincerely appreciate the reviewer's patience in taking the time to read through the following content. (For a comprehensive discussion, we refer the reviewers to the Section 2.1 of the book [1])
>
> Note that in Section 2 of the paper we have already provided a brief description of the fundamental concepts of quantum computing. We will further elaborate on the details below.
>
> A single qubit -- the smallest unit of quantum computing -- is mathematically represented as a vector $|\psi\rangle=\alpha|0\rangle+\beta|1\rangle$ parameterized by two complex numbers satisfying $|\alpha|^2+|\beta|^2=1$. Operations on a qubit must preserve this norm, and thus are described by $2 \times 2$ unitary matrices. Of these, some of the most important are the Pauli operators; it is useful to list them here:
> $$
>     X \equiv\left[\begin{array}{cc}
>     0 & 1 \\\\
>     1 & 0
>     \end{array}\right] , \quad Y \equiv\left[\begin{array}{cc}
>     0 & -i \\\\
>     i & 0
>     \end{array}\right] , \quad Z \equiv\left[\begin{array}{cc}
>     1 & 0 \\\\
>     0 & -1
>     \end{array}\right].
> $$ One could do some linear algebras and check that $|0\rangle = \big[\begin{smallmatrix}1\\\\0\end{smallmatrix}\big]$ and $|1\rangle = \big[\begin{smallmatrix}0\\\\1\end{smallmatrix}\big]$ are the eigenvectors of $Z$, $|+\rangle=\frac{1}{\sqrt{2}}\big[\begin{smallmatrix}1\\\\ 1\end{smallmatrix}\big]$ and $|-\rangle=\frac{1}{\sqrt{2}}\big[\begin{smallmatrix}1\\\\ -1\end{smallmatrix}\big]$ are the eigenvectors of $X$, $| \boldsymbol{i}\_{+}\rangle=\frac{1}{\sqrt{2}}\big[\begin{smallmatrix}1\\\\  \boldsymbol{i}\end{smallmatrix}\big]$ and $| \boldsymbol{i}\_{-}\rangle=\frac{1}{\sqrt{2}}\big[\begin{smallmatrix}1\\\\-\boldsymbol{i}\end{smallmatrix}\big]$ are the eigenvectors of $Y$. The same qubit can be decomposed in to different orthonormal basis. For example,
> $$
> \begin{aligned}
>         |\psi\rangle &= \alpha|0\rangle + \beta|1\rangle \\
>         &= \frac{1}{\sqrt{2}}(\alpha+\beta)|+\rangle + \frac{1}{\sqrt{2}}(\alpha-\beta)|-\rangle \\
>         &= \frac{1}{\sqrt{2}}(\alpha-\beta\boldsymbol{i})|\boldsymbol{i}\_{+}\rangle + \frac{1}{\sqrt{2}}(\alpha+\beta\boldsymbol{i})|\boldsymbol{i}\_{-}\rangle.
> \end{aligned}
> $$ Positive-operator valued measurement (POVM) is the testing or manipulation of a physical system to yield a numerical result. POVM is described by a set of measurement operators $\\{\mathbf{\Pi}\_k\\}\_{k=0}^{K-1}$ satisfying $\sum_{k} \mathbf{\Pi}\_k = \mathbf{I}$ and each $\mathbf{\Pi}\_k$ is positive semi-definite, where $K$ is the total number of measurement operators. In this paper, we consider the Pauli-6 POVM (also named as randomized single-qubit Pauli measurements in some literature) such that the measurement operators are $\\{\frac{1}{3}|0\rangle\langle 0|, \frac{1}{3}|1\rangle\langle 1|,\frac{1}{3}|+\rangle\langle +|, \frac{1}{3}|-\rangle\langle -|, \frac{1}{3}|\boldsymbol{i}\_+\rangle\langle \boldsymbol{i}\_+|, \frac{1}{3}|\boldsymbol{i}\_-\rangle\langle \boldsymbol{i}\_-|\\}$. It is easy to check that these operators satisfy the POVM definition and $K=6$. The reason for choosing the **Pauli-6 POVM is that this measurement protocol is easy to be implemented on current quantum devices (NISQ devices)** and is **informative-completed (IC)** (it means that all the information of the quantum state could be recovered classically by a sufficient large number of IC-POVM measurements. In other words, given the probability of each of the measurement outcomes of IC-POVM, we can uniquely determine the quantum state).
>
> Measuring a qubit leads to collapse of the qubit and produces an outcome $k$ with the probability $p(k)$ satisfying the Born rule, which states that $p(k) = \mathrm{tr}(\rho \mathbf{\Pi}_k)$, where $\rho=|\psi\rangle\langle\psi|$ and $\langle\psi|$ is the transpose conjugate of $|\psi\rangle$.

---

> ### Author Response · Authors · 2024-11-22
> **Response to Reviewer 9ozo (Part 3)**
>
> We may consider a system of $L$ qubits. It can be described by the **wave function**:
> $$
>     |\mathbf{\Phi}\rangle = \sum_{\sigma_1=1}^{M}\cdots\sum_{\sigma_L=1}^{M} \mathbf{\Psi}(\sigma_1,\ldots,\sigma_L)|\sigma_1,\ldots,\sigma_L\rangle,
> $$ where $\mathbf{\Psi}: \mathbb{Z}^{L}\rightarrow \mathbb{C}$ maps a fixed configuration $\sigma=(\sigma_1,\ldots,\sigma_L)$ of $L$ qubits to a complex number which is the amplitude satisfying $\sum_{\sigma_1=1}^{K}\cdots\sum_{\sigma_L=1}^{K} |\mathbf{\Psi}(\sigma_1,\ldots,\sigma_L)|^2=1$, and $\sigma_i \in \{1,\ldots,K\}$ is one of the $K$ possible outcomes by performing quantum measurement on the $i$-th qubit. It is formulated in a complex Hilbert space where the vector representation of the quantum state $|\mathbf{\Phi}\rangle \in \mathbb{C}^{K^L}$ and its density matrix $|\mathbf{\Phi}\rangle\langle\mathbf{\Phi}|\in \mathbb{C}^{K^L\times K^L}$, which becomes astronomical for large $L$.
>
> Performing quantum measurement independently on $L$ qubits is easy to be implemented. The most common strategy is to combine $L$ single-qubit measurement operators to $\mathbf{\Pi}\_{k,1}\otimes\cdots\otimes\mathbf{\Pi}\_{k,L}$ where $\otimes$ is the Kronecker product.  Such measurement procedure outputs a measurement string $\sigma=(\sigma_1,\ldots,\sigma_L)$ where $\sigma_i \in \{1,\ldots,K\}$ with probability $|\mathbf{\Psi}(\sigma_1,\ldots,\sigma_L)|^2$.
>
> Define $p(\sigma_1,\ldots,\sigma_L)=|\mathbf{\Psi}(\sigma_1,\ldots,\sigma_L)|^2$. We can reformulate the wave function of quantum states to a **classical joint distribution**. It is a valid and legal joint distribution since $\sum_{\sigma_1}\cdots\sum_{\sigma_L}p(\sigma_1,\ldots,\sigma_L)=\sum_{\sigma_1}\cdots\sum_{\sigma_L}|\mathbf{\Psi}(\sigma_1,\ldots,\sigma_L)|^2=1$ and $p(\sigma_1,\ldots,\sigma_L)\geq 0$.
>
>
> Thus, we connect the quantum wave function with a classical joint distribution and attempt to approximate this joint distribution using neural network methods, including autoregressive approaches [2,3,4] and our proposed non-autoregressive method QuaDiM.
>
> If there are any further unclear aspects in our explanation, we sincerely hope the reviewer can point them out, and we will do our best to provide clarifications.
>
> [1] Nielsen M A, Chuang I L. Quantum computation and quantum information. Cambridge university press, 2010.
>
> [2] Carrasquilla J, Torlai G, Melko R G, et al. Reconstructing quantum states with generative models[J]. Nature Machine Intelligence, 2019.
>
> [3] Tang Y, Xiong H, Yang N, et al. Towards LLM4QPE: Unsupervised Pretraining of Quantum Property Estimation and A Benchmark. The Twelfth International Conference on Learning Representations (ICLR), 2024.
>
> [4] Xiao T, Huang J, Li H, et al. Intelligent certification for quantum simulators via machine learning[J]. npj Quantum Information, 2022.
>
> **Q4: The paper only considers Pauli-6 POVM, while evaluations on other POVM measurements may be beneficial.**
>
> **A4:** Thank you for your comments. We emphasize the use of Pauli-6 POVM measurements for data collection because **this measurement protocol is easy to implement on current quantum devices (NISQ devices)** and is **informationally complete (IC)**. This means that all the information of the quantum state can be recovered classically with a sufficiently large number of IC-POVM measurements. In other words, given the probability of each measurement outcome of IC-POVM, the quantum state can be uniquely determined.
>
> To further validate our method, here we consider another type of IC-POVM: the **tetrahedral POVM** [1], to collect measurement data. The corresponding measurement operators are $\{\frac{1}{4}(\mathbf{I}+\mathbf{s}^{(a)}\cdot \mathbf{P})\}\_{a\in\\{0,1,2,3\\}}$, where $\mathbf{I}$ is the identity matrix, $\mathbf{P}$ represents the ensemble of Pauli operators $(X, Y, Z)$ and $\mathbf{s}^{(0)}=(0,0,1)$,$\mathbf{s}^{(1)}=(\frac{2\sqrt{2}}{3},0,-\frac{1}{3})$,$\mathbf{s}^{(2)}=(-\frac{\sqrt{2}}{3},\sqrt{\frac{2}{3}},-\frac{1}{3})$,$\mathbf{s}^{(3)}=(-\frac{\sqrt{2}}{3},-\sqrt{\frac{2}{3}},-\frac{1}{3})$. It is easy to check that $K=4$ for the tetrahedral POVM. Due to time constraints, we fixed $L=10$ with $M_{in}=1000$, re-run the simulations to collect data, and re-train our model and the baselines for predicting the correlations. The numerical results are reported as follows:
>
> | Method  | M=1000 | M=10000 |
> |---------|--------|---------|
> | CS      | 0.0512 | 0.0164  |
> | RBFK    | 0.0735 | -       |
> | NTK     | 0.0747 | -       |
> | RNN     | 0.0514 | 0.0163  |
> | LLM4QPE | 0.0503 | 0.0141  |
> | QuaDiM  | 0.0433 | 0.0107  |

---

> ### Author Response · Authors · 2024-11-22
> **Response to Reviewer 9ozo (Part 4)**
>
> As shown, QuaDiM still outperforms the baselines in this scenario. We hope this clarification addresses the reviewers’ concerns. Thank you again for the valuable feedback, and we look forward to further improving our work based on your comments.
>
> [1] Renes J M, Blume-Kohout R, Scott A J, et al. Symmetric informationally complete quantum measurements[J]. Journal of Mathematical Physics, 2004, 45(6): 2171-2180.
>
>
>
> **Q5: How does the choice of diffusion schedule (β_t values) affect the model's performance? Would adaptive scheduling improve results?**
>
> **A5:** We sincerely thank the reviewers for the feedback. In our experiments, we follow the implementation in [1,2] and adopt the square-root noise schedule for the noise coefficient. That is $\overline{\alpha}_t = 1-\sqrt{t/T + c}$, where $\overline{\alpha}\_t = \prod\_{s=0}^{t}(1-\beta\_s)$ and $c$ is a small constant and we set $c=0.0001$. We have clarified and explicitly stated this in the updated version of the paper.
>
> [1] Ho J, Jain A, Abbeel P. Denoising diffusion probabilistic models. Advances in neural information processing systems, 2020.
>
> [2] Li X, Thickstun J, Gulrajani I, et al. Diffusion-lm improves controllable text generation. Advances in Neural Information Processing Systems, 2022.
>
> **Q6: The paper uses Pauli-6 POVM measurements. How would the model perform with other measurement bases, and could this be made basis-independent?**
>
> **A6:** Thank you for your valuable question. In our response to A4, we have included results based on tetrahedral POVM. Regarding your inquiry about whether our current method can be made basis-independent, we admit that our proposed model has not yet achieved this.
>
> This limitation arises because different measurement protocols typically imply the use of different basis. As noted in our response in A3, the decomposition of the same quantum state generally differs depending on the chosen basis. Consequently, the coefficients $\Psi$ of the wave function $|\Phi\rangle$, as well as the form of the corresponding classical joint distribution $p$, are also different. This means that switching to a different IC-POVM would require retraining the model.
>
> By the way, we have noticed some recent pioneering research [1,2] exploring adaptive measurement strategies to find optimal measurement basis for specific quantum systems. These works may provide inspiration for our future efforts to make the model basis-independent. We remain open to further discussion with the reviewer on this intriguing new direction.
>
> [1] García-Pérez G, Rossi M A C, Sokolov B, et al. Learning to measure: Adaptive informationally complete generalized measurements for quantum algorithms[J]. PRX quantum, 2021, 2(4): 040342.
>
> [2] Glos A, Nykänen A, Borrelli E M, et al. Adaptive POVM implementations and measurement error mitigation strategies for near-term quantum devices[J]. arXiv preprint arXiv:2208.07817, 2022.
>
> **Q7: Could the authors comment on if the model can be used to obtain the log probabilities of each measurement outcome?**
>
> **A7:** Thank you for your comments. We would like to emphasize that the focus of this paper is to predict certain properties of quantum systems by training on measurement data and physical parameters. On the other hand, obtaining the log probabilities of each measurement outcome typically corresponds to quantum state tomography [1]. It is orthogonal to our research, aiming to reconstruct the full density matrix of the quantum state. This can be achieved through approaches such as supervised learning [2]—where the model is provided with paired data in the form of (measurement outcome, probability)—or optimization-based methods, such as variational Monte Carlo [3,4].
>
> In our preliminary and immature view, to adapt to quantum state tomography tasks, the proposed model architecture might not require modification. However, the data collection process and the loss function might need to be adjusted based on the specific task. For instance, under a supervised learning paradigm, training data might need to be collected in the form of (measurement outcome, probability), and the loss function could be replaced with a supervised loss function, such as Mean Squared Error (MSE) loss.
>
> We hope that the novel techniques proposed in our paper could inspire researchers in this field to explore a broader range of quantum state learning tasks in the future.
>
> [1] Torlai G, Mazzola G, Carrasquilla J, et al. Neural-network quantum state tomography[J]. Nature physics, 2018, 14(5): 447-450.
>
> [2] Zhu Y, Wu Y D, Bai G, et al. Flexible learning of quantum states with generative query neural networks[J]. Nature communications, 2022, 13(1): 6222.
>
> [3] Chen Z, Newhouse L, Chen E, et al. Antn: Bridging autoregressive neural networks and tensor networks for quantum many-body simulation[J]. Advances in Neural Information Processing Systems, 2023, 36: 450-476.

---

> ### Author Response · Authors · 2024-11-22
> **Response to Reviewer 9ozo (Part 5)**
>
> [4] Sehayek D, Golubeva A, Albergo M S, et al. Learnability scaling of quantum states: Restricted Boltzmann machines[J]. Physical Review B, 2019, 100(19): 195125.
>
>
> **Q8: POVM probabilities can be non-physical. Does the model always learn physical solutions?**
>
> **A8:** Thank you for your insightful question. In this work, we chose the **physically implementable** and **informationally complete** Pauli-6 POVM [1]. In our response to A4, we further supplemented the results  using the tetrahedral POVM, which is also informationally complete but more challenging to implement physically. The results demonstrate that our model outperforms the baselines under both measurement protocols.
>
> Since both POVMs in our experiments are informationally complete, theoretically, a sufficient amount of measurement records (meaning the exact probabilities of all possible measurement outcomes are available) uniquely determines the measured quantum state. This ensures that adequate measurement data enable the model to learn physical patterns. Consequently, our numerical experiments did not exhibit non-physical behavior, and the model always learned physical solutions.
>
> **We reasonably speculate that the reviewer's concern may relate to using informationally incomplete POVMs (e.g., measuring only with Pauli Z) for data collection and model training**, which might result in the model failing to capture the complete physical patterns. This negative impact is indeed possible and would require sophisticated neural network design and the incorporation of physical priors to achieve comparable performance to informationally complete POVMs, as studied in [2]. However, exploring the use of informationally incomplete POVMs is beyond the scope of this paper, and we are happy to investigate this in future research.
>
> [1] Huang H Y, Kueng R, Preskill J. Predicting many properties of a quantum system from very few measurements[J]. Nature Physics, 2020, 16(10): 1050-1057.
>
> [2] Koutný D, Ginés L, Moczała-Dusanowska M, et al. Deep learning of quantum entanglement from incomplete measurements[J]. Science Advances, 2023, 9(29).
>
>
> **Q9: Could the model be modified to predict other quantum properties beyond correlation and entanglement entropy? What architectural changes would be needed?**
>
> **A9:** Thank you for your insightful comments. In this paper, we focus primarily on two specific tasks: predicting correlation and entanglement entropy. These tasks are chosen because they are well-studied in the literature and have relatively established benchmarks [1,2].
>
> Broadly speaking, beyond these two tasks, our model is capable of predicting other properties of quantum states through quantum measurements. For example, it can estimate **linear functions of quantum states**, such as symmetry-breaking phases and expectation values of local observables, as well as **non-linear functions of quantum states**, such as purity and the energy variance of local Hamiltonians.
>
> We also anticipate that the model could be further adapted to enhance its representational capacity to capture more complex patterns of quantum states and even quantum processes. For instance, future work could involve designing new embedding layers that incorporate information about the observables used in quantum measurements as additional features. Moreover, new layers and multimodal approaches could be involved to input information of quantum process into the model by representing them as graph topologies. These adjustments would enable the model to better understand and leverage the intricate structure of quantum systems.
>
> In future work, we aim to extend our research to explore additional properties of quantum states and further evaluate the versatility of our model across diverse quantum systems. Thank you again for highlighting this direction, which we believe will be a valuable avenue for future exploration.
>
> [1] Huang H Y, Kueng R, Preskill J. Predicting many properties of a quantum system from very few measurements[J]. Nature Physics, 2020, 16(10): 1050-1057.
>
> [2] Tang Y, Xiong H, Yang N, et al. Towards LLM4QPE: Unsupervised Pretraining of Quantum Property Estimation and A Benchmark. The Twelfth International Conference on Learning Representations (ICLR), 2024.

---

> ### Author Response · Authors · 2024-11-25
> **Sincerely Awaiting Your Feedback**
>
> Dear Reviewer 9ozo,
>
> I hope this message finds you well. As the rebuttal deadline is approaching, we would greatly appreciate your valuable feedback at your earliest convenience. Your insights are highly important to us, and we are eager to address your comments thoroughly.
>
> Best regards

---

> ### Comment · Reviewer_9ozo · 2024-11-27
>
> Thanks for the detailed rebuttal. I would like to keep recommending acceptance of the paper. In the meantime, I may consider raising the score before the deadline.

---

> ### Author Response · Authors · 2024-11-27
>
> Dear Reviewer 9ozo,
>
> Thank you very much for your thoughtful and constructive feedback, and for considering the possibility of raising the score. We greatly appreciate your support and are encouraged by your positive comments regarding our work.
>
> We fully understand that your final decision is contingent on the review process, and we deeply value the careful consideration you’ve given our paper. If there are any further aspects that may require clarification or additional discussion, please don’t hesitate to let us know. **We are more than happy to provide any additional information that might help in your final assessment.**
>
> Once again, thank you for your time and effort in reviewing our submission. Your input is invaluable, and we are truly grateful for your support.
>
> Best regards
>
> On behalf of all authors

---

### Official Review · Reviewer_oe1d · 2024-11-04

**Soundness:** 3
**Presentation:** 3
**Contribution:** 3
**Rating:** 6
**Confidence:** 4

**Summary:**

The paper proposes QuaDiM, a new framework based on diffusion models that learns the ground state of quantum systems conditioned on system parameters and enables quantum property estimation. QuaDiM uses a transformer-based architecture, where at inference time, measurement outcomes are first generated as continuous-valued hidden vectors and then decoded into discrete values. Numerical experiments are performed on 1D anti-ferromagnetic Heisenberg model up to $100$ qubits and predicted properties include correlation and entanglement entropy.

**Strengths:**

* The paper is very readable, with clear and understandable notations and easy-to-follow reasoning across the paper.
* The proposed methods achieve SOTA performance on predicting correlation and entanglement entropy on 1D anti-ferromagnetic Heisenberg model with a reasonably large system size, justifying the effectiveness of the proposed approach.

**Weaknesses:**

* While the motivation for using non-autoregressive models is briefly discussed in the introduction, there could be more discussion and empirical evidence to support the claims about the sub-optimal performance of auto-regressive methods. While the measurements are not like language, the underlying physics model can still have some order structure (like a chain) and locality information is often important. Therefore, it's not completely obvious why diffusion models are necessarily better.
* QuaDiM lacks machine learning technical novelty. The conditional diffusion model are already well studied in the literature and the used score neural network seems to be standard transformer as well. Given that ICLR is a ML conference, it would be more interesting to see applications and inspiration that can be drawn from QuaDiM to more general problems.
* The experimental setup are not clearly presented. For example, the generation protocol of system parameter $x \in \mathbb{R}^{L - 1}$ is not described in the experiments section, not to mention the training and test set splitting process. This makes it hard to understand under what level QuaDiM extrapolates to "unseen parameters".
* The experiment design is relatively weak. It would be interesting to consider more complicated property estimation problems, such as the estimation of entanglement negativity, two site observables with long ranges in between, or even the estimation of system parameters in the Hamiltonian. Second, it would be more convincing if QuaDiM is examined on a more complicated physics model, such as some with non-trivial long-range entanglement. It would be more convincing if the extrapolation of QuaDiM to critical state of the physics model could be tested and examined.
* The benchmarks are not representative enough. For example, [1-3] are also works that model quantum states that should enable property estimation as downstream tasks, but are not discussed or compared.

References:

[1] Fitzek, David, et al. "RydbergGPT." arXiv preprint arXiv:2405.21052 (2024).

[2] Du, Yuxuan, et al. "Shadownet for data-centric quantum system learning." arXiv preprint arXiv:2308.11290 (2023).

[3] Zhang, Yuan-Hang, and Massimiliano Di Ventra. "Transformer quantum state: A multipurpose model for quantum many-body problems." Physical Review B 107.7 (2023): 075147.

**Questions:**

* Can you discuss more on in what cases diffusion models would be strictly better than auto-regressive models, in the quantum state modeling setting?
* I am not completely clear on how continuous latent vectors generated by QuaDiM are decoded back into measurement outcomes. Can you elaborate more on this?
* Why QuaDiM will generate measurements consistently sampled from a valid quantum state without violating structure constraints. How is this enforced in the learning algorithm?
* Does QuaDiM extrapolate to unseen system parameter $x$ distribution, likely with disjoint support as the training distribution, e.g., trained on $x \in [0, 0.8]$, tested on $x \in [0.9, 1]$?
* Can you comment on QuaDiM's training efficiency? First, it seems that a relatively large $M_{in}$ is required for learning one ground state to good accuracy, also $M_{out}$ is not a small number as well. Can $M_{in}$ and $M_{out}$ be improved? Second, how many distinct pairs of (ground state, measurements) are needed for learning the mapping between Hamiltonian parameter and system ground state correctly, and how does this scale with the number of qubits?

---

> ### Author Response · Authors · 2024-11-22
> **Response to Reviewer oe1d (Part 1)**
>
> Thank you for the time, thorough comments, and nice suggestions. We are pleased to clarify your questions step-by-step. We have addressed the typos in the revised version of the PDF and added necessary clarifications along with additional numerical experiments in the Appendix. The changes made in the PDF are highlighted in blue for easy reference.
>
> **Q1: While the motivation for using non-autoregressive models is briefly discussed in the introduction, there could be more discussion and empirical evidence to support the claims about the sub-optimal performance of auto-regressive methods. While the measurements are not like language, the underlying physics model can still have some order structure (like a chain) and locality information is often important. Therefore, it's not completely obvious why diffusion models are necessarily better.**
>
> **A1:** Thank you for the insightful questions and constructive feedback. To address your concerns more effectively, we would like to begin by briefly restating the motivation behind our work.
>
> Many advanced studies [1,2] have applied autoregressive machine learning models (e.g., RNNs and transformers) to quantum state modeling, with applications including reconstructing quantum states’ density matrices, and predicting certain nonlinear properties of quantum states (the focus of this paper) such as correlations and entanglement entropies.
>
> Intuitively, these methods treat the discrete data obtained from quantum measurements as analogous to a corpus in language modeling, where each qubit’s post-measurement collapsed result corresponds to a token. Consequently, deep learning models commonly used in natural language processing (e.g., RNNs and transformers) are leveraged to model quantum states (or a family of quantum states).
>
> However, **recent studies [3,4] have found that the inherent constraints of autoregressive methods**—specifically, the pre-defined factorization order (e.g., left-to-right or right-to-left order in 1D chains, or zig-zag orders in 2D systems)—often introduce **biases** into the model. This bias **influences** the ability of variational models including deep learning models to accurately describe correlations among qubits and **constrains** the expressivity of these models in learning quantum systems.
>
> This observation **aligns with** the experimental phenomena we observed. Specifically, for 1D chain-like Heisenberg systems, the performance of traditional autoregressive methods (trained and sampled under a pre-defined ordering) is often **inferior** to non-autoregressive diffusion models, when it comes to predicting certain quantum state properties such as correlations and entropies using limited measurement data (as listed in Table 1 and 2).
>
> Motivated by these findings, we propose the use of **non-autoregressive diffusion models** as an alternative for modeling quantum states. The idea to employ diffusion models as the backbone is based on their SOTA performance across various tasks, including image, video generation, and language modeling. In the subfield of quantum state modeling, **diffusion models share similarities with autoregressive models while also exhibiting unique characteristics**:
>
> The **similarity** is that both approaches aim to approximate the classical joint-probability distribution $p(\sigma_1,\ldots,\sigma_L)$ corresponding to the wave function $|\psi\rangle=\sum_{\sigma_1=1}^{K}\cdots\sum_{\sigma_L=1}^{K} \Phi(\sigma_1,\ldots,\sigma_L)|\sigma_1,\ldots,\sigma_L\rangle$ of a (family of) quantum state during training, written as
> $$p(\sigma_1,\ldots,\sigma_L) = ||\Phi(\sigma_1,\ldots,\sigma_L)||^2$$
> where $\Phi(\sigma_1,\ldots,\sigma_L)$ is the amplitude of the basis $(\sigma_1,\ldots,\sigma_L)$. According to Born rule in quantum mechanics, the joint distribution $p$ is a valid probability distribution since $p(\sigma_1,\ldots,\sigma_L)\geq 0$ and $\sum_{\sigma_1}\cdots\sum_{\sigma_L}p(\sigma_1,\ldots,\sigma_L)=1$.
>
>
> The key **difference** is as follows. The essence of any autoregressive model is the application of the product rule of probability to **factorize a joint-probability distribution of random variables (qubits) into a causal product of conditional probability distributions, one for each variable conditioned on the realizations of preceding variables**, which is given by $p(\sigma_1,\ldots,\sigma_L) = \prod_{i=1}^{L} p(\sigma_i|\sigma_{<i})$. However, this factorization requires **manually defining a fixed order** of the qubits (e.g., left-to-right or right-to-left in 1D chains) prior to training. During inference, the model also needs to **generate or sample one qubit at a time** following the pre-defined order.

---

> ### Author Response · Authors · 2024-11-22
> **Response to Reviewer oe1d (Part 2)**
>
> In contrast, non-autoregressive diffusion models encourage to **avoid** this manually defined ordering bias in both training and sampling phases. They **directly model the joint probability distribution** corresponding to the quantum state and sample from it, i.e., the sampled configurations of qubits $(\sigma_1,\ldots,\sigma_L)$ are genetated **simultaneously**.  This lack of bias offers hope for better predicting quantum state correlations and entropies while reducing the required number of training and decoding samples.
>
> To **further** compare the our model and the autoregressive baselines, we define an alternative ordering of qubits (from right to left or from the largest to the smallest index, as opposed to the left-to-right ordering used throughout the paper) and re-train the autoregressive baselines. The results of RMSE for predicting the correlations of all subsystems of size two on the test dataset, evaluated in the predefined order from right to left, are presented below. For comparison, we also include experimental results already reported in the paper. Due to time constraints, we only report results for $L=70$ and $L=100$ for various sample counts. Note that <- denotes from right to left, and -> denotes from left to right.
>
> | Method      | L=70   |        |        |        | L=100  |        |        |        |
> |-------------|--------|--------|--------|--------|--------|--------|--------|--------|
> |             | M=100  | 1000   | 10000  | 20000  | M=100  | 1000   | 10000  | 20000  |
> | RNN(<-)     | 0.2197 | 0.0721 | 0.0216 | 0.0165 | 0.2276 | 0.0763 | 0.0264 | 0.0159 |
> | RNN(->)     | 0.2137 | 0.0739 | 0.0240 | 0.0153 | 0.2325 | 0.0806 | 0.0251 | 0.0163 |
> | LLM4QPE(<-) | 0.1865 | 0.0538 | 0.0157 | 0.0108 | 0.1773 | 0.0542 | 0.0149 | 0.0122 |
> | LLM4QPE(->) | 0.1814 | 0.0527 | 0.0155 | 0.0116 | 0.1759 | 0.0531 | 0.0152 | 0.0114 |
> | QuaDiM      | 0.1679 | 0.0473 | 0.0117 | 0.0092 | 0.1686 | 0.0478 | 0.0125 | 0.0098 |
>
>
> As can be observed, the prediction performance of autoregressive baselines is **inferior** to that of the diffusion models **regardless of whether the predefined order is from left to right or from right to left**. This observation aligns with the findings in [1,2], which highlight that the predefined factorization order imposes constraints on the expressivity of these models when learning quantum systems.
>
> We hope this clarification addresses the reviewers' concerns. Thank you again for the valuable feedback, and we look forward to further improving our work based on your comments.
>
> [1] Gebhart V, Santagati R, Gentile A A, et al. Learning quantum systems[J]. Nature Reviews Physics, 2023, 5(3): 141-156.
>
> [2] Carrasquilla J. Machine learning for quantum matter[J]. Advances in Physics: X, 2020, 5(1).
>
> [3] Bortone M, Rath Y, Booth G H. Impact of conditional modelling for a universal autoregressive quantum state[J]. Quantum, 2024, 8: 1245.
>
> [4] Ibarra-García-Padilla E, Lange H, Melko R G, et al. Autoregressive neural quantum states of Fermi Hubbard models[J]. arXiv preprint arXiv:2411.07144, 2024.
>
> **Q2: QuaDiM lacks machine learning technical novelty. The conditional diffusion model are already well studied in the literature and the used score neural network seems to be standard transformer as well. Given that ICLR is a ML conference, it would be more interesting to see applications and inspiration that can be drawn from QuaDiM to more general problems.**
>
> **A2:** We sincerely thank the reviewer for the feedback. We would like to highlight that leaning quantum states hold critical importance for both the quantum computing and deep learning communities [1,2].
>
> Firstly, we emphasize that QuaDiM is proposed to address the challenges of **quantum state property estimation (QPE)**, a foundational problem in quantum many-body problems[1]. In recent years, many cutting-edge works have been published in AI conferences, focusing on both the adaptation and application of deep learning methods such as autoregressive models [3], pretraining strategies [4], and deep equilibrium models [5]. Beyond these, a series of contributions have emerged, arising from both experimental [6,7,8] and theoretical [9,10,11] perspective. We aspire for this paper to serve as a source of inspiration for researchers across both the AI and quantum computing domains.

---

> ### Author Response · Authors · 2024-11-22
> **Response to Reviewer oe1d (Part 3)**
>
> Note that while diffusion models and transformers are well-established in the ML community, their adaptation to the quantum many-body domain is **non-trivial**. Unlike typical applications like language modeling, quantum measurement data involve **non-sequential, exponentially large dimensions** with complex entanglement [9]. QuaDiM encourages to alleviate these issues by introducing a **non-autoregressive generative framework** for QPE. Here, we would like to briefly reiterate our contributions and innovations:
> - **Non-Autoregressive Diffusion-Based Approach:** QuaDiM introduces the **first non-autoregressive conditional generative model** specifically designed for QPE. Unlike standard approaches that rely on sequential dependencies (e.g., auto-regressive models), QuaDiM employs **diffusion models** to iteratively denoise Gaussian noise into the target quantum state distribution, with the hope of encouraging **equal treatment of all qubits** and removing bias introduced by sequential modeling. This framework is novel in the QPE context, addressing the limitations of existing machine learning models that fail to capture non-sequential, high-dimensional entanglement structures.
> - **Scalability to Large Systems:** Our empirical evaluation extends to quantum systems up to **100 qubits** (up to $2^{100}$ dimension). This involved a large-scale data collection process from classical simulation. Notably, the simulation and data collection process utilizing the Matrix Product State (MPS) algorithm required nearly two weeks of computation on a cluster equipped with 4 Intel Xeon Gold 6248 CPUs (total cores: 80, total threads: 160). Notably, with reduced sample complexity, our model outperforms state-of-the-art baselines.
>
> We again sincerely appreciate the reviewers’ thoughtful feedback. We hope that this clarification will aid in further understanding and help to underscore QuaDiM's potential as a valuable contribution to the field.
>
> [1] Gebhart V, Santagati R, Gentile A A, et al. Learning quantum systems[J]. Nature Reviews Physics, 2023, 5(3): 141-156.
>
> [2] Carrasquilla J. Machine learning for quantum matter[J]. Advances in Physics: X, 2020, 5(1).
>
> [3] Chen Z, Newhouse L, Chen E, et al. Antn: Bridging autoregressive neural networks and tensor networks for quantum many-body simulation. NeurIPS, 2023.
>
> [4] Tang Y, Xiong H, Yang N, et al. Towards LLM4QPE: Unsupervised Pretraining of Quantum Property Estimation and A Benchmark. ICLR, 2024.
>
> [5] Wang Z, Liu C, Zou N, et al. Infusing Self-Consistency into Density Functional Theory Hamiltonian Prediction via Deep Equilibrium Models. NeurIPS, 2024.
>
> [6] Carrasquilla J, Torlai G, Melko R G, et al. Reconstructing quantum states with generative models[J]. Nature Machine Intelligence, 2019.
>
> [7] Xiao T, Huang J, Li H, et al. Intelligent certification for quantum simulators via machine learning[J]. npj Quantum Information, 2022.
>
> [8] García-Pérez G, Rossi M A C, Sokolov B, et al. Learning to measure: Adaptive informationally complete generalized measurements for quantum algorithms[J]. PRX quantum, 2021.
>
> [9] Huang H Y, Kueng R, Torlai G, et al. Provably efficient machine learning for quantum many-body problems[J]. Science, 2022.
>
> [10] Lewis L, Huang H Y, Tran V T, et al. Improved machine learning algorithm for predicting ground state properties[J]. Nature Communications, 2024.
>
> [11] Huang H Y, Broughton M, Cotler J, et al. Quantum advantage in learning from experiments[J]. Science, 2022.

---

> ### Author Response · Authors · 2024-11-22
> **Response to Reviewer oe1d (Part 4)**
>
> **Q3: The experimental setup are not clearly presented. For example, the generation protocol of system parameter $\mathbf{x}\in\mathbb{R}^{L-1}$ is not described in the experiments section, not to mention the training and test set splitting process. This makes it hard to understand under what level QuaDiM extrapolates to "unseen parameters".**
>
> **A3:** We thank the reviewer for their insightful question. In Section 4.1, we briefly described the meaning of the system parameter $\mathbf{x} \in \mathbb{R}^{L-1}$. Below, we provide additional details on how $\mathbf{x}$ is generated in our simulation experiments, as well as the rules used to partition the training and test datasets. These clarifications have also been added to the revised version of the paper.
>
> First, we would like to emphasize that, unlike neural network quantum states (e.g., [3, 4]), where the neural network is trained to approximate the distribution of a single quantum state and aims to reconstruct the state's density matrix, our method **learns classical representations of a family of quantum states**. This approach reduces measurement costs and even allows us to study the properties of states for which no measurement data are available, including states that cannot currently be prepared on modern NISQ hardware.
>
> The parameter $\mathbf{x}$ is a sequence of real numbers representing the coupling strengths among qubits. These values serve as parameters for the Hamiltonian $H(\mathbf{x})$ (Eq.8 in the paper) and is given as
> $$
> H(\mathbf{x}) = \sum_i x_{i} \left(X_i X_{i+1} + Y_i Y_{i+1} + Z_i Z_{i+1}\right),
> $$ where different values of $\mathbf{x}$ define different dynamics of quantum systems. Each of the $L-1$ elements in $\mathbf{x}$ is uniformly sampled from the range $[0, 2]$.
>
> In our study, we perform simulated experiments on the one-dimensional anti-ferromagnetic Heisenberg model, scaling up to 100 qubits. Direct simulation of such a large-qubit quantum system, involving manipulation and storage of maximal $2^{100}$-dimensional complex vectors, is impractical due to computational constraints. To overcome this, we utilize Matrix Product State (MPS) methods. Specifically, we leverage the Julia implementation provided by ITensor Library and the DMRG algorithm in it, to calculate the ground state. Some of the code used to generate the system parameters and the ground state is provided below to help the reviewer better understand the process.
>
>
> ```
> for i = 1:samples
>     # initialize the simulation
>     J = 2 * rand(N-1)
>     sites = siteinds("S=1/2", N)
>
>     # build the Hamiltonian
>     os = OpSum()
>
>     for j = 1:N-1
>         os += J[j], "X",j,"X",j+1
>         os += J[j], "Y",j,"Y",j+1
>         os += J[j], "Z",j,"Z",j+1
>     end
>
>     H = MPO(os, sites)
>     psi0 = productMPS(sites, n -> isodd(n) ? "Up" : "Dn")
>
>     nsweeps = 5
>     maxdim = [10,20,100,100,200]
>     cutoff = [1E-12]
>
>     # run the DMRG algorithm
>     energy, psi = dmrg(H,psi0; nsweeps, maxdim, cutoff)
>
>     # generate the random Pauli6 POVM basis used for measurement (X:0, Y:1, Z:2)
>     random_basis_matrix = rand(0:2, NM, N)
>     # generate the samples with randomized measurement
>     storage_samples = sample_povm(deepcopy(psi), NM, N, sites, deepcopy(random_basis_matrix))
>     # compute the corrlation matrix
>     corr_matrix_zz = correlation_matrix(deepcopy(psi),"Z","Z")
>     corr_matrix_yy = correlation_matrix(deepcopy(psi),"Y","Y")
>     corr_matrix_xx = correlation_matrix(deepcopy(psi),"X","X")
>     corr_matrix = (corr_matrix_zz .+ corr_matrix_yy .+ corr_matrix_xx) ./ 3.0
>     # compute the entropy
>     entropy = calcuate_entropy_all(deepcopy(psi), N)
> ```
>
> Once the ground state is obtained, we perform measurement using Pauli-6 POVM measurements to produce the corresponding measurement record $R\in\mathbb{Z}^{L\times M_{in}}$. The pairs $(\mathbf{x}, R)$ are then used as inputs for model training, while the labels are derived via exact diagonalization (for $L \leq 10$) or classical shadow methods (for $L > 10$).
>
> The process described above is used for generating both training and test dataset samples. The difference lies in the test dataset, where measurement records $R$ are no longer required. For the training dataset, we pre-sample $N^{tr} = 100$ different $\mathbf{x}$ values, while the test dataset includes $N^{te} = 20$ samples. The training dataset and test dataset are denoted as $\mathcal{D}^{tr}$ and $\mathcal{D}^{te}$, respectively.
>
> During training, the model uses the training dataset’s $(\mathbf{x}, R)\in \mathcal{D}^{tr}$ pairs as inputs. At inference time (testing), the trained model generalizes to the Hamiltonians' ground states corresponding to previously "unseen" system parameters $\mathbf{x}\in \mathcal{D}^{te}$. Specifically, given a new input $x \in \mathcal{D}^{te}$, the model generates the measurement data and predicts properties of the quantum system, such as correlation and entropy, through post-processing.

---

> ### Author Response · Authors · 2024-11-22
> **Response to Reviewer oe1d (Part 5)**
>
> **Q4: The experiment design is relatively weak. It would be interesting to consider more complicated property estimation problems, such as the estimation of entanglement negativity, two site observables with long ranges in between, or even the estimation of system parameters in the Hamiltonian. Second, it would be more convincing if QuaDiM is examined on a more complicated physics model, such as some with non-trivial long-range entanglement. It would be more convincing if the extrapolation of QuaDiM to critical state of the physics model could be tested and examined.**
>
> **A4:** Thank you for your valuable feedbacks. It needs to be clarified that the predicting error we report for the correlations (equivalent to expectations of local observables) and entanglement entropies is averaged over different sites, including the sites with long-range distance.  Specifically:
>
> For the task of predicting the correlation function, our model outputs a symmetric matrix $V$ of size $L\times L$  for a given training sample, where $L$ is the total number of qubits. Here, $V_{ij}$ represents the predicted correlation function between the i-th and j-th qubits. Assuming the true correlation function is denoted as $V^{'}$, the MSE is $1/L^2 \sum_{ij} (V_{ij}-V^{'}_{ij})^2$, and the predicted error is the root squared value averaged over samples of test dataset. This incorporates the correlation functions associated with two quantum sites at different distances.
>
> For the task of predicting the entanglement entropies in our paper, the output is also a symmetric matrix and the predicted error is also averaged over different sites with different distances and over samples of test dataset.
>
> We will highlight this in the updated version of the paper.
>
> **Additional Numerical Results for Long-range XY Model.**
>
> To further generalize the effectiveness of our proposed model, we have supplemented our experiments with results from a more physically general and classically challenging system: the long-range XY model in the presence of a transverse field, whose Hamiltonian is given by
> $$
> H_{XY} = \sum_{i<j}J_{ij}(X_i X_j + Y_i Y_j) + \sum_j Z_j
> $$ where $J_{ij}=J_0/|i-j|^{a}$ with $a\in (1,2)$.
>
> This quantum model inherits the long-range interactions between every two quantum sites, leading to a complex dynamics which is hard to be simulated by classical computers. We restrict the system size $L=10$ due to memory limitations. The ground states of quantum systems with different physical conditions $J_{ij}$ are calculated by eigenvalue decomposition.
>
> Follow Xiao et al (npj Quantum Information, 2022), we random sample a series of $J_{ij}$ and conduct classical simulations to collect the data. This process is almost the same as in our paper (including that mentioned in A3) and we re-train the models using the collected data.
>
> The experimental results for $L=10$, with $M_{in}$ fixed at 1000 and $M_{out}\in \{1000,10000\}$, are presented below. Due to time constraints, we report the performance of our model and the baselines on the test dataset, specifically on predicting correlations and the results are listed as follows.
>
> | Method  | M=1000 | M=10000 |
> |---------|--------|---------|
> | CS      | 0.2575 | 0.0517  |
> | RBFK    | 0.1158 | -       |
> | NTK     | 0.1039 | -       |
> | RNN     | 0.2234 | 0.0502  |
> | LLM4QPE | 0.2139 | 0.0482  |
> | QuaDiM  | 0.1986 | 0.0367  |
>
> It can be observed that our proposed QuaDiM consistently outperforms the baselines. We appreciate your insightful comments, and we hope this clarification could address your concerns.
>
> **Q5: The benchmarks are not representative enough. For example, [1-3] are also works that model quantum states that should enable property estimation as downstream tasks, but are not discussed or compared.**
>
> **A5:** We sincerely appreciate the reviewer’s valuable suggestions. We have been carefully reviewed the three references provided, and we recognize the methodologies discussed in these papers offer significant insights for our future work. We are happy to cite these references in the updated version of the paper.
>
> Upon thorough analysis, we believe that the task addressed in [3] is **orthogonal** to our work, and the optimization methodologies differ substantially. Specifically, the approach in [3] is more aligned with neural network quantum states (NNQS), relying on variational Monte Carlo optimization (minimizing ground state energies) to approximate the wavefunction of the ground state. It requires two separate projection layers to represent the real and imaginary components of the complex amplitude independently.

---

> ### Author Response · Authors · 2024-11-22
> **Response to Reviewer oe1d (Part 6)**
>
> In contrast, our proposed method, while being a generative model, targets the prediction of specific properties of the quantum state without reconstructing the full density matrix. Additionally, our optimization relies on the more general gradient descent methods commonly used in machine learning, aiming to make the trained model's distribution approximate the classical joint distribution corresponding to the quantum state’s wavefunction. Therefore, we did not include [3] as a new baseline in our study.
>
> Regarding [2], we were unable to locate its implementation code. If the code becomes available, we will include it in our experiments in future updates.
>
> We would like to involve [1] as a new baseline. From a methodological perspective, [1] is quite similar to the existing baseline LLM4QPE in our paper. Both utilize autoregressive Transformers, employ KL divergence as the loss function, and learn from large quantum measurement datasets to approximate the classical joint distribution of the quantum state. Predictions of specific quantum properties are then made by sampling from this learned distribution.
>
> For a fair comparison, we configure the RydbergGPT's Transformer with 4 heads, 4 layers, and a hidden dimension of 128 which is the default configutation in our experiment. We also add a linear projection layer at the Transformer’s output (6 output units with softmax) to make the model outputs compatible with the Pauli-6 measurements used in our work. Due to time constraints, we report the RMSE of predicted correlations below for $L=100$ and $M_{in}=1000$.
>
> | Method     | $M_{out}=100$ | $M_{out}=1000$ | $M_{out}=10000$ | $M_{out}=20000$ |
> | ---------- | ------------- | -------------- | --------------- | --------------- |
> | RydbergGPT | 0.1843        | 0.0562         | 0.0170          | 0.0121          |
> | Ours       | 0.1686        | 0.0478         | 0.0125          | 0.0098          |
>
> [1] Fitzek, David, et al. "RydbergGPT." arXiv preprint arXiv:2405.21052 (2024).
>
> [2] Du, Yuxuan, et al. "Shadownet for data-centric quantum system learning." arXiv preprint arXiv:2308.11290 (2023).
>
> [3] Zhang, Yuan-Hang, and Massimiliano Di Ventra. "Transformer quantum state: A multipurpose model for quantum many-body problems." Physical Review B 107.7 (2023): 075147.
>
> **Q6: Can you discuss more on in what cases diffusion models would be strictly better than auto-regressive models, in the quantum state modeling setting?**
>
> **A6:** We sincerely thank the reviewers for the insightful questions. Our proposal to use non-autoregressive diffusion models for modeling quantum states instead of autoregressive approaches is based on the observation: **pre-defined factorization orders** (e.g., left-to-right or right-to-left orders in 1D chains, or zig-zag orders in 2D systems) may introduce biases into the model. These biases could impact the ability of variational models, including deep learning-based approaches, to accurately capture correlations among qubits and may constrain the expressivity of these models in learning quantum systems. This observation **aligns with** the perspectives presented in recent state-of-the-art works [1,2].
>
> In contrast, non-autoregressive diffusion models encourage to **avoid this manually defined ordering bias** during both the training and sampling phases. They directly model the **joint probability distribution** corresponding to the quantum state and sample from it, such that the configurations of qubits are generated **simultaneously**.
>
> Our preliminary hypothesis is that, for both 1D systems (as specifically studied in this paper) and 2D topologies of quantum systems, non-autoregressive diffusion models offer a **general** advantage. We hope that our work will inspire future research to examine more specific cases and instances to further validate this hypothesis. We remain open to continued discussions with the reviewers to explore this open-ended question in greater depth.
>
> [1] Bortone M, Rath Y, Booth G H. Impact of conditional modelling for a universal autoregressive quantum state[J]. Quantum, 2024, 8: 1245.
>
> [2] Ibarra-García-Padilla E, Lange H, Melko R G, et al. Autoregressive neural quantum states of Fermi Hubbard models[J]. arXiv preprint arXiv:2411.07144, 2024.

---

> ### Author Response · Authors · 2024-11-22
> **Response to Reviewer oe1d (Part 7)**
>
> **Q7: I am not completely clear on how continuous latent vectors generated by QuaDiM are decoded back into measurement outcomes. Can you elaborate more on this?**
>
> **A7:** Thank you for your feedback. To decode the continuous latent representations output by QuaDiM (denoted as $H \in \mathbb{R}^{B \times L \times d}$, $B$ is the batch size and $d$ is the hidden dimension) back into discrete measurement outcomes (denoted as $M \in \mathbb{Z}^{B \times L}$ and each entry $M_{ij}\in \\{1,\ldots,6\\}$), the decoding step at $t = 0$ involves a **softmax+sampling** operation. Specifically, we first apply a projection layer (a linear transformation) to transform $H$ into $Z \in \mathbb{R}^{B \times L \times 6}$, where 6 corresponds to the total number of possible measurement outcomes. Next, we apply the softmax operation along the last axis of $Z$ (converting the continuous latent representation into probabilities), and then sample from these probabilities to obtain the measurement outcomes $M \in \mathbb{R}^{B \times L}$.
>
> By the way, at denoising step with $t > 0$, we incorporate the rounding trick proposed in [1] (as discussed in Section 3.2.3 of the paper). This is because we empirically found it improves the convergence speed of the model. This step also involves decoding the continuous latent representation into discrete measurement outcomes. Below, we provide a brief description of the core of the rounding trick; for more details, please refer to the original paper.
>
> The main task remains to decode the continuous latent representations output by QuaDiM $H \in \mathbb{R}^{B \times L \times d}$ back into discrete measurement outcomes $M \in \mathbb{Z}^{B \times L}$. A typical metric—the $l_2$-norm distance is used to decode. Specifically, a unique $d$-dimensional embedding is assigned to each of the six possible discrete measurement outcomes. This embedding matrix $E \in \mathbb{R}^{6 \times d}$ is updated during model training and stored in memory.
>
> For each $d$-dimensional vector in the continuous latent representation $H$ produced by QuaDiM, we compute its $l_2$-norm distance to the six embedding vectors. The measurement outcome corresponding to the nearest embedding vector (i.e., the one with the smallest distance) is selected as the decoding result.
>
> This process (when $t > 0$) can be viewed, to some extent, as performing $k=1$ $k$-nearest neighbor (KNN) on the continuous latent representations, and is highly efficient in python implementation when vectorization techniques are employed.
>
> [1] Li X, Thickstun J, Gulrajani I, et al. Diffusion-lm improves controllable text generation[J]. Advances in Neural Information Processing Systems, 2022.
>
> **Q8: Why QuaDiM will generate measurements consistently sampled from a valid quantum state without violating structure constraints. How is this enforced in the learning algorithm?**
>
> **A8:** Thanks for your insightful comments. To help the review for a better understand, we first provide a brief review of necessary notations (please refer to the preliminary section in the paper for details).
>
> Given the trained model's output $Z$, we obtain a sample $(\sigma_1, \sigma_2, \ldots, \sigma_L)$ by sampling from $Z$. The sampling process corresponds to a joint distribution $p(\sigma_1, \sigma_2, \ldots, \sigma_L)$ that should satisfy the normalized contraints of the quantum wave function (arises from the Born rule in quantum mechanics). It means that we should prove the following equation:
> $$\sum_{\sigma_1}\cdots\sum_{\sigma_L}p(\sigma_1, \ldots, \sigma_L)= \sum_{\sigma_1}\cdots\sum_{\sigma_L}||\Phi(\sigma_1,\ldots,\sigma_L)||^2 = 1$$ where $\Phi(\sigma_1,\ldots,\sigma_L)$ is the complex-valued amplitude.
>
> To prove that generating measurements from the probability distribution specified by the output $Z$ of QuaDiM's last layer satisfies the classical joint distribution $p$ corresponding to the quantum wave function $|\mathbf{\psi}\rangle$, we need to establish that $Z$ represents $p$ and that sampling process preserves this representation.
>
> As disscussed in our response in $A7$, we mentioned that before sampling, a projection layer and a softmax operation is performed to output $Z \in \mathbb{R}^{B \times L \times 6}$, where 6 corresponds to the total number of possible measurement outcomes. This guarantees that $\sum_{k} Z_{b, l, k} = 1$ for each $b$ and $l$.

---

> ### Author Response · Authors · 2024-11-22
> **Response to Reviewer oe1d (Part 8)**
>
> Sampling from $Z$ involves selecting outcomes $k$ at each position $l$ according to $Z_{b, l, k}$. For each sample $b$, the joint probability of a single measurement string is
> $$p(\sigma_1=k_1,\ldots,\sigma_L=k_L) = \prod_{l} Z_{b, l, k_l} =\prod_{l} p(\sigma_1=k_1)$$ The first equation holds due to the non-autoregressive nature of the diffusion model.  Since $\sum_{k} Z_{b, l, k} = \sum_{k} p(\sigma_l=k)=1$, the following equation then be hold:
> $$\sum_{\sigma_1}\cdots\sum_{\sigma_L}p(\sigma_1, \ldots, \sigma_L)=\sum_{\sigma_1}\cdots\sum_{\sigma_L}\prod_l p(\sigma_l) = \prod_l\sum_{k}p(\sigma_l=k)= 1$$ The proof then completes.
>
>
> **Q9: Does QuaDiM extrapolate to unseen system parameter distribution, likely with disjoint support as the training distribution, e.g., trained $x\in [0,0.8]$, tested on $x\in [0.9,1]$?**
>
> **A9:** We appreciate the reviewer’s suggestion and apologize for not providing sufficient clarity regarding the distribution of data used in the paper. We would clairify it more clear in the revised vesion of the paper.
>
> In our current setup, the physical parameters $x$ for both training and fine-tuning are sampled from the same distribution [0, 2]. Addressing out-of-distribution (OOD) data is a direction we consider for future work. Even though, we have conducted some preliminary experiments on OOD data to provide additional insights into our model's performance.
>
> Specifically, we divide the sampled physical parameters $x$ into two segments: the training set is limited to [0, 1.5], while the test set exclusively spans [1.5, 2]. In alignment with setting of the paper, we set $N^{tr} = 100$ and $N^{te} = 20$. We report the RMSE of predicted correlations for both the sota baseline LLM4QPE, and our model QuaDiM under both OOD and non-OOD (both training and fine-tuning are sampled from the same distribution [0, 2]) conditions, with $M_{in} = 1000$ and $M_{out} = 10000$. The results are presented in the table below.
>
>
> |         | L=70   |        | L=100  |        |
> |---------|--------|--------|--------|--------|
> | Method  | no OOD | OOD    | no OOD | OOD    |
> | LLM4QPE | 0.0155 | 0.0526 | 0.0152 | 0.0598 |
> | QuaDiM  | 0.0117 | 0.0417 | 0.0125 | 0.0465 |
>
>
> From the results, it is evident that our model continues to outperform the baseline to some extent even under OOD conditions. However, the prediction accuracy drops significantly for both.
>
> Incorporating additional strategies—such as embedding prior knowledge of physical systems into the model design [1]—may further enhance the model's generalization capability on OOD data. Whether our proposed model retains its benefits under broader OOD scenarios remains an open question, which we plan to explore in future work.
>
> We hope this response could addresse your concerns. Please feel free to reach out with any further questions regarding OOD problems.
>
> [1] Ma H, Sun Z, Dong D, et al. Tomography of Quantum States from Structured Measurements via quantum-aware transformer[J]. arXiv preprint arXiv:2305.05433, 2023.
>
> **Q10: Can you comment on QuaDiM's training efficiency? First, it seems that a relatively large $M_{in}$ is required for learning one ground state to good accuracy, also $M_{out}$ is not a small number as well. Can $M_{in}$ and $M_{out}$ be improved? Second, how many distinct pairs of (ground state, measurements) are needed for learning the mapping between Hamiltonian parameter and system ground state correctly, and how does this scale with the number of qubits?**
>
> **A10:** Thank you for the thoughtful and constructive comments. It is worth noting that predicting properties of quantum states using deep learning techniques is indeed a cutting-edge research area, and the **efficiency of deep learning techniques in this domain remains an open question**. We would like to briefly share some of our observations here and maintain an open stance for further discussion with the reviewer.
>
> The ultimate goal in this field is to **accurately characterize the properties of quantum states using as few identical copies and measurements as possible**. In our context, the term $M_{in}$ refers to the number of measurements required for training the model.
>
> On the other hand, $M_{out}$ denotes the number of samples drawn from the trained generative model. It serves as a measure of how well the model, trained with a given $M_{in}$, approximates the classical probability distribution corresponding to the quantum state. In Table 1, Table 2, and Figure 5 of our paper, we have reported the performance of our model compared to baselines across different values of $M_{out}$ and $M_{in}$.

---

> ### Author Response · Authors · 2024-11-22
> **Response to Reviewer oe1d (Part 9)**
>
> Over the past few years, a series of theoretical [1,2] and experimental [3,4] studies have emerged in this field, with $M_{in}$ and $M_{out}$ **often reaching orders of thousands—comparable to the scales discussed in our paper**. Developing an effective method to characterize quantum states while minimizing $M_{in}$ and $M_{out}$ is a highly non-trivial challenge and has remained an active research direction. The specific analysis is as follows:
>
> From a theoretical perspective [1,2], for learning and predicting the correlations (equivalent to the expectation of local observables) and two-body entropy investigated in our paper, **the required $M_{in}$ and $M_{out}$ are independent of the number of qubits $L$**. However, in general cases, the required scales **increase exponentially with the size of the subsystem A** on which the operator acts. Meanwhile, the number of distinct ground states in the training dataset (fixed to 100 in our work) scales logarithmically with the number of qubits.
>
> From a numerical perspective, recent work [4] based on autoregressive models have explored new deep learning methodologies to reduce $M_{in}$ and $M_{out}$ and achieves SOTA results. In our paper, we approached this problem from the perspective of the **non-sequential nature of the distribution corresponding to quantum states**. Our numerical experiments demonstrated that the non-autoregressive diffusion model achieves a reduction in $M_{in}$ compared to SOTA autoregressive model when predicting the correlation and entropy of the 1-D Heisenberg model (as shown in Figure 5). Furthermore, under the same $M_{out}$, our model achieves better predictive performance (as presented in Tables 1 and 2).
>
> We sincerely hope these explanations address the reviewer's concerns and welcome any further discussions to refine and enhance our work.
>
> [1] Huang H Y, Kueng R, Torlai G, et al. Provably efficient machine learning for quantum many-body problems[J]. Science, 2022.
>
> [2] Lewis L, Huang H Y, Tran V T, et al. Improved machine learning algorithm for predicting ground state properties[J]. Nature Communications, 2024.
>
> [3] Xiao T, Huang J, Li H, et al. Intelligent certification for quantum simulators via machine learning[J]. npj Quantum Information, 2022.
>
> [4] Tang Y, Xiong H, Yang N, et al. Towards LLM4QPE: Unsupervised Pretraining of Quantum Property Estimation and A Benchmark. ICLR, 2024.

---

> > ### Comment · Reviewer_oe1d · 2024-11-25
> > **Minor comments on presentation**
> >
> > Thank you for the detailed responses to all my questions and most of my concerns have been addressed already. I have several follow-up comments regarding some minor presentation issues.
> >
> > - Regarding answer A3, thank you for the clear explanation and that indeed solves my questions. I suggest that you also add the detailed protocol for generating Hamiltonian parameter $x$ (i.e., sampling from uniform distribution between 0 and 2), to improve **clarity and paper reproducibility**. I don't think it has been mentioned in the current paper draft yet (but correct me if I am wrong). You should also clarify that this is in distribution generalization, as it contrasts with the OOD generalization experiments mentioned in answer A9.
> >
> > - Regarding answer A5, it's nice to see the comparison with RydbergGPT. I suggest that you also add this benchmark to the main experiments in the main text, or at least discuss it (same apply to Shadow-Net) to make the numerical experiments section more convincing.
> >
> > - Regarding answer A9, it's nice to see additional results on OOD generalization. I also suggest that you present the detailed setting in the appendix and emphasize that this is out-of-distribution generalization in contrast with previous results. From my perspective, this is way more impressive than in-distribution generalization and should be included in the main text if space permits.
> >
> > In general, I think this work is a good contribution to the field of deep learning and quantum sciences. I agree that despite there's not much innovation on the diffusion model algorithm, it's still a clever application to an important task in quantum sciences. I am also looking forward to seeing its applications in learning more complicated conditions instead of just Hamiltonian parameters.
> >
> > Due to the above, I have raised my rating and modified the scoring.

---

> > > ### Author Response · Authors · 2024-11-25
> > >
> > > Dear Reviewer oe1d,
> > >
> > > We deeply appreciate your thoughtful feedback and your willingness to consider raising your score. Your specific suggestions have been instrumental in enhancing our manuscript.
> > >
> > > Thank you again for your valuable review and support of our work.
> > >
> > > Best regards

---

### Author Response · Authors · 2024-11-25
**General Response to Reviewers**

Dear Reviewers,

As the rebuttal phase comes to a close, **we would like to kindly request any further clarifications or responses from the reviewers regarding our updates and clarifications**. Your additional feedback would be immensely valuable in helping us ensure the highest quality of this work and address any remaining concerns you may have.

We also extend our gratitude to the reviewers for their detailed feedback, constructive criticism, and valuable suggestions. The insightful reviews provided a broader perspective and helped refine our work. Overall, the reviewers recognized the novelty of QuaDiM in its application to quantum state property estimation (QPE) using a non-autoregressive diffusion model. Specifically, the reviewers deemed our approach “**important for AI4Science area**” (kHhj), “**effective**” for large systems (9ozo, MPMV), and highlighted its strength in addressing the limitations of autoregressive methods while demonstrating **state-of-the-art** performance in predicting quantum properties such as correlation and entanglement entropy with the system size **up to 100 qubits** (oe1d, 9ozo, MPMV).

### Contributions and Innovations
To address concerns regarding the novelty and audience fit for ICLR, we reiterate the key contributions and innovations of QuaDiM:
1. **Non-Autoregressive Diffusion Model for QPE:** QuaDiM introduces a novel application of conditional diffusion models to quantum systems, encouraging to eliminate the sequential bias inherent in autoregressive approaches, by allowing simultaneous sampling of qubit configurations and accurate property estimation.
2. **Scalability to Large Quantum Systems:** QuaDiM successfully scales to systems with up to 100 qubits (up to $2^{100}$ dimension), which represents a significant advance in quantum many-body problem.
3. **Generalization Across Parameters:** QuaDiM demonstrates strong extrapolation capabilities, effectively predicting quantum properties for unseen Hamiltonian parameters during inference, as detailed in the additional OOD experiments.
4. **Adaptation of Diffusion Models to Quantum Systems:** Unlike classical diffusion model applications, QuaDiM adapts to the non-sequential and entangled nature of quantum data, introducing techniques like token embeddings and specialized denoising schedules.

### Value of Deep Learning in Quantum Many-Body Problems
QPE is pivotal in quantum computing and condensed matter physics. It directly impacts the ability to characterize quantum systems, crucial for advancing quantum technologies. By leveraging deep learning, QuaDiM addresses the challenges of modeling complex quantum states, enabling scalable and efficient property predictions, even under limited measurement data scenarios. This work bridges machine learning and quantum physics, highlighting the potential for interdisciplinary innovation.

### Rebuttal Highlights and Clarifications
In response to reviewer feedback, we have strengthened our manuscript by addressing the following concerns:
1. **Empirical Evidence Supporting Non-Autoregressive Models:** We clarified how autoregressive methods could introduce bias through pre-defined qubit orderings and demonstrated the superior performance of QuaDiM through additional experiments under various ordering schemes.
2. **Experimental Setup and Generalization:** We expanded the explanation of parameter generation and dataset splitting, highlighting the model’s ability to generalize to unseen parameter distributions. Results from new OOD experiments further support this claim.
3. **Extended Benchmarks:** We included comparisons to new baselines, such as RydbergGPT, as suggested by reviewers, while highlighting the orthogonality of other approaches like neural quantum states.
4. **Additional Physics Models:** We supplemented the experiments with results on the long-range XY model, further showcasing QuaDiM’s robustness in capturing complex quantum interactions.
5. **Additional Measurements Protocols:** We supplemented the experiments with results on another type of IC-POVM: the tetrahedral POVM, showcasing QuaDiM’s feasibility in different measurement strategy generally used in quantum physics.
6. **Numerical Ablations:** We conducted ablation studies on the number of denoising steps and embedding dimensions, balancing computational efficiency with model performance.
7. **Clarifications on POVM Measurements and Physical Constraints:** We added an appendix section explaining the mapping of wavefunctions to classical joint probabilities for readers who may not familiar with quantum computing.


We sincerely hope that the additional experiments, clarifications, and revisions provided address your concerns. If there are any questions or further points you would like us to elaborate on, we would greatly appreciate your guidance. Your insights have been invaluable to the development of this work, and we look forward to any further feedback you may have during the remaining rebuttal period.

---

### Comment · Area_Chair_ZGFt · 2024-11-26

Dear Reviewers 9ozo, MPMV,
If not already, could you please take a look at the authors' rebuttal? Thank you for this important service.
-AC

---

### Meta-Review · Area_Chair_ZGFt · 2024-12-19

**Metareview:**

The paper proposes to apply diffusion model to create a framework for learning the ground states of quantum systems conditioned on system parameters. It also enables the estimation of properties of the quantum state. Empirical successes, including remarkable scalability to large quantum systems, were demonstrated using classically simulated data on the 1D anti-ferromagnetic Heisenberg model. Overall, both reviewers and I feel that this work, as a submission to Area "applications to physical sciences", makes physical contributions that are worthy acceptance, especially given that applications of GenAI to quantum many-body problems are relatively nascent but practically useful.

**Additional Comments On Reviewer Discussion:**

Most of the reviewers' concerns seemed to have resolved post rebuttal.

---

### Decision · Program_Chairs · 2025-01-22

Accept (Poster)